# Local orchestration of distributed functional patterns supporting loss and restoration of consciousness in the primate brain

Andrea I. Luppi [1,2,10] ✉, Lynn Uhrig[3,4,10], Jordy Tasserie [3,5,10], Camilo M. Signorelli[3,6,7], Emmanuel A. Stamatakis [1], Alain Destexhe [8,11], Bechir Jarraya [3,9,11] & Rodrigo Cofre [8,11] ✉

A central challenge of neuroscience is to elucidate how brain function supports consciousness. Here, we combine the specificity of focal deep brain stimulation with fMRI coverage of the entire cortex, in awake and anaesthetised non-human primates. During propofol, sevoflurane, or ketamine anaesthesia, and subsequent restoration of responsiveness by electrical stimulation of the central thalamus, we investigate how loss of consciousness impacts distributed patterns of structure-function organisation across scales. We report that distributed brain activity under anaesthesia is increasingly constrained by brain structure across scales, coinciding with anaesthetic-induced collapse of multiple dimensions of hierarchical cortical organisation. These distributed signatures are observed across different anaesthetics, and they are reversed by electrical stimulation of the central thalamus, coinciding with recovery of behavioural markers of arousal. No such effects were observed upon stimulating the ventral lateral thalamus, demonstrating specificity. Overall, we identify consistent distributed signatures of consciousness that are orchestrated by specific thalamic nuclei.

Despite significant progress in recent years, understanding how the activity and connectivity of the brain support and causally determine different states of consciousness remains a major challenge for modern neuroscience[1,2]. The traditional approach to the neuroscience of consciousness has been to look for specific brain regions (and more recently, collections of brain regions in the form of circuits and spatially discontiguous networks) that support consciousness[3,4]. However, there is now increasingly compelling evidence for a role of distributed, large-scale patterns of cortical organisation supporting cognitive function, dysfunction[5–10] and consciousness[11–14]. Recent work has demonstrated that multiple pathological and pharmacological perturbations of consciousness induce consistent reorganisation of the brain's functional architecture along functional and anatomical axes of cortical organisation[15–18]. This distributed approach goes beyond viewing the brain in terms of brain regions and fixed networks, emphasising the role of dynamics and functional organisation. For instance, when consciousness is lost due to anaesthesia or disorders of consciousness, the functional interactions between brain regions

[1]Division of Anaesthesia and Department of Clinical Neurosciences, University of Cambridge, Cambridge, UK. [2]Montreal Neurological Institute, McGill University, Montreal, QC, Canada. [3]Cognitive Neuroimaging Unit, CEA, INSERM, Université Paris-Saclay, NeuroSpin Center, 91191 Gif-sur-Yvette, France. [4]Department of Anesthesiology and Critical Care, Necker Hospital, AP-HP, Université de Paris Cité, Paris, France. [5]Center for Brain Circuit Therapeutics, Department of Neurology, Brigham & Women's Hospital, Harvard Medical School, Boston, MA, USA. [6]Laboratory of Neurophysiology and Movement Biomechanics, Université Libre de Bruxelles, 1070 Brussels, Belgium. [7]Department of Computer Science, University of Oxford, Oxford, 7 Parks Rd, Oxford OX1 3QG, UK. [8]Institute of Neuroscience (NeuroPSI), Paris-Saclay University, Centre National de la Recherche Scientifique (CNRS), Gif-sur-Yvette, France. [9]Department of Neurology, Hopital Foch, 92150 Suresnes, France. [10]These authors contributed equally: Andrea I. Luppi, Lynn Uhrig, Jordy Tasserie. [11]These authors jointly supervised this work: Alain Destexhe, Bechir Jarraya, Rodrigo Cofre. ✉e-mail: al857@cam.ac.uk; rodrigocofre@gmail.com

("functional connectivity") become increasingly similar to the pattern of direct anatomical connections between regions ("structural connectivity")[19–24]. More recently, evidence has shown that the organisation of distributed brain function is perturbed in a scale-specific manner under several pharmacological and pathological perturbations of consciousness in humans[15,25], increasing the dependence of brain function on structural connectivity across scales.

Therefore, the first goal of the present investigation is to establish whether these distributed markers of consciousness that have been recently identified in the human brain, can be generalised to the non-human primate brain, and whether they behave consistently across anaesthetics with distinct molecular mechanisms. To this end, we leverage a previously published dataset of resting-state functional MRI (fMRI) data from non-human primates in the awake state, and after the animals have been rendered behaviourally unresponsive by three different anaesthetics (sevoflurane, propofol, ketamine). Since non-human primates cannot use language to report their subjective experience, we rely on behavioural measures of arousal as our proxy for consciousness, as is common practice when working with animal models of anaesthesia, and also widely used in clinical practice[26,27]. Nonetheless, we acknowledge that unresponsiveness is not equal with unconsciousness[28,29]. We return to this issue in the Discussion.

The recent evidence for distributed functional bases of consciousness, needs to be reconciled with additional evidence that selective stimulation of the macaque intralaminar nuclei of the thalamus can awaken monkeys from anaesthesia, despite continuous anaesthetic infusion[21,30–33], suggesting a potential causal involvement of this thalamic nuclei, and specifically central thalamus, in supporting consciousness. Similar results were also reported in rodents, with electrical stimulation of the centro-median thalamus inducing awakening from anaesthesia in rats[34]. The activity and connectivity of the thalamus have been repeatedly associated with pathological and pharmacological perturbations of consciousness, in both humans and other species[30–33,35–55]. This raises the intriguing question: does the central thalamus act as a local controller of distributed functional patterns? A positive answer would contribute to reconciling the distributed and location-specific perspectives on the neural bases of consciousness.

Therefore, our second goal is to determine whether the reorganisation of distributed brain function induced by anaesthesia can be reversed by targeted electrical stimulation of the central thalamus, concomitantly with restoration of behavioural evidence of consciousness. To address this question, we leverage the experimental accessibility of animal models, analysing a unique recently published dataset of macaque resting-state fMRI acquired under deep propofol anaesthesia with and without direct intervention on the thalamus, in the form of deep brain stimulation (DBS) of the centro-median (CT) or ventral-lateral thalamus (VT)[21]. Electrical stimulation of the central thalamus has become an established approach for modulating behavioural and neural markers of consciousness in non-human primates[21,30,31]. In this dataset that combined thalamic DBS with fMRI, it was shown that high-amplitude CT stimulation restores behavioural markers of arousal (eye opening, movement, reflexes and exploratory behaviour) that were suppressed by anaesthesia, as well as restoring neural signatures including dynamic uncoupling of functional from structural connectivity, the complexity of EEG signals, and neural processing of auditory stimuli beyond sensory cortices, which have been argued to be signatures of consciousness[11,12,56]. This effect was greatly diminished at lower amplitude DBS, and entirely absent upon low and high-amplitude VT stimulation, demonstrating an exquisite level of specificity[21].

Taking advantage of the broad spatial coverage afforded by fMRI, which is ideally suited for the investigation of distributed functional organisation of the brain, here we consider three distinct markers of distributed function. First, we consider the brain's hierarchical organisation across scales, by studying the principal gradient of functional connectivity[5]. The brain's intrinsic functional organisation can be represented in terms of continuous, spatially overlapping whole-brain patterns, termed functional gradients[5,57]. Analogous to the principal components of PCA, gradients map functional connectivity to a low-dimensional space where proximity indicates functional similarity. Each gradient corresponds to a dimension in this space. Along each dimension (represented by an eigenvector), regions will be located closer in space, the more similar they are in terms of their functional connectivity with the rest of the brain. The regions whose FC is most different along that dimension constitute opposite extremes that are maximally functionally different. In particular, the principal functional gradient (eigenvector associated with the principal eigenvalue) captures the direction of maximal spatial variation in functional organisation: its range can be interpreted as reflecting the distance between the two extremes of the cortical processing hierarchy, reflecting the depth of information processing[18,58]. This principal gradient can also be reshaped by pharmacological intervention[15,18,58], rendering this approach a promising perspective to consider in the present study.

Second, we study the hierarchical integration of functional signals across scales[59]. Whereas the range of the principal functional gradient reflects the putative depth of the information-processing hierarchy, hierarchical organisation can also manifest in terms of nested relationships between the system's parts at different scales: lower elements in the hierarchy are recursively combined to form the higher elements[60]. This perspective makes it possible to consider the interplay of integration and segregation of brain signals across scales - which is a central feature of prominent scientific accounts of consciousness[11,12]. Classical graph-theoretic measures such as small-worldness and modularity quantify integration and segregation at a single scale, making them inadequate to capture these properties across multiple hierarchical modules[59,61,62]. Instead, we can obtain insight into the hierarchical relationships between different scales of distributed activation in the brain by going beyond the principal gradient alone (or first 2–3 gradients), and instead considering all functional eigenmodes, as recently introduced by[60]. This is achieved by characterising the concordance or discordance of regions' allegiance across eigenmode scales, corresponding to regions that are jointly activated (same sign) or alternate (opposite sign) at a given scale. This process results in a nested, modular structure that identifies a hierarchical sub-division of the functional connectome into nested modules, up to the level where each module coincides with a single region, indicative of completely segregated activity. Hierarchical integration and segregation can then be quantified by the relative prevalence of the different eigenmodes, indicated by their associated eigenvalues[60].

Finally, functional brain activity and connectivity unfold over the network of physical white matter pathways between brain regions: the structural connectome. Therefore, for our final investigation we jointly consider brain structure and function, by leveraging the mathematical framework of "harmonic mode decomposition"[63] to decompose brain activity into distributed patterns of structure-function coupling: the "harmonic modes" of the structural connectome[63,64]. Through this decomposition, we quantify the extent to which brain activity is constrained by the underlying network of structural connectivity. Each harmonic mode is a cortex-spanning activation pattern (eigenmode of the structural connectome) characterised by a specific granularity. The use of connectome-specific harmonic decomposition of cortical activity allows us to quantify the contribution of structural organisation to brain activity, across different spatial granularities scales: from large-scale to fine-grained. This approach therefore goes beyond previous investigations that assessed the similarity of structural and functional connectivity at a single scale[19,20,22,23]. Additionally, this approach is of particular interest because recent results in humans have shown that distributed harmonic patterns of structure-function

dependence relate with human consciousness[25], so we aim to investigate here the cross-species generalisation of these results, and their potential susceptibility to thalamic stimulation.

Comparing the brain-wide effects of different anaesthetics enables us to characterise which aspects of the brain's distributed functional organisation are consistently targeted by anaesthetics, despite their distinct molecular mechanisms. By further assessing whether different aspects of distributed brain function are restored upon concomitant restoration of behavioural arousal (our indicator of consciousness) by subregion-specific thalamic stimulation, we can further strengthen their association with consciousness. Specifically, a genuine neural correlate of consciousness should not only be compromised under anaesthesia: it should also be restored when consciousness (here indicated by arousal) is restored, even in the presence of continuous anaesthetic infusion.

This dual intervention—pharmacology and electrical stimulation—provides us with a unique opportunity to study functional changes that are observed during loss of consciousness and reappear upon recovery, despite continuous anaesthetic infusion. By combining the broad spatial coverage of fMRI with stimulation of specific nuclei of the thalamus, we can simultaneously consider both the role of specific regions, and distributed functional organisation. Through this approach, we show that multiple anaesthetics impact distributed cortical signatures of consciousness, while also demonstrating that they are susceptible to orchestration specifically by the central thalamus, thereby linking the distributed and spatially-localised perspectives. To foreshadow our results, we show that general anaesthesia increases the harmonic energy, reflecting the interdependence of functional connectivity on underlying anatomical structure, and reduces the gradient range and hierarchical integration, reflecting the processing capacity and the network integration respectively. These effects are reversed by central thalamic stimulation, which is associated with increased behavioural responsiveness under general anaesthesia.

## Results

Here, we studied distributed functional patterns in the macaque cerebral cortex. Specifically, we measure how these patterns are reorganised under different anaesthetic agents and by direct stimulation of different thalamic nuclei.

### Anaesthetic-induced collapse of the principal functional gradient and restoration by thalamic stimulation

Our first approach to study distributed brain function and its alteration under anaesthesia is through the lens of the principal gradient of functional connectivity. Functional gradients are built in an analogous way as PCA's principal components, but rely on nonlinear techniques (such as Diffusion Map Embedding) whereas PCA is linear[5,57]. Both techniques enable data compression by highlighting the most relevant patterns, encoded in the form of eigenmodes (eigenvectors), which correspond to the dimensions of spatial variation of functional connectivity. Along each gradient (eigenmode), regions whose FC with the rest of the brain is more similar will be located closer—and conversely, the extremes of the gradient correspond to regions whose FC with the rest of the brain is maximally functionally different[5,57]. In particular, the principal functional gradient captures the direction of maximal spatial variation in functional organisation. Its range (difference between maximal and minimal values) has been interpreted as reflecting the distance between the two extremes of the cortical processing hierarchy, reflecting the depth of information processing[58]. Here, we hypothesise that this depth should be reduced in the unconscious brain, when processing of information is manifestly impaired. Indeed, such an observation was very recently reported in humans, with different gradients of functional connectivity collapsing as a result of different pharmacological and pathological perturbations, including anaesthesia[15].

Here, we delineate gradients using the common nonlinear dimensionality reduction technique known as diffusion map embedding (DME; see "Methods"). We observed a significant effect of anaesthesia on the principal gradient of macaque functional connectivity ($F_{(5, 113)} = 5.87$, $p = 7.26 \times 10^{-5}$), and in the DBS dataset, a significant effect of stimulation condition ($F_{(5, 150)} = 21.49$, $p = 3.49 \times 10^{-16}$). The range of the principal gradient of macaque functional connectivity is significantly reduced by both ketamine and deep propofol anaesthesia (Fig. 1 and Supplementary Data 1).

In the DBS dataset, behavioural assessment outside the scanner indicated that in both monkeys (N and T), the clinical arousal score (see "Methods") dropped from the maximum score of 11 to the minimum score of 0, when switching from wakefulness to deep propofol anaesthesia[21] (see Fig. S1 for exact electrode locations). Low-amplitude DBS of the centro-median thalamus increased the score to 3/11: a level comparable to light anaesthesia, restoring some eye opening, corneal reflex, and some search for external cues (Fig. S2)[21]. Under high CT-DBS, both animals woke up, opened their eyes spontaneously, and had spontaneous limb movements and breathing activity, thus regaining a high clinical score (9/11). Meanwhile, under low or high VT-DBS, the clinical score remained unchanged at the level of deep anaesthesia (0/11)[21].

At the neural level, comparing wakefulness with propofol anaesthesia without stimulation ("off" condition) replicated the collapse of the principal gradient's range observed in the Multi-Anaesthesia dataset (Fig. 2 and Supplementary Data 2). Crucially, despite continuous anaesthesia, we found that the range of the principal gradient could be significantly increased towards the direction of wakefulness by low amplitude stimulation of the centro-median thalamus (CT). In fact, the evidence provided strong Bayesian support for the null hypothesis of no difference between Awake and low-amplitude CT stimulation, for medium to large effect sizes (Fig. 2 and Fig. S3A). In other words, the evidence is sufficient to conclude that low-amplitude CT stimulation restored gradient range values back to awake levels. Surprisingly, although CT stimulation at high amplitude induced a stronger behavioural response than low intensity (Fig. S2)[21], the Bayesian evidence was inconclusive to determine a difference between high CT stimulation and no stimulation (Supplementary Data 2 and Fig. S3B). In contrast, both low and high amplitude of stimulation of the control site (VT, ventral-lateral thalamus) did not merely fail to show a statistically significant effect against no-stimulation anaesthesia: they provided evidence explicitly supporting the null hypothesis (Supplementary Data 2 and Fig. S3C, D), indicating that they had no effect on the principal gradient's range.

We focused here on the range spanned by the gradient, because it is believed to reflect the depth of processing across the cortical hierarchy, and it was recently shown to be affected by several pharmacological and pathological perturbations of consciousness[15,18,58]. Indeed, we show that a different measure, the ratio of the eigenvalue associated with the principal gradient over the sum of all eigenvalues, does not display changes with anaesthesia (Fig. S4 and Supplementary Data 3, 4). Our hypothesis revolves around the principal gradient because it is of primary importance, both in terms of explaining most of the variance in the functional connectivity, and in terms of recent literature relating it to alterations of consciousness in humans[15]. Nonetheless, here we also considered the second gradient, showing that its range exhibits additional sensitivity to deep sevoflurane anaesthesia, in addition to ketamine and deep propofol (Fig. S5A and Supplementary Data 5); and it displays similar susceptibility to low-amplitude CT stimulation (Fig. S5B and Supplementary Data 6). As a result, we also considered the dispersion of the first three gradients (sum of squared Euclidean distance of all regions to the overall centroid in the 3D cortical gradient space), as done by Huang and colleagues[15], showing analogous results (Fig. S6 and Supplementary Data 7, 8).

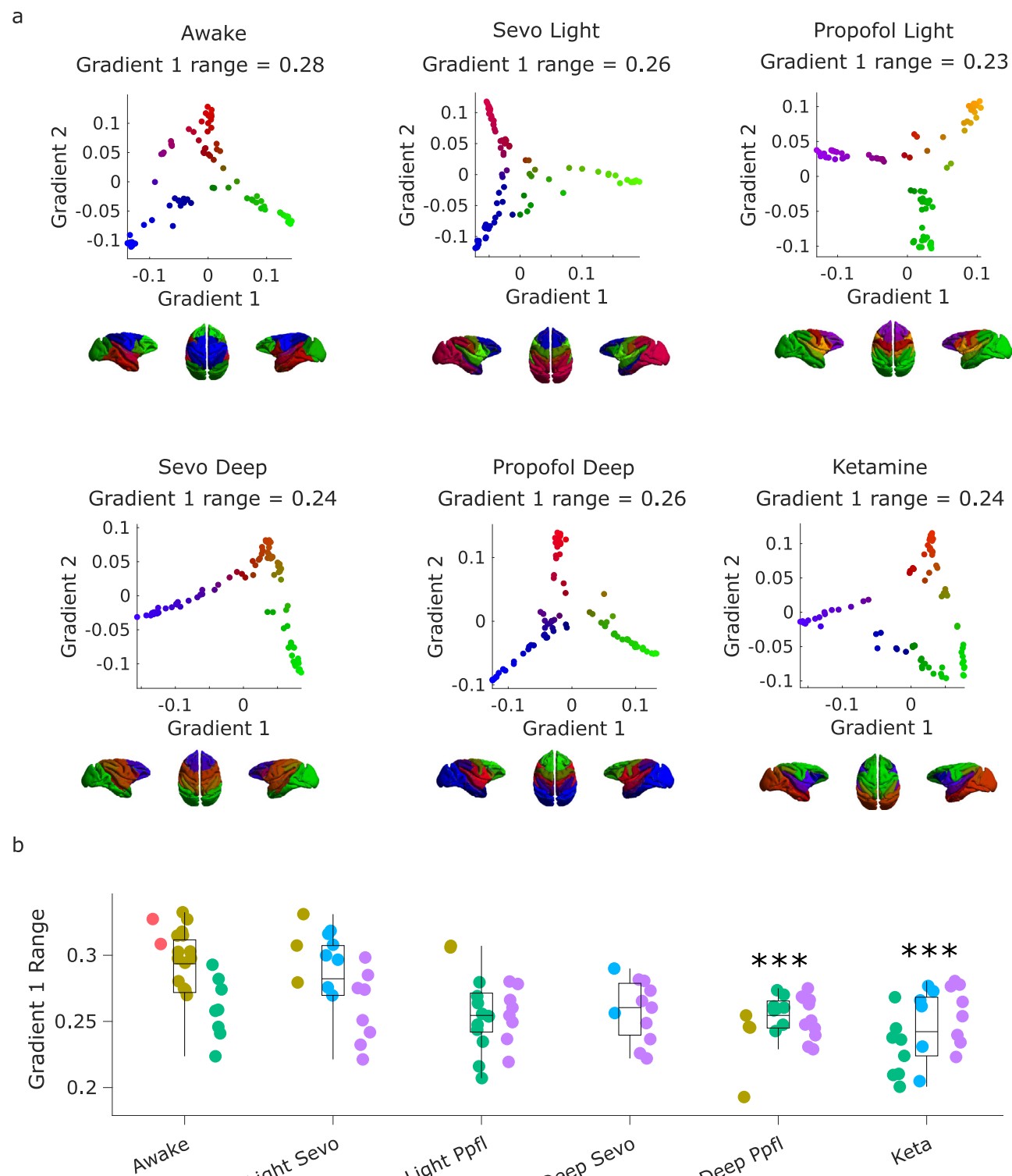

**Fig. 1 | Anaesthetic-induced collapse of the principal gradient of macaque functional connectivity. a** | Scatter plots show the first two principal gradients of macaque functional connectivity (obtained from diffusion graph embedding: see "Methods") for the group-averaged FC matrix of the awake condition, and each anaesthetised condition. The gradients are also plotted on the cortical surface of the macaques, with colour representing the position of each region along each gradient. **b** | The range of the principal gradient of macaque functional connectivity across wakefulness and different anaesthetic conditions. Box plots: central line,

median; box limits, upper and lower quartiles; whiskers, 1.5× interquartile range; dots of the same colour are provided by the same animal. ***$p < 0.001$ from linear mixed effects modelling (two-sided, FDR-corrected), compared against Awake condition; see Supplementary Data 1 for statistical results. $N = 24$ runs from 3 animals for Awake; 18 runs from 3 animals for Light Sevoflurane; 21 runs from 3 animals for Light Propofol; 11 runs from 2 animals for Deep Sevoflurane; 23 runs from 3 animals for Deep Propofol; 22 runs from 3 animals for Ketamine anaesthesia.

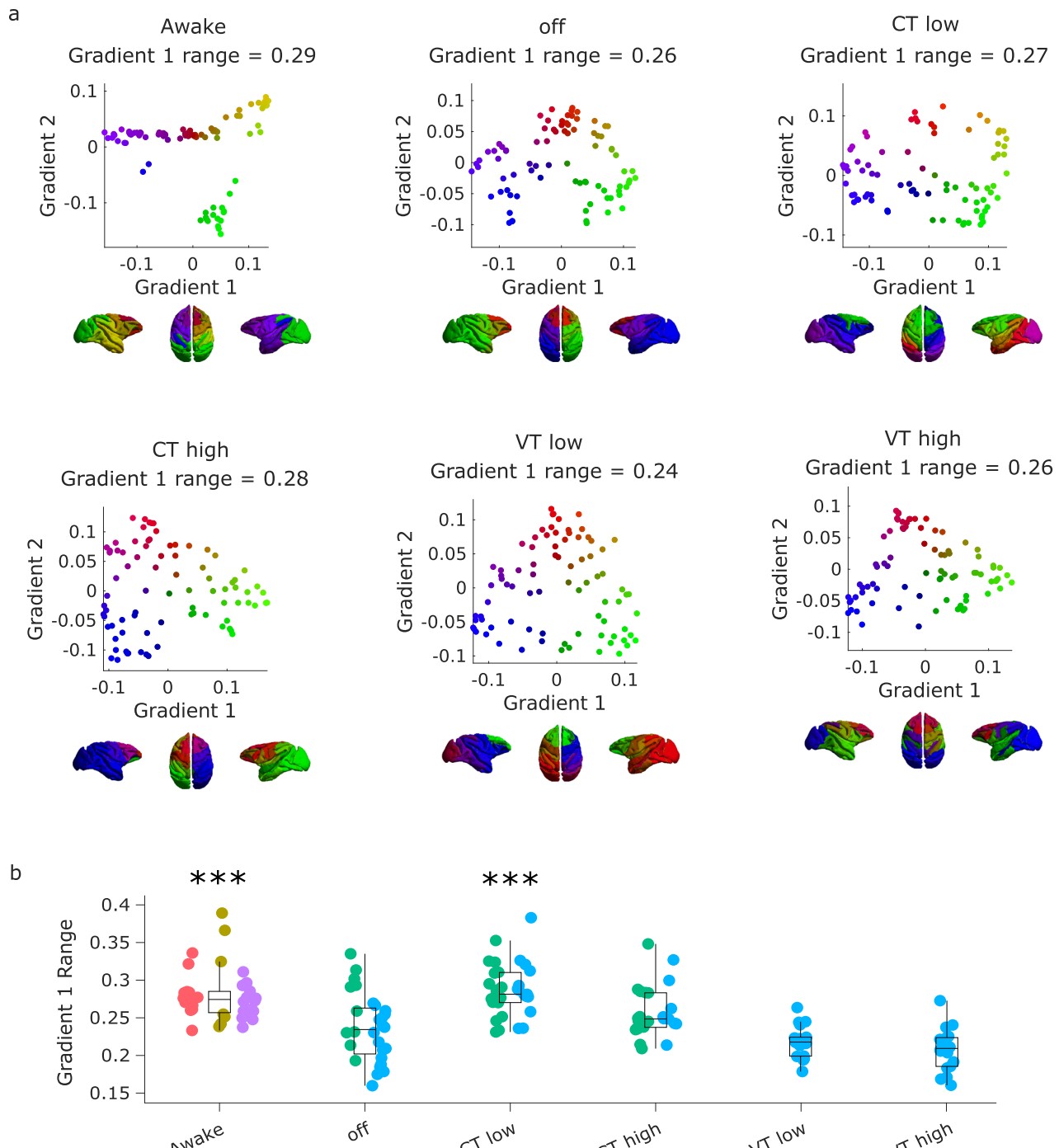

**Fig. 2 | Anaesthetic-induced collapse of the principal gradient of macaque functional connectivity is restored by thalamic stimulation. a** | Scatter plots showing the first two principal gradients of macaque functional connectivity (obtained from diffusion graph embedding) for the group-averaged FC matrix of the awake condition, and of the anaesthetized state, with and without DBS thalamic stimulation. The gradients are also plotted on the cortical surface of the macaques, with colour representing the position of each region along each gradient. **b** | Effect of thalamic deep-brain stimulation on the range of the principal gradient of functional connectivity in macaques during anaesthesia. Box plots: central line, median; box limits, upper and lower quartiles; whiskers, 1.5× interquartile range; dots of the same colour are provided by the same animal. *$p < 0.05$; **$p < 0.01$; ***$p < 0.001$ from linear mixed effects modelling (two-sided, FDR-corrected), compared against no stimulation ("off") condition during anaesthesia; see Supplementary Data 2 for full statistical reporting. $N = 36$ runs from 3 animals for Awake; 28 runs from 2 animals for anaesthesia (DBS-off); 31 runs from 2 animals for low amplitude centro-median thalamic DBS; 25 runs from 2 animals for high amplitude centro-median thalamic DBS; 18 runs from 1 animal for low amplitude ventro-lateral thalamic DBS; 18 runs from 1 animal for high amplitude ventro-lateral thalamic DBS.

We used standard parameters for the diffusion map embedding, as it is a widely used technique[5,15,18,57,58]. However, the algorithm has a number of parameters to set: the density of the FC matrix (default: 10% density); the anisotropic diffusion parameter α, which controls the influence of the density of sampling points on the manifold (α = 0, maximal influence; α = 1, no influence, with 0.5 being the default, representing a balance of local and global diffusion); and the similarity kernel used (default: normalised cosine angle). To demonstrate the robustness of our results, we show that they are replicated even when using very different α parameters: 0.1 (Fig. S7; Supplementary

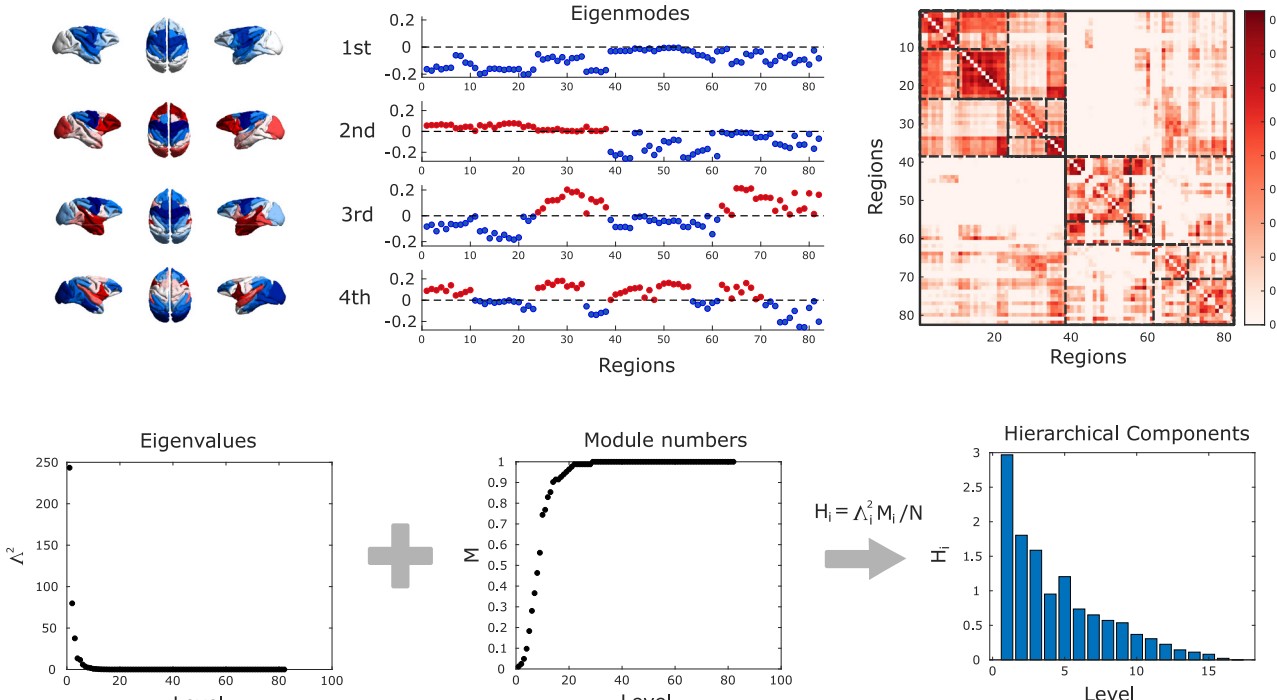

**Fig. 3 | Hierarchical integration quantified from brain functional eigenmodes.** Each eigenmode (except the first) divides cortical regions into two groups, at progressively finer scales. Combining the different groupings identifies a hierarchical sub-division of the functional connectome into nested modules. The relative weight of each eigenmode is given by its associated eigenvalue. The first eigenmode corresponds to integration, and segregation is then reflected by the contribution of the other eigenmodes.

Data 9 and 10) and 0.9 (Fig. S8; Supplementary Data 11 and 12). Results are even more pronounced when using higher density levels for the input FC matrix: 50% density (Fig. S9; Supplementary Data 13 and 14) and 90% density (Fig. S10; Supplementary Data 15 and 16). At both of these greater densities, the distinction between wakefulness and anaesthesia becomes visually more pronounced, and significant (FDR-corrected) differences can also be found for both light propofol and deep sevoflurane, in addition to deep propofol and ketamine (Fig. S9–S10). Crucially, even more striking results can be obtained if the similarity kernel is set to cosine angle similarity: all anaesthesia conditions induce significant reductions in the range of the principal gradient, compared with wakefulness (Fig. S11A and Supplementary Data 17), and both low and high amplitude CT stimulation produce significant increases in the gradient's range, which are not observed for stimulation of the VT control site (Fig. S11B and Supplementary Data 18). These validation analyses indicate that the effects of both anaesthesia and DBS on the principal functional gradient are consistent across methodological implementations, but they also show that changing from the default parameters can grant greater sensitivity: both to a broader range of anaesthetics, and to additional stimulation conditions. Likewise, we saw that considering additional gradients can provide additional insight, as also observed in humans by Huang and colleagues[15].

**Hierarchical integration across scales is compromised under anaesthesia and restored by thalamic DBS**

A second perspective on distributed brain function that can be obtained from studying the brain's eigenmodes pertains to the brain's hierarchical organisation, and how the latter supports the balance between integration and segregation across scales - which is a central feature of many prominent scientific theories of consciousness. Each eigenmode of functional connectivity identifies a distinct pattern of regions that are jointly activated (same sign) or alternate (opposite sign) (Fig. 3). Therefore, hierarchical modules can be identified based on the concordance or discordance of signs between regions across eigenmodes, progressively partitioning the FC into a larger number of modules and submodules, up to the level where each module coincides with a single region, indicative of completely segregated activity. The first eigenmode has the same sign throughout the entire cortex, reflecting whole-brain integration. At the next level, two partitions can be detected based on their different signs in the second eigenmode, and each in turn is subdivided at the following level of the hierarchy (i.e., from the third eigenmode) on the basis of regional signs. Thus, segregated modules at one level of the hierarchy can become integrated by being part of the same superordinate module (note that this hierarchical modularity based on eigenmodes is not equivalent to the clustering or modularity maximisation methods[61,62]). Hierarchical integration and segregation can then be quantified by the relative prevalence of the different eigenmodes, indicated by their associated eigenvalues[59].

We observe that the hierarchical brain integration of macaque eigenmodes is reshaped by loss and recovery of consciousness. Whereas hierarchical segregation behaved inconsistently (Fig. S12 and Supplementary Data 19, 20), we observed a significant effect of anaesthesia on eigenmode-based hierarchical integration ($F_{(5,113)} = 43.81$, $p = 6.38 \times 10^{-25}$ Fig. 4a and Supplementary Data 21). Likewise, we observed a significant effect of stimulation condition on hierarchical integration ($F_{(5, 150)} = 14.33$, $p = 1.8 \times 10^{-11}$). All anaesthetics reduced hierarchical integration, and this reduction was also replicated in the no-stimulation ("off") condition of the DBS dataset (Fig. 4b and Supplementary Data 22).

The effect of electrical stimulation delivered in the centro-median nucleus of the thalamus (at both low and high amplitude) is in the opposite direction as anaesthesia, significantly increasing eigenmode-based hierarchical integration (Fig. 4b). A significant but more modest increase was also observed for low-amplitude VT stimulation, although this stimulation did not have noticeable behavioural effects, suggesting that a minimum level of hierarchical integration may need to be

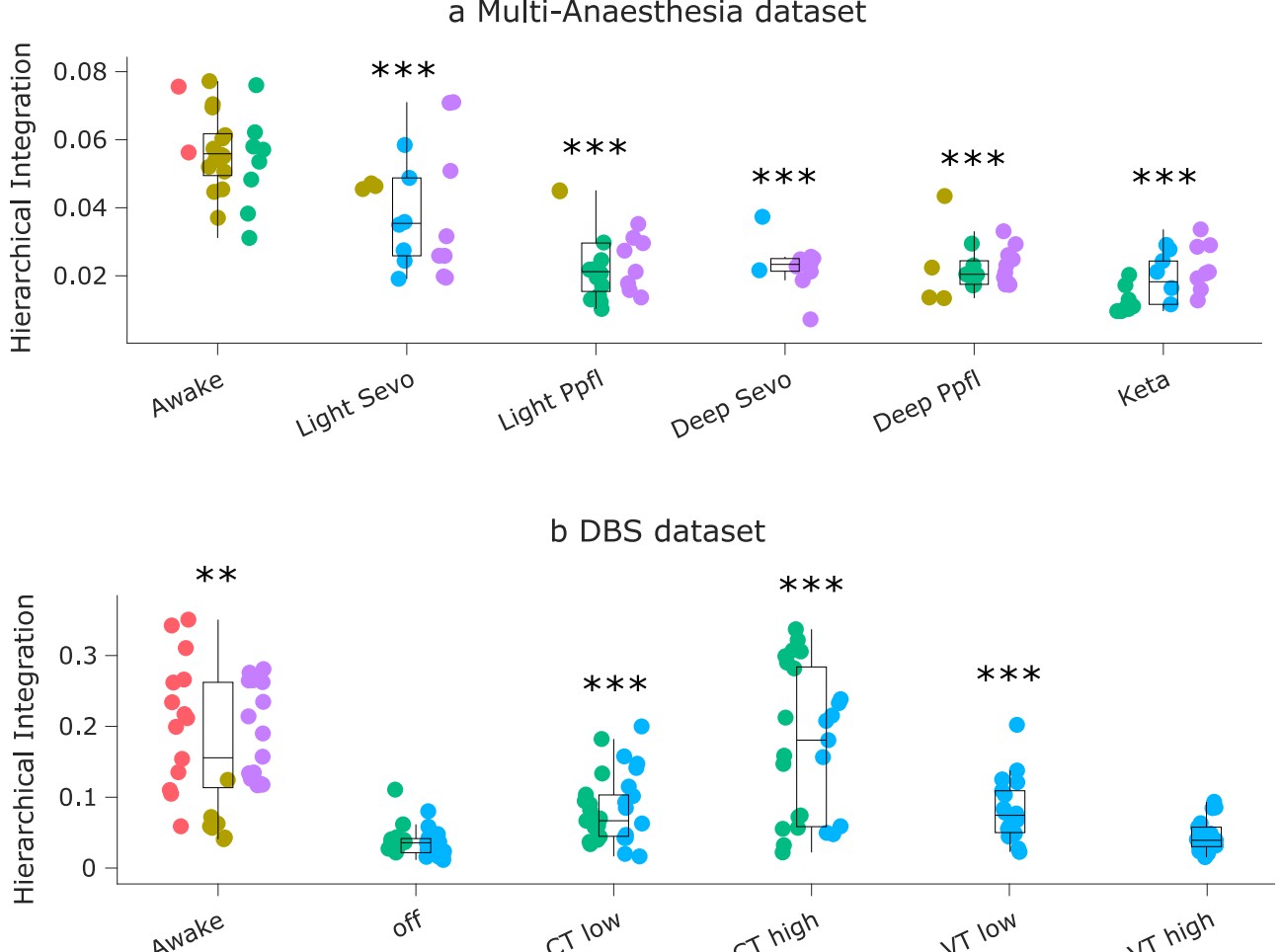

**Fig. 4 | Hierarchical brain integration from macaque eigenmodes is reshaped by loss and recovery of consciousness. a** | Eigenmode-based hierarchical integration is significantly reduced in the macaque brain under anaesthesia, whether induced by sevoflurane, propofol, or ketamine. ***$p < 0.001$ from linear mixed effects modelling (two-sided, FDR-corrected), compared against Awake condition. $N = 24$ runs from 3 animals for Awake; 18 runs from 3 animals for Light Sevoflurane; 21 runs from 3 animals for Light Propofol; 11 runs from 2 animals for Deep Sevoflurane; 23 runs from 3 animals for Deep Propofol; 22 runs from 3 animals for Ketamine anaesthesia. **b** | Eigenmode-based hierarchical integration is reduced by anaesthesia and increased by electrical stimulation of the centro-median thalamus with high current, as well as stimulation of the ventral-lateral thalamus. **$p < 0.01$; ***$p < 0.001$ from linear mixed effects modelling (two-sided, FDR-corrected), compared against no stimulation ("off") condition during anaesthesia. $N = 36$ runs from 3 animals for Awake; 28 runs from 2 animals for anaesthesia (DBS-off); 31 runs from 2 animals for low amplitude centro-median thalamic DBS; 25 runs from 2 animals for high amplitude centro-median thalamic DBS; 18 runs from 1 animal for low amplitude ventro-lateral thalamic DBS; 18 runs from 1 animal for high amplitude ventro-lateral thalamic DBS. Box plots: central line, median; box limits, upper and lower quartiles; whiskers, 1.5× interquartile range; within each panel, dots of the same colour are provided by the same animal. See Supplementary Data 21 and Supplementary Data 22 for full statistical results.

met, before this neural change is reflected at the behavioural level. The effect of high-amplitude CT stimulation was large enough to provide Bayesian evidence for the null hypothesis of no difference between Awake and high-amplitude CT stimulation (Fig. S13A), and further provided strong Bayesian evidence in favour of an effect greater than the effect of low-amplitude stimulation to either CT or VT (Fig. S13B, C). These results reveal signatures of consciousness restored by deep brain stimulation of the central thalamus under anaesthesia.

**Harmonic mode decomposition reveals increased structural constraints on brain activity under anaesthesia that are reversed by thalamic DBS**
Functional brain activity and connectivity unfold over the network of physical white matter pathways between brain regions: the structural connectome. Therefore, for our final investigation we go beyond function alone, by explicitly integrating brain activity and structural connectivity. To this end, inspired by the work of Atasoy and

colleagues[63] in humans, we leverage the mathematical framework of harmonic mode decomposition, which decomposes brain activity in terms of contributions from multi-scale patterns of coactivation using the network organisation of the structural connectome (Fig. 5)[63,64].

Each harmonic mode is a cortex-spanning activation pattern (eigenmode of the structural connectome) characterised by a specific granularity (spatial frequency, Fig. 6): large-scale, coarse-grained patterns (e.g., left and right or front and back of the brain) are strongly constrained by the organisation of the structural connectome, such that strongly interconnected regions are predicted to exhibit similar activation[25]. In contrast, high-frequency harmonic modes are relatively unconstrained by the underlying network structure, such that regions may exhibit different activity even if they are strongly connected in the structural network (Fig. 5). Therefore, decomposing functional MRI signals in terms of contributions from different harmonic modes of the connectome provides a quantification of the extent to which brain activity is constrained by the network structure of the connectome. In

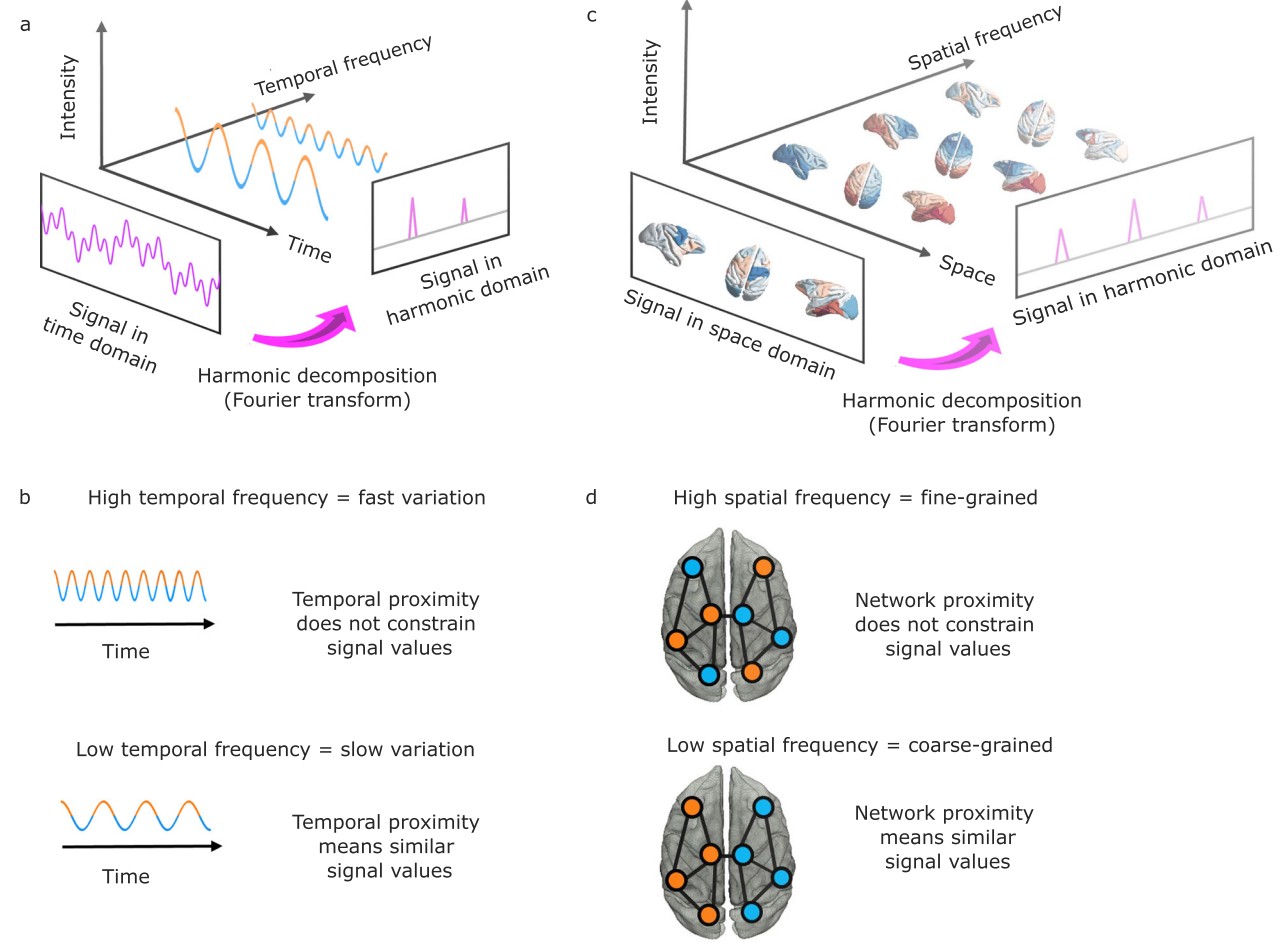

**Fig. 5 | Structural eigenmode decomposition generalises the Fourier transform to the network structure of the brain. a** | In traditional Fourier analysis, a signal in the time domain (represented in terms of successive time points) is decomposed into temporal harmonics of different frequencies and thereby rendered in terms of a different set of basis functions. **b** | High-frequency temporal harmonics correspond to rapidly varying signals, such that data points may have very different values even if they are close in time. In contrast, low-frequency temporal harmonics correspond to signals that change slowly over time, such that temporally contiguous data points have similar values, reflecting a greater time dependence of the signal. **c** | Harmonic decomposition of the connectome involves decomposing a signal in the spatial domain (represented in terms of fMRI activation at discrete spatial locations over the cortex) into harmonic modes of the structural connectome, resulting in a different set of basis functions in terms of whole-brain

distributed patterns of activity propagation distributed throughout the brain at different spatial scales (granularity), from large-scale patterns of smooth variation along geometrical axes (left–right and anterior–posterior being the most prominent) to increasingly fine-grained patterns. Note that here, frequency is not about time, but about spatial scale. **d** | Low-frequency (coarse-grained) connectome harmonics indicate that the spatial organisation of the functional signal closely matches the underlying organisation of the structural connectome: Nodes that are strongly connected exhibit similar functional signals (indicated by colour). High-frequency (fine-grained) patterns indicate divergence between the spatial organisation of the functional signal and the underlying network structure, where nodes may exhibit different functional signals even if they are closely connected in the structural network[25].

other words, harmonic mode decomposition is analogous to the well-known Fourier decomposition, but operating in the spatial domain[25].

The use of connectome-specific harmonic decomposition of cortical activity allows us to quantify the contribution of structural organisation to brain activity, across different spatial granularities. This approach therefore goes beyond previous investigations that simply assessed the similarity of structural and functional connectivity at a single scale[19,20,22]. Specifically, for each time-point, the magnitude of contribution of each harmonic to the BOLD activation across the brain (weighted by its associated eigenvalue) is termed the energy (see "Methods"). Through this formalism, we tested whether the different anaesthetics induced changes in the contribution of structural connectivity to functional activity, and whether such changes (if any) would be reversed upon restoration of responsiveness by thalamic DBS.

Our results indicate that the energy of harmonic modes in the macaque is reshaped by loss and recovery of

consciousness ($F(5,113) = 12.15$; $p = 1.96 \times 10^{-9}$). In particular, the energy of harmonic modes is significantly increased in the macaque brain under deep (but not light) anaesthesia, whether induced by sevoflurane, propofol, or ketamine (Fig. 7a and Supplementary Data 23).

Remarkably, this pattern is also significantly influenced by thalamic stimulation condition ($F(5,150) = 16.54$, $p = 5.39 \times 10^{-13}$). Specifically, the increase in harmonic energy is replicated in an independent dataset of propofol anaesthesia, and this increase is partly reversed under the effects of thalamic stimulation with both amplitudes and site dependence (Fig. 7b and Supplementary Data 24). High amplitude stimulation of the centro-median thalamic nucleus brought the overall harmonic energy significantly back closer to awake levels, with Bayesian evidence supporting the null hypothesis of no difference with Awake (Fig. S14A). Low-amplitude stimulation of both the CT and the VT control site induced a significant difference from anaesthesia without stimulation. However,

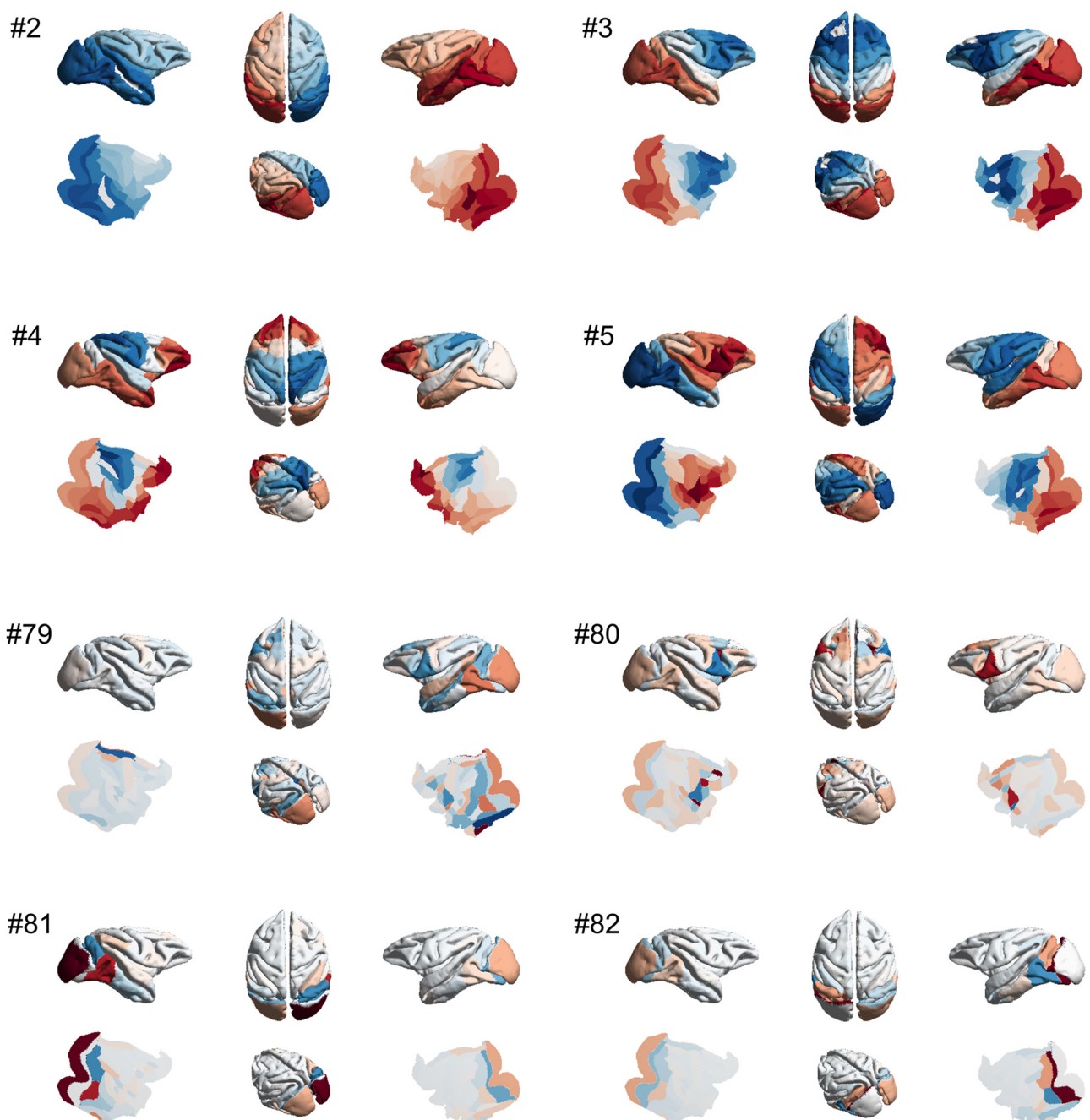

**Fig. 6 | Harmonic modes of the macaque structural connectome.** The first four non-uniform harmonic modes of the macaque structural connectome and the last four are shown on the surface of the macaque brain. Note that the first, low-frequency harmonics reveal large-scale patterns, while the last, high-frequency harmonics correspond to fine-grained patterns. The total number of harmonics, and therefore the maximum resolution, corresponds to the number of brain regions in the connectome (here 82). Please note that in the original formulation of Atasoy and colleagues, "connectome harmonics" are specifically defined as the harmonic modes of a high-resolution human structural connectome, obtained from combining long-range white matter tracts and local connectivity within the grey matter. Here we use instead the harmonic modes obtained from a parcellated macaque connectome. To avoid confusion, we refer to the eigenmodes obtained in this way as "harmonic modes".

the effects elicited by low-amplitude stimulation were smaller for both CT and VT, and comparison against wakefulness indicated either Bayesian support for the hypothesis of a difference against Awake condition, or no support in either direction (Fig. S14B, C) (Fig. 7b). In other words, although high-amplitude CT stimulation as well as both types of low-amplitude stimulation significantly increased hierarchical integration compared with no stimulation during anaesthesia, only high-amplitude CT stimulation restores hierarchical integration to the level of wakefulness.

## Multivariate brain-behaviour association across anaesthesia and DBS

Finally, we bring together our three complementary analyses to generate a multivariate characterisation of the neural effects of anaesthesia and its reversal with thalamic DBS. To this end, we use dominance analysis, which determines the relative contribution of each independent variable to the overall fit (adjusted $R^2$) of a multiple linear regression model[65,66], partitioning the total variance accounted for by each predictor. Here, we use as predictors the data pertaining to

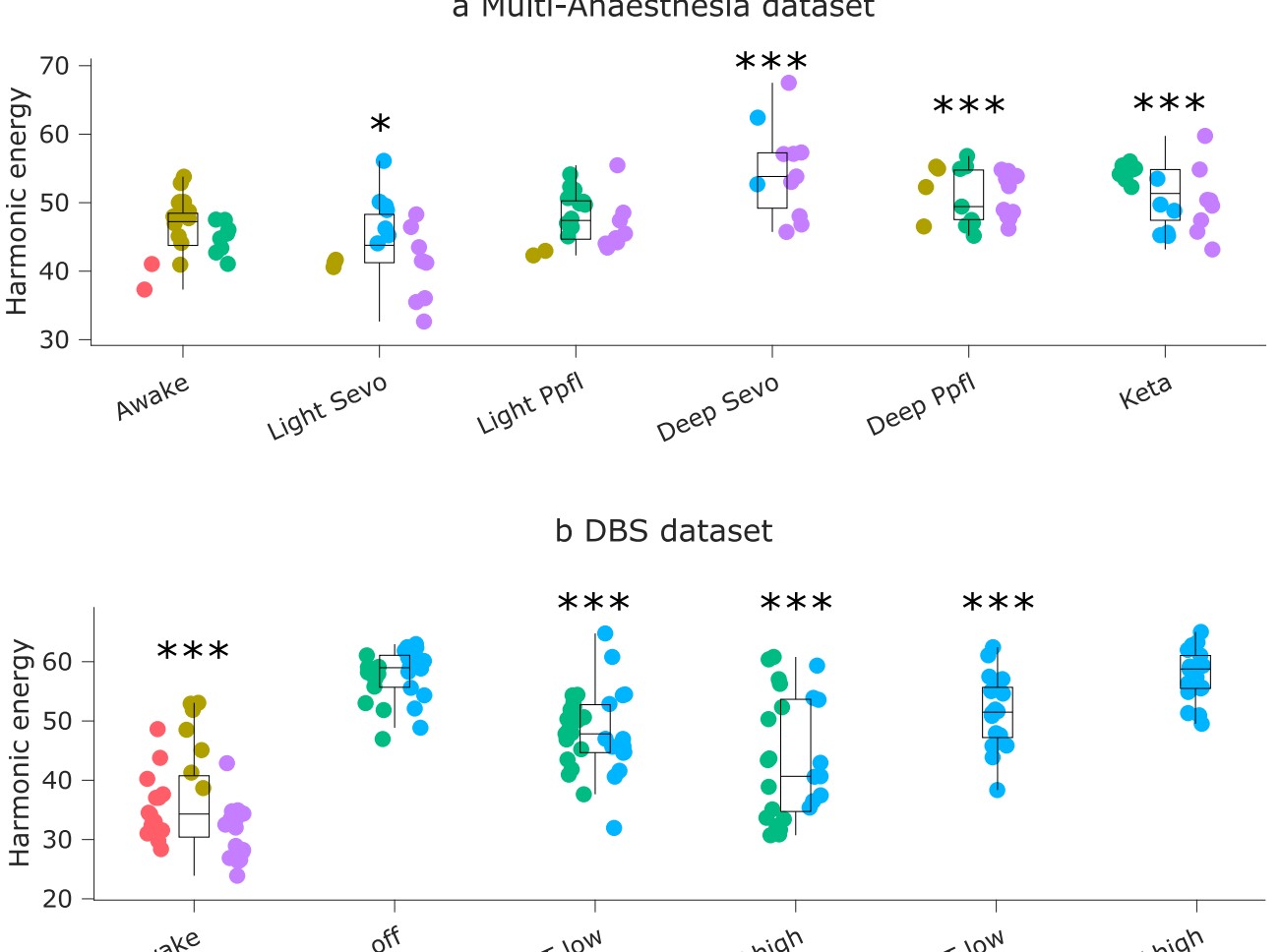

**Fig. 7 | Energy of harmonic modes in the macaque is reshaped by loss and recovery of responsiveness. a** | Energy of harmonic modes is significantly increased in the macaque brain under deep (but not light) anaesthesia, whether induced by sevoflurane, propofol, or the dissociative anaesthetic ketamine. *$p < 0.05$; **$p < 0.01$; ***$p < 0.001$ from linear mixed effects modelling (two-sided, FDR-corrected), compared against Awake condition. $N = 24$ runs from 3 animals for Awake; 18 runs from 3 animals for Light Sevoflurane; 21 runs from 3 animals for Light Propofol; 11 runs from 2 animals for Deep Sevoflurane; 23 runs from 3 animals for Deep Propofol; 22 runs from 3 animals for Ketamine anaesthesia. **b** | Effect of thalamic deep-brain stimulation on the energy of harmonic modes during

anaesthesia; ***$p < 0.001$ from linear mixed effects modelling (two-sided, FDR-corrected), compared against no-stimulation ("off") condition during anaesthesia. $N = 36$ runs from 3 animals for Awake; 28 runs from 2 animals for anaesthesia (DBS-off); 31 runs from 2 animals for low amplitude centro-median thalamic DBS; 25 runs from 2 animals for high amplitude centro-median thalamic DBS; 18 runs from 1 animal for low amplitude ventro-lateral thalamic DBS; 18 runs from 1 animal for high amplitude ventro-lateral thalamic DBS. Box plots: central line, median; box limits, upper and lower quartiles; whiskers, 1.5× interquartile range; within each panel, dots of the same colour are provided by the same animal. See Supplementary Data 23 and Supplementary Data 24 for full statistical results.

our three main neural markers of interest: range of the principal gradient, hierarchical integration, and harmonic energy. As target variable, we use the arousal score corresponding to each condition, on a scale from 0 to 11[67] (Fig. S2 and Supplementary Data 25). For this analysis, we combine data across the Multi-Anaesthesia and DBS datasets.

Our results reveal that the multiple linear regression model accounting for arousal score as a function of gradient range, hierarchical integration, and harmonic energy (Fig. 8a; see Fig. S15 for individual regression lines) has a total R² of 0.44, which is significantly greater than chance ($p < 0.001$, permutation-based; Fig. 8b). Among the three markers, over half (56%) of the total variance explained is accounted for by the hierarchical integration, with harmonic energy accounting for 30%, and gradient range for 14% (Fig. 8a). Similar results are obtained if instead of considering arousal scores as continuous, we dichotomise them with a cut-off of 9, thereby separating wakefulness and high-amplitude stimulation of the centro-median thalamus from all other conditions (anaesthesia with different drugs, and less

effective stimulations) (Fig. S16A). If instead we adopt a more lenient criterion, and set a arousal score cut-off of 3 so that low-amplitude CT stimulation and light anaesthesia are included along with high-amplitude CT and wakefulness (against deep anaesthesia and VT stimulation), then we see a different picture: the order of predictors' relative importance is preserved (hierarchical integration > harmonic energy > gradient range), but the values are nearly equal, with each neural marker accounting for almost exactly one-third of the total variance explained (Fig. S16B). Adding gradient dispersion and hierarchical segregation as predictors provided little difference: hierarchical integration and harmonic energy remained the main contributors; gradient dispersion exhibited greater contribution (15%) than gradient range (8%); and hierarchical segregation was negligible, with only 3% relative importance (Fig. S17; see Fig. S18 for individual regression lines).

In other words, our main three neural markers are complementary when it comes to distinguishing the complete absence of arousal from its even minimal restoration via CT stimulation. In contrast, when

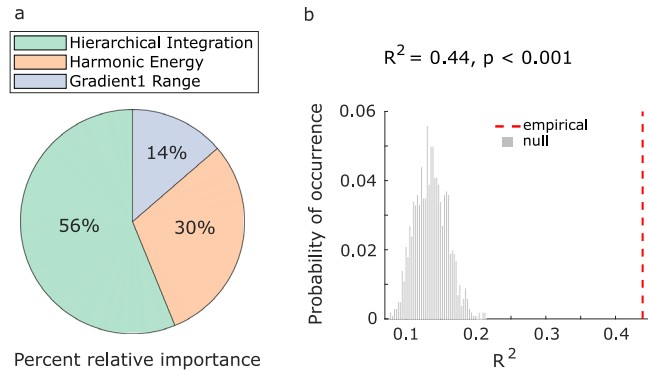

**Fig. 8 | Relating structural and functional eigenmodes of the brain to changes in arousal scores induced by anaesthesia and thalamic deep brain stimulation. a** | Dominance analysis compares all possible models obtained from distinct combinations of predictors, to distribute the variance explained between the predictors, in terms of percentage of relative importance (represented as pie chart). **b** | We establish the statistical significance of our model ($p < 0.001$) using a non-parametric permutation test (one-sided), by comparing the empirical variance explained ($R^2$) against a null distribution of $R^2$ obtained from repeating the multiple regression with randomly reassigned arousal scores.

requiring a more fine-grained characterisation that distinguishes high levels of arousal (which are achieved by high-amplitude but not low-amplitude CT stimulation) from low or no arousal, then the marker of hierarchical integration is the most informative. Overall, this multi-dimensional representation in terms of structural and functional eigenmode reorganisation identifies relevant axes along which perturbations of consciousness can manifest, linking brain organisation in different states and behaviour.

## Discussion

Here, we sought to combine the location-centric and distributed approaches to characterise the functional architecture of the macaque brain across perturbations of consciousness. We capitalised on two unique datasets of macaque resting-state functional MRI: one comparing wakefulness with anaesthesia induced by different agents (propofol, ketamine, sevoflurane)[23]; and another comparing wakefulness against propofol anaesthesia with versus without simultaneous intracranial electrical stimulation of different thalamic nuclei[21]. Leveraging the broad spatial coverage afforded by fMRI, we investigated anaesthetic-induced reorganisation of macaque brain functional architecture, its relationship to brain structure across brain scales, and the restoration of distributed function upon electrical stimulation of different thalamic sub-regions.

Our goals were twofold. First, we sought to determine whether distributed signatures of consciousness that have been recently identified in the human brain, can be generalised to the non-human primate brain, and whether they behave consistently across anaesthetics with distinct molecular mechanisms. Second, we aimed to determine whether the reorganisation of distributed brain function consistently induced by different anaesthetics can be reversed by targeted stimulation of different subregions of the thalamus, concomitant with the restoration of behavioural arousal.

Thanks to the broad spatial coverage provided by fMRI, we were able to characterise distributed brain function in terms of its relationship to the underlying structural organisation and in terms of how it supports different facets of hierarchical organisation. We showed that large-scale patterns are reliably altered during anaesthesia, regardless of the anaesthetic agent. Crucially, we also demonstrated that they are restored upon DBS-induced awakening, even in the presence of continued anaesthetic infusion. Taken together, these observations indicate that such changes are neither specific to a

particular anaesthetic agent, nor exclusively related to the mere presence of the anaesthetic in the bloodstream: rather, they are associated with behavioural markers of arousal, with the stimulation results being both location- and amplitude-dependent[21].

By studying a dataset of functional MRI recordings from macaque monkeys anaesthetised with propofol, sevoflurane, or ketamine (see the "Methods" section and[23] for details), we showed that the signatures that we identified generalise across anaesthetic agents. Unlike sevoflurane and propofol, ketamine does not act primarily as an agonist of GABA-A receptors, but rather as an antagonist of NMDA receptors. Despite the different molecular mechanisms of action, at anaesthetic doses such as the ones used here ketamine induces full loss of behavioural responsiveness, as well as suppression of complex cortical and thalamic responses to deviant stimuli in the local-global auditory paradigm[67]. However, there is evidence that even at anaesthetic doses, ketamine may induce loss of responsiveness not by suppressing consciousness (as propofol and sevoflurane are believed to do) but rather by inducing sensory disconnection from the environment, while potentially preserving some aspects of subjective experience[17,68]. The preservation of subjective experience despite sensory disconnection from the environment (and consequently, loss of behavioural responsiveness to the environment) is commonly experienced during dreaming - and indeed, ketamine-anaesthetised volunteers are known to report vivid hallucinations and dream-like experiences[69]. This phenomenon highlights the fact that behavioural unresponsiveness is an imperfect marker of unconsciousness, since it can also occur as a result of sensory disconnection, or motor impairment, neither of which is the same as unconsciousness[28,29]. Since it is not presently possible to obtain subjective reports from the animals involved in our study, a more conservative interpretation of our results is that they pertain to sensory disconnection from the environment, which is a common effect shared by ketamine, propofol, and sevoflurane. Although propofol (which was the drug used for the DBS experiments) and sevoflurane have also been reported to induce occasional dreaming in humans[70,71], it seems likely that they produce epochs of true unconsciousness, while it is unclear if ketamine does.

Therefore, in the remainder we interpret our results as pertaining to sensory disconnection from the environment (SDE).

Previous work has consistently shown increased correspondence between structural and functional connectivity under anaesthesia[19–22]. This measure was also restored by CT DBS in the original analysis of the present DBS dataset[21], and in humans it tracks the severity of disorders of consciousness[20]. However, existing investigations of structure-function correspondence during anaesthesia have typically relied on correlation and analogous distance metrics, which operate at a single spatial scale, whereas it is well established that brain structure and function both exhibit multi-scale, network organisation. This motivates the use of a multi-scale approach, provided by the framework of structural eigenmode decomposition, which simultaneously considers all available scales of organisation: from a single region to the entire cortex[25,63,72]. Our results showed that increased multi-scale coupling between brain activity and structural eigenmodes, as recently observed in the human brain during pharmacological and pathological loss of consciousness[25], can be generalised to the macaque brain - thereby supporting previous results that operated at a single scale (and which were obtained in terms of functional connectivity rather than activity)[19–22]. Note that this result is not in contrast with the ketamine-induced decreased structural coupling observed in humans by Luppi and colleagues[25], because that study employed a sub-anaesthetic dose of ketamine (i.e., having psychedelic-like effects), whereas the dose of ketamine used in the present study induced general anaesthesia. Thus, showing that previously identified signatures of unconsciousness in humans can be generalised to this additional pharmacological perturbation, and to an additional species, improves the robustness of these signatures, while also suggesting that ketamine can have

opposite effects on structure-function relationships in the brain, depending on its dosage and the corresponding changes in subjective experience. Biologically, it is possible that these different effects at different doses may be a result of ketamine's engagement of less-specific molecular targets at higher (i.e., anaesthetic) doses, beyond its principal action as NMDA antagonist.

In the human brain, considering highly fine-grained harmonic modes of the human connectome in the order of 18,000 surface vertices[25], harmonic mode decomposition has revealed that anaesthesia and disorders of consciousness induce a shift in the relative contribution of different harmonics, increasing the contribution of low-frequency (structurally-constrained) harmonics at the expense of high-frequency (liberal) ones. We note that the resolution of harmonic modes is fundamentally limited by the resolution of the underlying structural connectome: in the present case, 82 regions, each spanning several millimetres of cortex. Therefore, even the most fine-grained eigenmodes from the macaque 82-node connectome correspond to dividing the cortex in fewer than 100 patches - a number that lies firmly in the low-frequency (structurally-constrained) range for the human 18,000-eigenmode decomposition. In other words, when only a resolution of fewer than 100 harmonic eigenmodes is considered, the present results coincide with those reported by Luppi and colleagues[25]: unconsciousness manifests as an increased contribution of structural constraints to cortical functional activation. Future work using higher-resolution anatomical connectivity may enable us to obtain finer-grained insights about the respective roles of low-frequency versus high-frequency structural eigenmodes, such as by using the structural decoupling index of Preti and Van De Ville to quantify their balance[73].

Decomposition in terms of harmonic modes of the structural connectome characterises distributed brain function in terms of the relationship between functional activation and distributed patterns derived from brain anatomical network organisation, showing how this relationship changes upon sensory disconnection from the environment, and possibly loss of consciousness altogether. However, the harmonic modes themselves do not change, being based on structural connectivity, which is relatively stable at short timescales and unaffected by anaesthesia. Therefore, we complemented our investigation of distributed structure-function relationships with an additional investigation of distributed patterns arising from functional connectivity, termed functional gradients. This alternative perspective identified diminished hierarchical character of brain function under SDE: both in terms of the processing distance (contraction of the principal functional gradient, though with some variability depending on exact operationalisation), and in terms of diminished hierarchical integration across nested functional modules, when considering all functional eigenmodes, rather than just the principal one (or principal 2–3).

We also note that in our multivariate dominance analysis, both approaches that considered all eigenmodes (hierarchical integration, which considers all functional eigenmodes; and harmonic mode decomposition, which considers all structural eigenmodes) exhibited greater relative importance for predicting behavioural arousal, than consideration of the principal gradient (principal functional eigenmode, or dispersion of the first three) alone. Indeed, the measure of gradient range also exhibited a deviation from the behavioural pattern, being more influenced by low- than high-amplitude stimulation of the central thalamus. This indicates a non-linear relationship between CT stimulation and gradient range. As previously reported, low-amplitude stimulation of the central thalamus has weaker effect on behavioural arousal, than high-amplitude stimulation[21]. Specifically, it does not restore behavioural markers based on motion, whereas it restores eye-opening, corneal reflex, and minor visual exploration of the environment. In contrast, the structural eigenmode decomposition and hierarchical integration, which consider all eigenmodes (structural and functional, respectively), both displayed the largest effect for high-

amplitude stimulation of the central thalamus, coinciding with the fullest restoration of behavioural markers. Upon considering together the results of gradient analysis and hierarchical integration, it appears that the effects of low-amplitude CT stimulation primarily reflect on the principal functional gradient(s), whereas at high-amplitude the effect is less restricted to the principal gradients only, and instead exhibits a broader reach, concomitant with greater restoration of sensory connectedness to the environment. These observations reinforce the value of considering distributed brain function across multiple scales: the full effects of high-amplitude stimulation appear to be spread across multiple functional eigenmodes, such that only considering the first one provides an incomplete picture with weaker link to behaviour, whereas greater insight is obtained by considering all, and by combining both functional and structural eigenmodes.

The observation that hierarchical integration is particularly sensitive to anaesthesia and its reversal (Fig. 8), dovetails with both theoretical and empirical evidence. Numerous reports indicate that diverse markers of hierarchical brain organisation are impaired by perturbations of consciousness: across species, across modalities, and across ways of perturbing consciousness[15,16,74–76]. In particular, thanks to our multivariate analysis we were able to establish that the hierarchical character of integration (rather than hierarchical segregation, or hierarchical processing along the principal gradient alone) is of particular relevance for tracking the effects of anaesthesia and DBS on behaviour. This result is particularly relevant since prominent theories of consciousness converge in attributing a central role to integrative processes in the brain[11,12,77,78].

Crucially, restoration of such distributed cortical patterns can be triggered by selective stimulation of a specific subcortical region, as demonstrated by our spatially specific intervention: the centro-median thalamic nucleus - in contrast to the much weaker effects elicited by control site stimulation of the ventral lateral nucleus of the thalamus. This is of particular relevance because being compromised upon sensory disconnection from the environment is a necessary feature for a signature of consciousness, but it is not sufficient. A more stringent requirement is that the neural signature should also be restored when sensory connectedness is restored. Although Luppi et al.[25] did show that harmonic energy is restored upon post-anaesthetic recovery, they used spontaneous recovery after anaesthetic discontinuation, and therefore could not dissociate the presence of propofol in the bloodstream (and the various consciousness-unrelated effects that it may have on the brain) from the presence of environmental connectedness - only show that both co-varied with the harmonic energy. Our DBS results do achieve such a dissociation, both with harmonic energy and with other markers, by restoring both distributed brain function and sensory connectedness despite continuous anaesthetic presence. Although low-intensity VT stimulation also partly restored some markers of distributed function to a level comparable with low-intensity CT stimulation, they did not reach the level of high-intensity CT DBS, and had no effect on arousal - nor on processing of sensory stimuli and dynamic structure-function correspondence, as reported by Tasserie and colleagues' previous analysis of the same dataset. Thus, our unique combination of broad recording coverage (from fMRI) and spatially-specific intervention (from DBS) enables us to show that distributed functional patterns are reshaped by anaesthesia and under orchestration by a very selective locus, thereby showing how the location-focused and distributed approaches to the neural correlates of consciousness are not antithetical, but rather complement each other.

Our findings regarding thalamic influence on distributed cortical functions are consistent with recent views on the thalamus as a controller of brain dynamics[79,80], and more broadly on its role in supporting consciousness[38–40,81], including connectivity with cortical regions displaying consciousness-specific signatures[49] and the effects of selective thalamic stimulation on consciousness and arousal in animals[21,30–32,51,52,82] and human patients[36,37]. The central thalamus (CT)

consists of several intra and paralaminar thalamic nuclei that act as intermediaries between the brainstem/basal forebrain arousal systems and the cortex. These central thalamic neurons play a pivotal role in regulating arousal by establishing connections with extensive cortical networks[83]. Intralaminar thalamic (ILT) nuclei, also discriminated by 'matrix' cells (in opposition to 'core') serve dual functions, with orthodromic connections (exchanges from deep structures) and antidromic functions (strong calbindin protein staining, broadcasting signals to the supra-granular layer I of the cortex)[84–86]. ILT nuclei receive afferent connections from the mesencephalic reticular formation, which is part of the ascending reticular activating System[87,88]. They also receive inputs from the brainstem, superior colliculus, pedunculopontine tegmentum and basal ganglia[35,89] Efferent ILT pathways extend to the striatum (comprising the caudate nucleus and putamen)[90,91]. At the cortical level, CT projects to the medial pre-frontal, anterior cingulate and somatosensory cortex, as well as primary and supplementary motor areas, frontal eye field, associative regions and nucleus accumbens through well-established structural pathways[35,92].

In the current study, DBS electrodes targeted the centro-median thalamic nucleus or the the ventrolateral thalamus as a control site. However, as opposed to microstimulation (like used by Redinbaugh and colleagues), DBS modulation extended beyond CM neurons and broadcasted to a larger group of CT neurons. Thus, thalamic DBS achieves bicortical input via stimulation of corticothalamic axons (retrograde mechanism) and stimulation of thalamocortical axons (anterograde mechanism). This is supported by recent evidence that in another non-human primate (Marmoset), the central thalamus projects to higher-order frontal, cingulate, and posterior parietal regions[92]. Indeed, as per the results of Tasserie et al.[21], fMRI maps provided evidence for the strong cortical modulation effects of our DBS protocol. Specifically, although high VT-DBS could activate a fronto-parietal network, no modulation effect on the cingulate cortex activity was evident, whereas CT-DBS activated a fronto-parieto-cingular network, consistent with CT anatomical projection. Only CT-DBS elicited robust modulation of the striatum activity, probably due to axonal projections between CT and the striatum[92,93]. Restoration of the behavioural arousal and neural responsiveness to stimuli was observed only during high CT-DBS but never occurred in the other experimental conditions (low CT-DBS, low VT-DBS and high VT-DBS), suggesting that it may be key to stimulate among the nuclei included in the volume of activated tissue modelled for high CT-DBS[21]. Other groups previously applied optogenetic in rodent models of consciousness loss and could demonstrate a direct link between the modulation of CT neurons and the transition in the state of consciousness[80,94]. In particular, our findings are consistent with previous studies[30,31,33] since modelling the volume of activated tissue[21] indicated that our electrical stimulation encompassed the same targets as these groups, including intra-laminar central thalamic nuclei, as reported by Schiff[83].

Indeed, the original analysis of our DBS dataset indicated that high-amplitude CT stimulation also restores long-range functional correlations and processing of auditory stimuli beyond sensory cortex, as well as re-increasing the dynamic decoupling between structural and functional connectivity that had been suppressed by anaesthesia, and re-increasing the complexity of EEG signals[21]. Earlier studies had already reported that electrical stimulation of specific central-lateral thalamic nuclei (but not control sites) can restore arousal and putative electrophysiological markers of consciousness during anaesthetic-induced loss of responsiveness in macaques[21,30,31], counteracting the loss of high-frequency (gamma-band) activity and communication between the thalamus and deep cortical layers, which is induced by anaesthesia. However, these electrophysiological studies of thalamic stimulation during anaesthesia used electrodes for recording, given the authors' interest in different frequency bands of neural activity–

consequently limiting spatial coverage to 2–8 sites (although with layer-specificity not available to our present analysis). In contrast, the present work combined thalamic stimulation with coverage of the entire cortex, thanks to the use of functional MRI. This unique set-up enabled us to combine the spatial specificity of direct stimulation, while assessing its distributed effects on brain function.

Our evidence that selective thalamic DBS can restore not only arousal, but also distributed features of brain function that are disrupted across anaesthetics, lends further support for the translational potential of DBS as an avenue for treating the challenging condition of patients suffering from disorders of consciousness[36,95–98] - although with the clear caveat that such patients typically exhibit widespread cortical and subcortical damage, unlike the non-human primates in the present study. Of note, the evidence of anatomical connectivity between central thalamus and PFC[92] means that our results are also consistent with rodent evidence that PFC stimulation via carbachol administration awakens rats from sevoflurane anaesthesia[99], whereas tetrodotoxin-mediated inactivation of rat medial PFC had the opposite effect, delaying recovery[100]. Likewise, PFC inactivation also diminishes the ability of basal forebrain stimulation to promote arousal from sevoflurane anaesthesia[101]. Although such studies are yet to be translated to humans or non-human primates, they suggest the possibility that PFC stimulation may also represent an avenue of achieving similar effects as CT stimulation, both thanks to direct connections between the two, as well as via striatum-mediated disinhibition of the thalamus via descending arousal pathways[102].

Of particular relevance, a previous study in humans indicated that propofol-induced loss of behavioural responsiveness (LOBR) does not coincide with loss of brain responsiveness: noxious stimuli still elicited brain responses after volunteers had ceased overtly responding[41]. Loss of brain responsiveness (assessed with fMRI) occurred at higher propofol doses, and coincided with a plateau of EEG slow waves, termed "slow wave activity saturation" (SWAS)[41]. It is tempting to speculate that the period between LOBR and SWAS, characterised by isolation of the thalamocortical system from sensory stimuli, may correspond to the situation of sensory disconnection despite preserved subjective experience, prior to full unconsciousness at SWAS, and possibly analogous to the effect of ketamine anaesthesia. Given the central role of the thalamus in this phenomenon, it would be of great interest to determine whether SWAS is also observed in the macaque, whether it coincides with disruption of distributed brain function as assessed here, and whether it is countered by high- or low-amplitude CT stimulation. More broadly, it will be of great interest to assess whether CT DBS restores not only the prevalence of high-gamma oscillations and the complexity of spontaneous EEG signals (as already demonstrated), but also the complexity of EEG signals after perturbation with Transcranial Magnetic Stimulation pulses: this "Perturbational Complexity Index", inspired by Integrated Information Theory's emphasis on the need for both complexity and integration of brain signals in order to support consciousness,[78,103,104] represents one of the most accurate markers of consciousness currently available in humans, sensitive to both the severity of disorders of consciousness and the type and depth of anaesthesia[56,105–108]. This raises the question: does CT activation favour complex propagation of cortical perturbations? Of note, a recent mouse model indicated that under anaesthesia, thalamic stimulation (in the ventral posteromedial thalamic nucleus) elicited a more complex PCI response than stimulation of motor cortex[109]. Bringing together the diversity of existing markers of consciousness into a unified framework represents an ongoing challenge, and will be essential for translational applications to the clinic.

An important limitation of this work is that our distributed approach to brain function does not provide insights about the role of specific brain regions: instead, we focused on different whole-brain patterns. A substantial body of previous work has focused on specific regions, and our goal here was to provide an alternative perspective.

Additionally, thanks to the DBS dataset we were still able to identify the role of a specific thalamic sub-region for anaesthesia and its stimulation-induced reversal. Nevertheless, future analytic approaches capable of simultaneously identifying both region-specific and distributed signatures may hold additional promise for unveiling the rich interplays that support consciousness. Another limitation is that our analysis was exclusively cortical (except for the thalamic stimulation), yet other subcortical structures such as the basal ganglia and brainstem[43,87,110–113] are known to play an important role in anaesthetic-induced and pathological loss of responsiveness. Future work including subcortical regions may therefore provide additional insights. Likewise, although here we did not focus on the specific topography of functional gradients, but rather on their extent and relationships, we expect that future work will also benefit from such an approach.

Future directions can also combine methods from eigenmode analysis and high-order information-theoretic signals[114], which are a hallmark of many complex systems including the brain in its different states of consciousness. Additionally, here we relied on a symmetrised version of the macaque connectome, which is required for eigendecomposition in order to ensure real eigenmodes. However, since a directed macaque connectome is available, we expect that generalisation of harmonic mode analysis to account for directed connections, may provide a more refined picture. Pertaining to alternative avenues of eigenmode decomposition, recent work suggested that structural eigenmodes obtained from the geometry of the human brain may outperform those obtained from the human connectome, in terms of explaining variance in the functional signals[115]. Thus, future work may investigate whether cortical geometry provides insights into the reorganisation of consciousness, beyond those provided by the connectome.

In addition to the above-mentioned limitation of using behavioural unresponsiveness as a proxy for unconsciousness, a further potential limitation is that our analysis is based on fMRI acquired in resting state, i.e. in the absence of a stimulus or a task. Though beyond the scope of the present work, analysis of distributed patterns pertaining to the dysfunctional spread of naturalistic and synthetic stimuli, which was also observed in the same anaesthetised animals[21], will provide a more comprehensive understanding of how such patterns underpin information-processing and its perturbation by anaesthetics. Conversely, a strength of this work is the replication of our results not only across multiple anaesthetics, but also in the independent DBS-fMRI dataset. Direct comparisons between the neural data obtained in the two datasets are not straightforward: the Multi-Anaesthesia dataset used MION as a contrast agent to improve signal, whereas the DBS used the more commonly used BOLD signal (see Methods). Additionally, the acquisition parameters were different (e.g., 2.4 s vs 1.25 s TR), and the DBS dataset did not compare anaesthesia and wakefulness in the same animals, as did the Multi-Anaesthesia dataset. Despite these dissimilarities between the datasets, however, significant differences in the same direction were found for the deep propofol condition of the Multi-Anaesthesia dataset, and the propofol anaesthesia condition without stimulation ("off") of the DBS dataset, which is the only contrast that was present in both datasets. This replication demonstrates the robustness of our markers to acquisition differences, in addition to their generalisation across anaesthetics and across species. Likewise, the DBS data enabled us to assess the respective roles of both location (central versus ventral-lateral thalamus) and stimulation strength.

Overall, we showed that distributed patterns of functional activity and connectivity of the primate brain co-vary with sensory connectedness to the environment (and possibly consciousness), its suppression by different anaesthetics, and its restoration by subregion-specific thalamic stimulation. The resulting insights align both with the well-known increase in structure-function coupling observed across a variety of pharmacological and pathological states of unconsciousness across species, and also with prominent theories of consciousness that postulate a central role for integrative processes, which we show here to be reliably disrupted in the primate brain upon anaesthetic-induced sensory disconnection from the environment. Thus, the present work provides several advances. First, we generalise recent findings about the effects of anaesthesia on the human brain (connectome harmonics, gradient range) to non-human primates. Second, we show that these markers generalise across different anaesthetics, including with very different molecular mechanisms of action. Third, we show that these markers of distributed brain function are restored when sensory connectedness to the environment is restored by electrical stimulation of a specific thalamic sub-region, despite continuous anaesthesia, thereby dissociating them from the consciousness-unrelated effects of anaesthetics. Additionally, the DBS results show that although these markers pertain to the distributed functioning of the cortex, they can be influenced by the state of a specific subregion of the thalamus, thereby relating localised and distributed brain function. In this sense, our results help reconcile the traditional locationist approach to the neural correlates of consciousness with recent advances in understanding brain function in terms of distributed patterns. Thus, the neural markers that we consider here are not specific to humans, they are not specific to a particular drug, they track connectedness with the environment even in the presence of continuous anaesthetic administration, and they are under the influence of a specific thalamic sub-region. Altogether, the present work broadens our understanding of the link between brain network organisation and distributed function in supporting consciousness, and the interplay between local and distributed functional architecture.

## Methods
### Data
Both datasets used in the present study have been reported before[19,23]. Relevant details from these previous studies are provided below.

**Animals.** For the Multi-Anaesthesia dataset, five rhesus macaques were included for analyses (Macaca mulatta, one male, monkey J, and four females, monkeys A, K, Ki, and R, 5–8 kg, 8–12 yr of age), in a total of six different arousal conditions: awake state, ketamine, light propofol, deep propofol, light sevoflurane, and deep sevoflurane anaesthesia. Three monkeys were used for each condition: awake state (monkeys A, K, and J), ketamine (monkeys K, R and Ki), propofol (monkeys K, R, and J), sevoflurane (monkeys Ki, R, and J). Each monkey had fMRI resting-state acquisitions on different days and several monkeys were scanned in more than one experimental condition. Sex was not considered in this study. Because of the small sample sizes, the sex balance per group could not be secured. All procedures are in agreement with the European Convention for the Protection of Vertebrate Animals used for Experimental and Other Scientific Purposes (Directive 2010/63/EU) and the National Institutes of Health's Guide for the Care and Use of Laboratory Animals. Animal studies were approved by the institutional Ethical Committee (Commissariat à l'Energie atomique et aux Énergies alternatives; Fontenay aux Roses, France; protocols CETEA \#10-003 and 12-086). Additional details of the acquisitions can be found in the original publications[19,23].

For the DBS dataset, five male rhesus macaques (Macaca mulatta, 9 to 17 years and 7.5 to 9.1 kg) were included, three for the awake (non-DBS) experiments (monkeys B, J, and Y) and two for the DBS experiments (monkeys N and T). Only males were included in the DBS dataset in order to avoid the menstrual cycle and hormone variations. All procedures are in agreement with 2010/63/UE, 86-406, 12-086 and 16-040. For additional details, we refer the reader to the original publication[21].

**Anaesthesia protocol.** For the Multi-Anaesthesia dataset, monkeys received anesthesia either with ketamine, propofol, or sevoflurane[19,23], with two different levels of anesthesia for propofol and sevoflurane

anesthesia (light and deep). The anesthesia levels were defined according to the monkey sedation scale, based on spontaneous movements and the response to external stimuli (presentation, shaking or prodding, toe pinch), and corneal reflex[67]. For each scanning session, the clinical score was determined at the beginning and end of the scanning session, together with continuous electro-encephalography monitoring[67]. Monkeys were intubated and ventilated[19,23]. Heart rate, noninvasive blood pressure, oxygen saturation, respiratory rate, end-tidal carbon dioxide, and cutaneous temperature were monitored (Maglife, Schiller, France) and recorded online (Schiller).

During ketamine, deep propofol, and deep sevoflurane anesthesia, monkeys stopped responding to all stimuli, reaching a state of general anesthesia. For ketamine anesthesia, ketamine was injected intramuscular (20 mg/kg; Virbac, France) for induction of anesthesia, followed by a continuous intravenous infusion of ketamine (15 to 16 mg · kg−1 · h−1) to maintain anesthesia. Atropine (0.02 mg/kg intramuscularly; Aguettant, France) was injected 10 min before induction, to reduce salivary and bronchial secretions. For propofol anesthesia, monkeys were trained to be injected an intravenous propofol bolus (5 to 7.5 mg/kg; Fresenius Kabi, France), followed by a target-controlled infusion (Alaris PK Syringe pump, CareFusion, USA) of propofol (light propofol sedation, 3.7 to 4.0 μg/ml; deep propofol anesthesia, 5.6 to 7.2 μg/ml) based on the Paedfusor pharmacokinetic model (Absalom and Kenny 2005). During sevoflurane anesthesia, monkeys received first an intramuscular injection of ketamine (20 mg/kg; Virbac) for induction, followed by sevoflurane anesthesia (light sevoflurane, sevoflurane inspiratory/expiratory, 2.2/2.1 volume percent; deep sevoflurane, sevoflurane inspiratory/expiratory, 4.4/4.0 volume percent; Abbott, France). Only 80 min after the induction, the scanning sessions started for the sevoflurane acquisitions to get a washout of the initial ketamine injection (Schroeder et al. 2016). To avoid artefacts related to potential movements throughout magnetic resonance imaging acquisition, a muscle-blocking agent was coadministered (cisatracurium, 0.15 mg/kg bolus intravenously, followed by continuous intravenous infusion at a rate of 0.18 mg · kg−1 · h−1; GlaxoSmithKline, France) during the ketamine and light propofol sessions.

For the DBS dataset[21], anaesthesia was induced with an intramuscular injection of ketamine (10 mg/kg; Virbac, France) and dexmedetomidine (20 μg/kg; Ovion Pharma, USA) and then the same method as reported above for deep propofol sedation was used (Monkey T: TCI, 4.6 to 4.8 μg/ml; monkey N: TCI, 4.0 to 4.2 μg/ml). Awake scanning data were obtained from the remaining three animals, who did not provide anaesthesia data.

**Deep brain stimulation protocol.** The deep brain stimulation protocol is reported in detail in[21]. Briefly, N = 2 macaques (N and T) were implanted with a clinical DBS electrode (1.5-mm contact length, 0.5-mm spacing, and 1.27-mm diameter; Medtronic, Minneapolis, MN, USA, lead model 3389). DBS electrodes were implanted during stereotactic surgery, with the extracranial part of the DBS lead being accommodated with a custom-made 3D-printed MRI-compatible chamber. The rhesus macaque atlases of Paxinos[116] and Saleem and Logothetis[117] were used to guide surgery to target the right centromedian thalamus according to a neuronavigation system (BrainSight, Rogue, Canada), based on preoperative and intraoperative anatomical MRI [MPRAGE, T1-weighted, repetition time (TR) = 2200 ms, inversion time (TI) = 900 ms, 0.80-mm isotropic voxel size, and sagittal orientation][21]. Anatomical location of the DBS lead and contacts was ensured by in vivo MRI, as well as post-mortem in one of the two animals (for more details, see[21]). DBS-fMRI experiments were started no sooner than 20 days post-implantation. As previously reported, "For stimulation, the DBS electrode was plugged to an external stimulator (DS8000, World Precision Instrument, USA), and all the

parameters were tuned to a fixed value of frequency (f = 130.208 Hz, T = 7.68 ms), waveform (monopolar signal), and length of width pulse"[21]. Length of width pulse was set to w = 320 μs for monkey N, and w = 140 μs for monkey T.

We used two stimulation regimes: "low" DBS corresponded to 3 V voltage amplitude, and "high" DBS to 5 V. Each regime was applied during anaesthesia, to either VT or CT thalamic nuclei, with simultaneous rs-fMRI imaging. As reported by Tasserie et al.[21], high CT DBS had significant effects on general physiology, including an increase of mean heart rate ($P = 3.38 \times 10^{-26}$) and mean blood pressure ($P = 5.78 \times 10^{-17}$). Imaging was also acquired during wakefulness, and during anaesthesia without stimulation ("off" condition). The DBS stimulation was initiated just prior to the start of fMRI imaging, and terminated right after[21].

**Behavioural assessment of arousal.** We used a preclinical behavioural scale adapted from Uhrig et al.[67] to assess the arousal levels of the monkeys. This scale, based on the Human Observers Assessment of Alertness and Sedation Scale[118] and previously utilised in non-human primate (NHP) research[119], was used consistently across all experimental conditions, in both datasets. The arousal testing occurred outside the MRI environment and was conducted at the beginning and end of each scanning session, for each condition, once the animals were no longer under paralysis.

The assessment encompassed six criteria as follows:
- exploration of the surrounding world, from 0 to 2:
0 = total absence,
1 = small search of external clue,
2 = total investigation of the environment (such as head orientation to a sound);
- spontaneous movements, from 0 to 2:
0 = total absence,
1 = small torso and/or limb movement,
2 = large torso and/or limb movement
- shaking / prodding, from 0 to 2:
0 = total absence,
1 = small body movement,
2 = large body movement;
- toe pinch, from 0 to 2:
0 = total absence,
1 = small reflex (weak body movement or eye blinking or cardiac rate change),
2 = clear reaction (strong body movement and eye blinking or eye opening and cardiac rate change);
- eyes opening, from 0 to 2:
0 = total absence,
1 = small blinks or eye movements,
2 = full eye opening;
- corneal reflex, from 0 to 1:
0 = absent,
1 = present.
The behavioural score ranged from 0 to 11, where 11 represented the maximum note achievable and 0 the lowest.

In all cases, and for both datasets, we observed no differences in arousal scores between different animals in the same condition. For both datasets, the behavioural score during wakefulness was the maximum of 11/11 for all the animals (Monkey A, Monkey K, and Monkey J from the Multi-Anaesthesia dataset, and Monkeys B, J and Y from the DBS dataset): exploration of the surrounding world = 2; spontaneous movements = 2; shaking/prodding = 2; toe pinch = 2; eyes opening = 2; corneal reflex = 1.

For results pertaining to the different anaesthesia conditions of the Multi-Anaesthesia dataset, see Supplementary Data 25. As a summary, deep anaesthesia with ketamine, propofol, or sevoflurane induced an arousal score of 0, consistently in all animals. In contrast,

light anaesthesia induced an arousal score of 3 for sevoflurane, and 4 for propofol.

In the anesthesia without DBS ("off") condition, monkeys N and T displayed the minimum behavioral score of 0 over 11, same as the deep anaesthesia from the Multi-Anaesthesia dataset: exploration of the surrounding world = 0; spontaneous movements = 0; shaking/prodding = 0; toe pinch = 0; eyes opening = 0; corneal reflex = 0.

For anesthetized macaques under CT DBS at low amplitude (3 V), we measured a clinical score of 3 over 11 (exploration of the surrounding world = 0; spontaneous movements = 0; shaking/prodding = 0; toe pinch = 1, eyes opening = 1; corneal reflex = 1).

When the CT electrical stimulation amplitude was increased to 5 V (high-amplitude CT DBS), animals reached a total score of 9 over 11 (exploration of the surrounding world = 1; spontaneous movements = 1; shaking/prodding = 2; toe pinch = 2; eyes opening = 2; corneal reflex = 1).

For VT DBS, both low (3 V) and high (5 V) amplitude stimulation led to a clinical score of 0, identical to what is observed in the absence of any stimulation. See also Fig. S2.

**Functional magnetic resonance imaging data acquisition.** For both Multi-Anaesthesia and DBS datasets, for the awake condition, monkeys were implanted with a magnetic resonance compatible head post and trained to sit in the sphinx position in a primate chair[120]. For the awake scanning sessions, monkeys sat inside the dark magnetic resonance imaging scanner without any task and the eye position was monitored at 120 Hz (Iscan Inc., USA). The eye-tracking was performed to make sure that the monkeys were awake during the whole scanning session and not sleeping. The eye movements were not regressed out from rfMRI data. For the anesthesia sessions, animals were positioned in a sphinx position, mechanically ventilated, and their physiologic parameters were monitored. No eye-tracking was performed in anesthetic conditions.

For the Multi-Anesthesia dataset, before each scanning session, a contrast agent, monocrystalline iron oxide (MION) nanoparticle (Feraheme, AMAG Pharmaceuticals, USA; 10 mg/kg, intravenous), was injected into the monkey's saphenous vein[121]. Monkeys were scanned at rest on a 3-Tesla horizontal scanner (Siemens Tim Trio, Germany) with a single transmit-receive surface coil customised to monkeys. Each functional scan consisted of gradient-echo planar whole-brain images (repetition time = 2400 ms; echo time = 20 ms; 1.5-mm3 voxel size; 500 brain volumes per run).

For the DBS dataset, monkeys were scanned at rest on a 3-Tesla horizontal scanner (Siemens, Prisma Fit, Erlanger Germany) with a customised eight-channel phased- array surface coil (KU Leuven, Belgium). The parameters of the functional MRI sequences were: echo planar imaging (EPI), TR = 1250 ms, echo time (TE) = 14.20 ms, 1.25-mm isotropic voxel size and 500 brain volumes per run. Event-related data pertaining to auditory stimulation were also acquired and are reported in Tasserie et al.[21], but here we only used the resting-state fMRI data, and will not discuss the event-related data further. Scalp EEG data were also acquired using an MR-compatible system and custom-built caps (EasyCap, 13 channels), an MR amplifier (BrainAmp, Brain Products, Germany), and the Vision Recorder software (Brain Products). These results are reported in Tasserie et al.[21], but here we did not consider the EEG data and will not discuss them further,

**Data preprocessing and time series extraction.** For the Multi-Anaesthesia dataset, a total of 157 functional magnetic imaging runs were acquired[23]: Awake, 31 runs (monkey A, 4 runs; monkey J, 18 runs; monkey K, 9 runs), Ketamine, 25 runs (monkey K, 8 runs; monkey Ki, 7 runs; monkey R, 10 runs), Light Propofol, 25 runs (monkey J, 2 runs; monkey K, 11 runs; monkey R, 12 runs), Deep Propofol, 31 runs (monkey J, 9 runs; monkey K, 10 runs; monkey R, 12 runs), Light Sevoflurane, 25 runs (monkey J, 5 runs; monkey Ki, 10 runs; monkey R, 10 runs), Deep

Sevoflurane anaesthesia, 20 runs (monkey J, 2 runs; monkey Ki, 8 runs; monkey R, 10 runs). Additional details are available from the original publications[19,23,75].

Functional images were reoriented, realigned, and rigidly co-registered to the anatomical template of the monkey Montreal Neurologic Institute (Montreal, Canada) space with the use of Python programming language and FMRIB Software Library (FSL) software (United Kingdom, http://www.fmrib.ox.ac.uk/fsl/; accessed February 4, 2018)[23]. From the images, the global signal was regressed out to remove any confounding effect due to physiologic changes (e.g., respiratory or cardiac changes).

For the DBS dataset, a total of 199 Resting State functional MRI runs were acquired: Awake 47 runs (monkey B: 18 runs; monkey J: 13 runs; monkey Y: 16 runs), anaesthesia (DBS-off) 38 runs (monkey N: 16 runs; monkey T: 22 runs), low amplitude centro-median thalamic DBS 36 runs (monkey N: 18 runs; monkey T: 18 runs), low amplitude ventro-lateral thalamic DBS 20 runs (monkey T), high amplitude centro-median thalamic DBS 38 runs (monkey N: 17 runs; monkey T: 21 runs), and high amplitude ventro-lateral thalamic DBS 20 runs (monkey T: 20 runs)[21].

Images were preprocessed using Pypreclin (Python preclinical pipeline)[122]. Functional images were corrected for slice timing and B0 inhomogeneities, reoriented, realigned, resampled (1.0 mm isotropic), masked, coregistered to the MNI macaque brain template (Frey et al. 2011), and smoothed (3.0-mm Gaussian kernel). Anatomical images were corrected for B1 inhomogeneities, normalised to the anatomical MNI macaque brain template, and masked.

For both datasets, data were parcellated according to the Regional Map parcellation[123]. This parcellation comprises 82 cortical ROIs (41 per hemisphere. Voxel time series were filtered with low-pass (0.05-Hz cutoff) and high-pass (0.0025-Hz cutoff) filters and a zero-phase fast-Fourier notch filter (0.03 Hz) to remove an artifactual pure frequency present in all the data[19,23,75].

Furthermore, an extra quality control (QC) procedure was performed to ensure the quality of the data after time-series extraction[75]. This quality control procedure is based on trial-by-trial visual inspection by an expert neuroimager (C.M.S.), and it is the same as was previously implemented in Signorelli et al.[75]. Its adoption ensures that we employ consistent criteria across our two datasets, by adopting the more stringent of the two. We plotted the time series of each region, as well as the static functional connectivity matrix (FC), the dynamic connectivity (dFC) and a Fourier analysis to detect unconventional spikes of activity. For each dataset, visual inspection was first used to become familiar with the characteristics of the entire dataset: how the amplitude spectrum, timeseries, FC and dynamic FC look. Subsequently, each trial was inspected again with particular focus on two main types of potential artefacts. The first one may correspond to issues with the acquisition and is given by stereotyped sinusoidal oscillatory patterns without variation. The second one may correspond to a head or other movement not corrected properly by our preprocessing procedure. This last artefact can be sometimes recognized by bursts or peaks of activity. Sinusoidal activity generates artificially high functional correlation and peak of frequencies in the amplitude spectrum plot. Uncorrected movements generate peaks of activity with high functional correlation and sections of high functional correlations in the dynamical FC matrix. If we observed any of these anomalies we rejected the trial, opting to adopt a conservative policy. See Figs. S19–S21 for examples of artifact-free and rejected trials.

As a result, for the Multi-Anaesthesia data set a total of 119 runs are analysed in subsequent sections (the same as used in Signorelli et al.[75]): awake state 24 runs, ketamine anaesthesia 22 runs, light propofol anaesthesia 21 runs, deep propofol anaesthesia 23 runs, light sevoflurane anaesthesia 18 runs, deep sevoflurane anaesthesia 11 runs. For the DBS data set, a total of 156 runs are analysed in subsequent sections: awake state 36 runs, Off condition (propofol anaesthesia without

stimulation) 28 runs, low-amplitude CT stimulation 31 runs, low-amplitude VT stimulation 18 runs, high-amplitude CT stimulation 25 runs, high-amplitude VT stimulation 18 runs.

**Anatomical parcellation and structural connectivity.** Anatomical (structural) connectivity data were derived from the recent macaque connectome of[124], which combines diffusion MRI tractography with axonal tract-tracing studies, representing the most complete representation of the macaque connectome available to date. Structural (i.e., anatomical) connectivity data are expressed as a matrix in which the 82 cortical regions of interest are displayed in x-axis and y-axis. Each cell of the matrix represents the strength of the anatomical connection between any pair of cortical areas.

## Analysis

**Structure-function coupling via harmonic mode decomposition.** Functional spatiotemporal patterns of neural activity derived from fMRI can be decomposed in terms of anatomically-based distributed building blocks: eigenvectors of the graph Laplacian of the structural connectome[13,63,64]. Here we are inspired by the original Connectome Harmonic Decomposition formulated by of Atasoy and colleagues[63], who used the harmonic modes of a high-resolution human structural connectome, obtained from combining long-range white matter tracts and local connectivity within the grey matter. Since here we use instead the harmonic modes obtained from a parcellated macaque connectome of diffusion MRI and tract-tracing, we refer to the eigenmodes obtained in this way as "harmonic modes", and reserve the term "connectome harmonics" for those developed by Atasoy and colleagues.

Following the method developed by Atasoy and colleagues[63], we compute the symmetric graph Laplacian $\Delta_G$ on the matrix $C$ that represents the macaque structural connectome. To ensure symmetry of the corresponding connectivity matrix, and real eigenvalues, we averaged entries above and below the diagonal to obtain an undirected connectome. Thereafter, we estimate the connectome Laplacian (discrete counterpart of the Laplace operator $\Delta$ applied to the network of the macaque structural brain connectivity):

$$\Delta_G = D^{-1/2} L D^{-1/2} \tag{1}$$

with $L = D - C$, where $D$ is the diagonal degree matrix of the graph $C$ i.e.

$$D(i,i) = \sum_{j=1}^{82} C(i,j) \tag{2}$$

We then calculate the harmonic modes $\varphi_k$ $k \in \{1, \ldots, 82\}$ by solving the following eigenvalue equation:

$$\Delta_G \varphi_k = \lambda_k \varphi_k \tag{3}$$

$\forall k \in \{1, \ldots, 82\}$, with $0 < \lambda_1 < \lambda_2 < \ldots < \lambda_n$ where $\lambda_k, k \in \{1, \ldots, 82\}$ is the corresponding eigenvalue of the eigenvector $\varphi_k$. In other words, $\lambda_k$ and $\varphi_k$ are the eigenvalues and eigenvectors of the Laplacian of the primate structural connectivity matrix (macaque connectome). Therefore, if $\varphi_k$ is the harmonic pattern of the $k^{th}$ spatial frequency, then the corresponding eigenvalue $\lambda_k$ is a term relating to the intrinsic energy of that particular harmonic mode. With an increasing harmonic number $k$, we obtain more complex and fine-grained spatial patterns (see Figs. 5, 6).

**Decomposition of fMRI data.** At each timepoint $t \in \{1, \ldots, 500\}$, (corresponding to one TR), the spatial pattern of cortical activity over brain regions at time $t$, denoted as $F_t$, was decomposed as a linear combination of the set of harmonic modes $\{\varphi_k\}_{k=1}^{82}$:

$$F_t = \omega_1 \varphi_1 + \omega_2 \varphi_2 + \ldots + \omega_{82} \varphi_{82} = \sum_{k=1}^{82} \omega_k(t) \varphi_k \tag{4}$$

with the contribution $\omega_k(t)$ of each harmonic mode $\varphi_k$ at time $t$ being estimated as the projection (dot product) of the fMRI data $F_t$ onto $\varphi_k$:

$$\omega_k(t) = \langle F_t, \varphi_k \rangle. \tag{5}$$

**Energy of harmonic modes.** Once the fMRI cortical activation pattern at time $t$ has been decomposed into a linear combination of harmonic modes, the magnitude of each harmonic's contribution to the cortical activity of each harmonic $\varphi_k$, $k \in \{1, \ldots, n\}$ (regardless of sign) at any given timepoint $t$, denoted $P(\varphi_k, t)$, is called its power, for analogy with the Fourier transform, and it is computed as the amplitude of its contribution:

$$P(\varphi_k, t) = |\omega_k(t)|. \tag{6}$$

The normalized frequency-specific contribution of each harmonic $\varphi_k$, $k \in \{1, \ldots, n\}$ at timepoint $t$, termed "energy", is estimated by combining the magnitude strength of activation (power) of a particular harmonic mode with its own intrinsic energy given by the associated eigenvalue $\lambda_k^2$:

$$E(\varphi_k, t) = |\omega_k(t)|^2 \lambda_k^2. \tag{7}$$

**Functional gradient mapping from diffusion map embedding.** Macaque cortical functional gradients were calculated using the BrainSpace toolbox https://github.com/MICA-MNI/BrainSpace[57] as implemented in MATLAB, with default parameters for density, similarity kernel, and anisotropic diffusion parameter α (see below).

We calculate the functional connectivity matrix (FC) as the Pearson correlation between each pair of regional fMRI signals per scan per condition.

Following previous work, each matrix was z-transformed and thresholded row-wise to achieve 10% density (i.e., retaining 10% of entries), retaining only the strongest connections in each row[58]. The normalised cosine similarity was calculated on the thresholded z-matrix to generate a similarity matrix reflecting the similarity in whole-brain connectivity patterns between vertices. While the FC matrix reflects how similar each pair of regions are in terms of their temporal co-fluctuations, this similarity matrix reflects how similar two regions are in terms of their patterns of FC. The similarity matrix is required as input to the diffusion map embedding algorithm that we have used here in agreement with previous work on functional gradients and how they are reshaped by pharmacological interventions[58].

Diffusion Map Embedding is a nonlinear manifold learning technique[5,57] that exploits the properties of the graph Laplacian to model the diffusion process, and is therefore related to harmonic mode decomposition - though performed on functional data rather than using a common structural connectome, to reveal its contributions to the functional activations (indeed, there is a deep mathematical analogy exists between diffusion graph embedding on the functional connectome and the recent extension of CHD called functional harmonics[72,125]).

The relative influence of density of sampling points on the manifold is controlled by an additional parameter (α) in the range of 0 to 1, which for diffusion map embedding is set to 0.5 to provide a balance between local and global contributions to the embedding space estimation[126].

The high-dimensional similarity matrix is treated as a graph, with connections (entries of the similarity matrix) reflecting the similarity

between the regional patterns of FC. The technique estimates a low-dimensional set of embedding components (gradients); in this low-dimensional space, proximity reflects similarity of the patterns of FC: regions with similar FC patterns (which are strongly connected in the network) are placed close to each other, and regions with low similarity are placed far apart. In this way, each gradient represents one dimension of covariance in the inter-regional similarity between FC patterns, with a small number of gradients capturing most of the dimensions of inter-regional similarity, which can then be visualised by a low-dimensional scatter plot[5,57,126]. In the embedding space, each gradient can be understood to be anchored at regions that have the strongest values for that gradient, suggesting that this particular embedding dimension captures their similarity profiles of FC well. In contrast, regions that are close to the origin (i.e. have a low absolute value for a particular gradient) mean that they are only minimally similar to the anchor points of that gradient, which overall does not strongly capture their FC similarity profile well overall[5,57,126].

Therefore, the more different the extremes of a gradient differ along the axis of the gradient, the more the differentiation between regions is being captured by that gradient. To quantify this formally, we calculated the difference between the maximum and minimum values of each scan along the first gradient[18,58] (which mathematically captures most of the variation in FC profiles within each scan) and compared these differences across conditions. Note that the dimension of greatest variability (first gradient) is assessed in a data-driven manner and need not be identical across different scans. The range of the second gradient is computed in the same way. Additionally, we consider the dispersion of the first three gradients, calculated as per Huang et al.[15] as the sum of squared Euclidean distance of all regions to the global centroid in the 3D cortical gradient space. Finally, we also consider an additional measure, the ratio of the eigenvalue associated with the first gradient, to the sum of all gradients' eigenvalues, reflecting its relative importance.

To demonstrate the robustness of our results, we show that they are replicated even when the α parameter is set to very different values (0.1 or 0.9); when using higher density levels for the input FC matrix (50% density or 90% density); and when using cosine similarity instead of normalised angle similarity.

**Hierarchical integration and segregation.** A third perspective on distributed brain function that can be obtained from studying the brain's eigenmodes emerges from its hierarchical organisation, and how the latter supports the balance between integration and segregation across scales - which is a central feature of many prominent scientific theories of consciousness. Classical graph-theoretic measures such as small-worldness, clustering, and modularity quantify integration and segregation at a single scale and are not suitable for capturing these properties across multiple hierarchical modules[61,62]. However, a recently introduced formalism based on eigenmodes of functional connectivity can provide such quantification[59].

The functional connectivity (FC) is a symmetric matrix, which can be decomposed as $FC = U\Lambda U^T$ where $U$ is an orthogonal matrix whose columns are eigenvectors (eigenmodes) of FC, and $\Lambda$ is a diagonal matrix whose entries are the eigenvalues of FC. Each eigenmode of functional connectivity identifies a distinct pattern of regions that are jointly activated (same sign) or alternate (opposite sign). Therefore, hierarchical modules can be identified based on the concordance or discordance of signs between regions across eigenmodes, progressively partitioning the FC into a larger number of modules and submodules: from the first eigenmode with all regions exhibiting the same sign, up to the level where each module coincides with a single region, indicative of completely segregated activity. Thus, segregated modules at one level of the hierarchy can become integrated by being part of the same superordinate module. During this nested partitioning process, we obtain the module number $M_i$, $(i = 1...N)$ and the modular size $M_j$, $(j = 1...M_i)$ at each level.

Each level $i$ of the hierarchy is characterised by two quantities: the number of modules $M_i$ into which the cortex is divided, and the covariance explained by the corresponding eigenmode (given by its squared eigenvalue $\Lambda_i^2$). However, the number of modules alone may not properly describe the picture of nested segregation and integration because the size of modules may be heterogeneous. The correction factor was calculated as $p_i = \sum_j |m_j - N/M_i|/N$ which reflects the deviation from the optimised modular size in the $i^{th}$ level. Thus, the correction effect is stronger for a larger deviation of modular size from homogeneity[59].

Since the first eigenmode encompasses the entire cortex into one whole-brain module, the corresponding eigenvalue $\Lambda_i^2$ quantifies the overall contribution of whole-brain integration,

$$H_{In} = \Lambda_i^2 M_1(1 - p_1)/N^2. \tag{8}$$

The overall level of segregation across the hierarchy is quantified as

$$H_{Se} = \sum_{i=2}^{N} \Lambda_i^2 M_i (1 - p_i)/N^2, \tag{9}$$

the sum of contributions from all eigenmodes except the first (i.e., all eigenmodes that involve a partitioning of the cortex), each weighted by the corresponding number of modules $M_i$ (further corrected for heterogeneous modular sizes, since a partition into two modules of size 1 and N-1 is clearly not as segregated as a partition into two equally-sized modules).

**Dominance analysis**
Dominance analysis is a method of assessing predictor importance in the presence of multiple predictors, while accounting for their interactions[65,66]. To assess the relative contribution of each predictor, the same multiple linear regression model is fit using every combination of predictors. This amounts to running $2^p - 1$ possible combinations, with $p$ being the number of predictors. For each predictor, it is then possible to obtain the average of the relative increase in overall model fit (adjusted $R^2$) in the presence versus absence of that predictor, across all $2^p - 1$ combinations. This is termed "total dominance". The total adjusted $R^2$ of the complete model is given by the sum of the dominance of all predictors. The total variance that each predictor explains, is termed its percentage of relative importance[65,66].

Here, we use as regressors the data pertaining to our three markers of interest: range of the principal gradient, hierarchical integration, and harmonic energy. For this analysis, we combine data across the Multi-Anaesthesia and DBS datasets. As target variable, we use the arousal score corresponding to each condition, on a scale from 0 to 11[67]. To ensure that data could be combined across the DBS and Multi-Anaesthesia datasets, values of each marker were normalised between 0 and 1 separately within each dataset (by subtracting the minimum value observed within the dataset, and then dividing by the maximum) before aggregating the two datasets.

In addition to using dominance analysis for multiple regression, it is also possible to use it for multivariate classification. To do so, we dichotomised the arousal scores: either using a cut-off of 9, to combine together wakefulness and the effects of high-amplitude CT stimulation, against everything else (corresponding to the difference between high arousal and low or no arousal); or using a cut-off value of 3 instead, thereby separating deep anaesthesia and VT stimulation, from wakefulness, CT stimulation, and light anaesthesia (i.e., no arousal versus any level of arousal).

## Statistical reporting

Overall significance was assessed with a one-way mixed effects analysis of variance (ANOVA, implemented using MATLAB's *fitlme* function), with condition as a fixed effect, and animal identity as random effect. Subsequently, we performed individual pairwise contrasts (two-sided) using linear mixed effects modelling (also implemented using MATLAB's *fitlme* function), with condition as the fixed effect and animal identity as the random effect. This approach enabled us to take into account the fact that the same animal could provide more than one data-point to each condition, as well as contributing data-points for more than one condition.

For the Multi-Anaesthesia dataset, each anaesthesia condition was compared against wakefulness, to test the hypothesis that our neural markers are affected by anaesthesia. For the DBS dataset, we compared the DBS-off (no stimulation) anaesthesia condition, against all other conditions: wakefulness, and all stimulation types. This allowed us to replicate the effects of anaesthesia on our neural markers of interest (by comparing Awake against no-stimulation anaesthesia), and to test the hypothesis that CT stimulation should counter the effects of anaesthesia on our neural markers. In both datasets, correction for multiple comparisons was carried out using the False Discovery Rate correction[127]. We report all statistical tests and descriptive statistics in Supplementary Data S1 to S24.

To assess statistical significance of the dominance analysis multiple regression, we performed a one-sided non-parametric permutation test: we built a null distribution of adjusted $R^2$ values by repeating the multiple regression 1000 times, but with permuted assignment of the arousal scores, and evaluating how often the null adjusted $R^2$ was greater than the empirical adjusted $R^2$.

A drawback of traditional significance testing is that it cannot formally accept the null hypothesis, and a failed rejection of the alternative hypothesis can occur because the null hypothesis is true, but also because statistical power is insufficient. In addition to standard significance testing, we therefore also adopted the recently developed R package for Bayes Factor Functions[128] to quantify Bayesian evidence in support of the alternative hypothesis versus the null hypothesis, based on our test statistics. This approach makes it possible to distinguish whether the evidence actively supports the alternative hypothesis, or it actively supports the null hypothesis, or it is insufficient to discriminate between them - as a function of the expected effect size[128]. In our Supplementary Data reporting statistical results, we report for each contrast the corresponding Bayes Factor across four ranges of standardised effect sizes: negligible (0 to 0.15); small (0.15 to 0.35); medium (0.35 to 0.65) and large (0.65 to 1). In each range, the summary BF10 was obtained as the geometric mean of the Bayes Factor Function values returned for the effect sizes in the range, in steps of 0.01 (geometric mean was used because Bayes Factors are interpreted as ratios).

### Reporting summary

Further information on research design is available in the Nature Portfolio Reporting Summary linked to this article.

## Data availability

Source data are provided with this manuscript. For the Multi-anaesthesia dataset, raw data are available for access from author B.J. through academic collaboration. For the DBS dataset, raw data are available for access from author B.J. through academic collaboration. Source data are provided with this paper.

## Code availability

The Python processing for PreClinical data pipeline, Pypreclin version 1.0.1, is freely available at https://github.com/neurospin/pypreclin. FMRIB Software Library (FSL) is freely available online (http://www.fmrib.ox.ac.uk/fsl/; version accessed February 4, 2018). Gradients were computed using the freely available BrainSpace toolbox (http://github.com/MICA-MNI/BrainSpace) for MATLAB (version 2019a), and further MATLAB code for gradient dispersion calculation is freely available from Huang et al. (2023) at https://doi.org/10.5281/zenodo.6955280. Third-party MATLAB code to quantify hierarchical integration and segregation from[59] is freely available at https://github.com/TobousRong/Hierarchical-module-analysis. Third-party Python software (version 3.8 was used) for Dominance Analysis is freely available at https://github.com/dominance-analysis/dominance-analysis. Third-party code for Bayes Factor Functions (package: BFF) in R (version 4.3.1) is freely available at https://cran.r-project.org/web/packages/BFF/index.html.

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

## Acknowledgements

This work was supported by a Gates Cambridge Scholarship (OPP 1144) (to A.I.L.); a Travel Grant of the Boehringer Ingelheim Fonds (to A.I.L.); and the visitor program of the European Institute for Theoretical Neuroscience (to A.I.L.); the Fondation pour la Recherche Médicale (FRM grant number ECO20160736100, to J.T.); FNRS Belgium, project MIS/VA - F.4512.21 (to C.M.S.) and grant Embodied-Time – 40011405 (to C.M.S.); the Stephen Erskine Fellowship of Queens' College, Cambridge (to E.A.S.); CNRS and the European Community (Human Brain Project, H2020-945539, to A.D. and R.C.); the Fondation Bettencourt Schueller (to B.J.); Fondation de France (to B.J.), Human Brain Project (Corticity project FLAG-ERA JTC2017, to B.J.); Institut National de la Santé et de la Recherche Médicale (to B.J.), UVSQ (to B.J.), Commissariat à l'Energie Atomique (to B.J.), Collège de France (to B.J.).

## Author contributions

A.I.L., R.C., A.D., conceived the analysis; J.T., L.U., and B.J. designed the experiments; J.T., L.U., and B.J. collected the data; A.I.L. and C.M.S. analysed the data; A.I.L., C.M.S., and R.C. wrote the manuscript with feedback from all co-authors. B.J., A.D., E.A.S., supervised the project. All authors approved the manuscript.

## Competing interests

The authors declare no competing interests.
