## [Peer Review File · Nature Communications]

Local Orchestration of Distributed Functional Patterns Supporting Loss and Restoration of Consciousness in the Primate BrainREVIEWER COMMENTS

Reviewer #1 (Remarks to the Author):

Local Orchestration of Global Functional Patterns Supporting Loss and Restoration of Consciousness in the Primate Brain

Andrea I. Luppi, Lynn Uhrig, Jordy Tasserie, Camilo M. Signorelli, Emmanuel Stamatakis, Alain Destexhe, Bechir Jarraya, Rodrigo Cofre

Summary

This paper aims to identify ‘signatures of consciousness’ across different forms of anesthesia, and test whether this can be reversed using deep-brain stimulation (DBS) of the thalamus. The results reveal summary measures of large-scale brain patterns that are altered under anesthesia and recovered upon DBS-induced awakening.

General

A strength of this paper is the combination of DBS and pharmacology. However, I have questions about the novel insights and the claims about causality. Please find my detailed comments below, approximately in order of importance.

Major

1. Although I appreciate the value of DBS, to me the results still remain relatively correlational in nature because the summary measures of the large-scale pattern are not explicitly targeted with DBS. Therefore, although it is shown that the disappearance/appearance coincides with changes in consciousness, the brain pattern does not necessarily cause consciousness. As such, I would suggest that the authors reduce the strength of the causal interpretations throughout the paper.
2. The ‘signatures of consciousness’ in this paper involve complex methodology yet end up in overly simplistic single-number summary values (gradient 1 range, hierarchical integration, harmonic energy). It is unclear to me how mechanistically meaningful these

results are. What insights does this really offer in terms of the biological basis of consciousness?

3. How consistent were the actual observed whole-brain patterns across animals and types of anesthesia? Given the complex methodology and reduction to simple summary metrics, more extensive validations are needed throughout this work.

4. The methods state that: "Trials were kept when the row signal did not present signs of artifactual activity, functional connectivity was coherent with the average and dynamic connectivity presented consistent patterns across time". These criteria are overly vague and it would be useful if the authors could specify explicit thresholds to define 'coherent' and 'consistent' used for trial selection.

5. Why were matrices thresholded row-wise to retain only the strongest connections? This approach risks losing a substantial amount of useful information.

6. Why was the difference between the maximum and minimum values used for the gradient range, rather than the variance explained? Wouldn't the variance explained offer a more robust index of the gradient strength?

Minor

7. The figures used to describe anesthesia versus deep-brain stimulation are strange and confusing. I would strongly encourage the authors to label images using words instead of figures.

8. Triple shared first authorship and triple shared last authorship seems a little bit excessive, leading to 6 major contributing authors. This is a question for the Editorial Team, but perhaps there should be some limits on shared roles?

9. Figure 2 compares statistically against the CT high condition. Why is this condition especially picked out for comparison?

Reviewer #2 (Remarks to the Author):

The authors present compelling research making use of two valuable and unique fMRI datasets from macaque monkeys used for previous publications. One dataset involves whole-brain imaging of macaques under the influence of different general anesthetics. The other dataset makes use of a now established technique to reverse anesthesia using central thalamic deep brain stimulation first demonstrated by Redinbaugh & colleges in 2020, followed by Bastos & colleges in 2021, and last by the authors of this manuscript in 2022. The authors demonstrate that several measures relating to structural and functional connectivity are consistently altered under general anesthesia and selectively reversed by the central thalamic stimulation, not control stimulation of the VT. Ultimately, they show that anesthesia disrupts signatures of functional integration, which are reversed by stimulations that rouse the monkeys from anesthesia. These results are well-aligned with recent papers using similar methods and support existing theories of consciousness and suggest that conscious experience is enabled by the integration of complex information across different brain regions.

The most notable result of the paper relates to the three-fold finding on global changes of structural and functional connectivity. In general, the authors report that general anesthesia increases the harmonic energy, reflecting the interdependence of functional connectivity on underlying anatomical structure, and reduces the gradient range and hierarchical integration, reflecting the processing capacity and the network integration respectively. These effects are reversed by central thalamic stimulation, which is associated with increased behavioral responsiveness under general anesthesia. These results are well-aligned with growing evidence that consciousness depends on key network interactions enabling the complex flow of information, which are consistently perturbed under anesthesia and during other less-conscious states.

This work is likely to be of significance to the field for many reasons. First, as the authors point out, these findings take advantage of “whole-brain” data afforded by fMRI and are well-aligned with different theoretical conceptualizations of consciousness. However, this significance is masked in the current version of the manuscript due to overuse of jargon and limited effort made to contextualize the findings relative to other metrics. Non-expert readers will struggle to conceptualize these analyses and interpret them into functional mechanisms of the brain that might contribute to consciousness. This is exacerbated by the

fact that the authors do not adequately contextualize their research relative to other studies and proposed metrics of consciousness, despite citing a number of studies that have used them with similar state-based comparisons.

The methods of the paper appear sound and state of the art, taking strong advantage of an interesting experimental design as the benefits of fMRI data. However, while the reported results are visually compelling, the current manuscript obscures much of the statistical information, making it difficult to assess the validity of various claims. This lack of transparency will foster distrust amongst readers and greatly diminishes the significance of the work. Overall, I would not recommend publication without substantial major revisions of the paper to correct three major flaws.

1) Readability: the current manuscript uses too much jargon and does not do a good job of applying terms in a clear, consistent manner throughout the manuscript.

2) Statistical transparency: the current manuscript does not accurately describe all experimental results and makes it hard for readers to find the necessary statistical information to back up their claims.

3) Contextual Significance: the current manuscript does not fully flesh out the relationship between the authors' results and the growing literature not only on consciousness, but central thalamic DBS in macaques, burying the lead.

Introduction:

1) The introduction is somewhat confusing and makes it hard to track the goal of the paper. The authors introduce many concepts with increasingly opaque jargon that are not fleshed out and connected the neural mechanisms until deep in the results section. Readers are likely familiar with concepts such as structural and functional connectivity, but may struggle with terms like "eigenmode decomposition" without additional context and help from the authors. This could be improved by offering clearer definitions of terms, as well as citations drawing links between the proposed measures and the plethora of terms and concepts included in the introduction.

2) The goals of the paper as listed in the introduction (test the local vs distributed approach) do not quite match the goals as listed in the discussion (to show selective reinstatement of effects in the DBS condition). In the introduction, the methods are treated like a means to

an end, while they are treated like the end goal in the discussion. This should be fixed and made more consistent.

3) Some terms vital to the reader's understanding of the paper go undefined across the manuscript. This is especially noticeable with the term "consciousness". As the authors are no-doubt aware, consciousness is a term with many working definitions across the field of neuroscience, but most popularly, as a multivariate construct combining elements of animal arousal and awareness. Consciousness can be somewhat separated from the idea of conscious states, where consciousness can be assumed to be higher, lower, or possibly not present (a claim that requires substantial evidence and analysis). Unfortunately, the authors make no effort currently to provide their working definition of consciousness and flexibly switch across the paper between terms. The DBS effects are described as increasing conscious arousal, or sometimes behavioral responsiveness, while anesthesia is referred to as producing a "loss of consciousness" and is described as a consciousness state. Overall, the terminology is imprecise and mixed, greatly hindering both readability and the fidelity of their conclusions later in the paper.

4) Because the authors are introducing so many terms and not providing clear definitions in the introduction, it is easy to get confused or misunderstand the authors' logic, especially because some of the same terms are used to reference different things throughout the paper. For example, the authors refer both to a local vs distributed approach to the study of consciousness, meaning to differentiate between studies that seek to assess the relevance of a given brain area (local) from studies that seek to address global metrics of neural connectivity and function. They also then use the term localized to refer to smaller-scale network interactions from global interactions. Thus, in a given sentence, local may refer to either a specific brain region, or the scale of network connectivity, which is confusing to readers and muddies interpretation of the author's words. The authors also use the term "integration" to describe functional connectivity, while also using it to reference their approach of combining different scales of analysis with fMRI. Readability would be greatly improved if the authors took more care to do the following:

- a. use unique terms to reference unique concepts
- b. define terms when they are first used conceptually, and then operationalized as different

computational metrics

c. Strictly adhere to terms and refrain from using them out of context.

5) The way the authors suggest that one innovative component of the current study is its use of whole-brain imaging to provide a global approach. While I agree that this is a strength of the current study, I disagree with the framing of the debate, as it suggests that studies either test the contributions of individual areas, or operate at global network scales, and there is no overlap between the two. Indeed, there is a sizable middle ground inhabited by many studies that test the contributions of individual areas as well as complex information transfer in specific long-range projection pathways or through complex networks. It is unclear where the authors delineate between what counts as a local vs global study, or why they seem to suggest that electrophysiology studies mostly reside in the local category.

6) The authors introduce the overall structure of their manuscript by comparing metrics across different anesthetics, and then following DBS of the central thalamus to restore consciousness under anesthesia with the following sentence: "To this end, we leverage functional MRI (fMRI) data from non-human primates in the awake state, under loss of consciousness induced by three different anaesthetics (sevoflurane, propofol, ketamine) and restoration of consciousness by deep brain stimulation (DBS)." The authors should provide the relevant references for this DBS effect (Redinbaugh 2020, Bastos 2021, & Tasserri 2022) and clarify to readers that this is an established well-documented technique in macaque monkeys.

7) The authors make some bold claims about the goal of the paper and the interpretation of the findings in the introduction: "Comparing the brain-wide effects of different anaesthetics enables us to disentangle which aspects of the brain's functional organisation support consciousness, being consistently targeted by anaesthetics, despite their distinct molecular mechanisms." This claim does not seem born out by their own conclusions and discussion. The current manuscript does little to disentangle aspects of functional organization that play a causal role towards consciousness, likely because the experimental design does not test these causal factors. I welcome the authors to provide mechanistic arguments in their conclusion, but in the current version of the manuscript, they are not present and not

warranted by the statistical analysis or design.

8) The authors make a serious logical error in their introduction related to the interpretation of their results by claiming they demonstrate a causal link between their structural measures and consciousness. The authors claim: "Crucially, we also aim to obtain more stringent evidence for the causal relevance of distributed signature of consciousness, by determining whether the reorganisation of distributed brain function consistently induced by different anaesthetics can be reversed by targeted stimulation of different subregions of the thalamus, a brain structure that has been repeatedly associated with supporting consciousness." The authors do causally influence consciousness with both their anesthetic and DBS manipulations. These manipulations are associated with changes in functional organization, and presumably, consciousness. However, it is not appropriate to assume a causal link between the functional reorganization and consciousness without substantial analysis not provided in the current manuscript. While anesthesia is expected to reduce consciousness, it is unclear when loss of consciousness happens in this study in relation to the specific reductions noted in the measures of functional organization. The same is true for the DBS experiments. It is perfectly possible that the structural reorganization is an unnecessary correlate of consciousness, or fully insufficient to support consciousness. All the authors can truly claim in this paper is that it is a correlate of conscious states. Thus, it would be more appropriate for the authors to state that their causal manipulations allow them to reveal specific neural correlates of consciousness that hold across anesthetics and DBS conditions. Any arguments about why this effect might be causal should be reserved as a hypothesis for the discussion, and well substantiated by existing literature or theory. A later sentence offered by the authors accurately portrays the power of their design: "This dual causal manipulation - pharmacology and electrical stimulation - provides us with a unique opportunity to study functional changes that are observed during loss of consciousness and reappear upon recovery of consciousness, despite continuous anaesthetic infusion." As the authors seem aware, their design does not afford the logical or computational power to make causal claims, but does help demonstrate that the effects are not specific to the stringent chemical effects of the anesthetic used in their DBS experiments.

Results/Methods:

1) The authors provide considerable useful context in this section explaining the methods they use and constructs they seek to characterize. For example, they clearly describe how the eigenmode gradient depth relates to the capacity of neural computation and processing. Some of this information should be provided in the introduction to prime the readers with context and provide citations about the links between these constructs and consciousness. In general, I would recommend using more space in the introduction to describe measures of neural computation and their theoretical relationship to consciousness, and space in the results to flesh out the operationalization used in this paper.

2) In the introduction and methods, the authors make it clear that this paper is based on datasets which have already been used in other publications. Many of the sections in the methods reference readers back to the original publications for specific details. While this is fine in principle, the authors must do more to clarify the link between the datasets and any differences that might exist. Are the data sets in question identical (all using the exact same animals, trials, stimulation events and anesthetic sessions with no changes)? If so, that should be clearly stated to alleviate any questions from readers. If not, the authors should clarify which portions of the dataset are shared between the source papers and why data might have been omitted. This is critical for readers to interpret the results between both papers. For example, some data from the Tasserie paper included auditory stimulation in a local-global paradigm. Is this data included in the current manuscript? This seems unlikely as the authors claim their analyses did not include task data, which is present in Tasserie 2022.

3) The authors should provide more context from the Tasserie 2022 paper in the results and methods section to make it clear to readers how they measured changes in consciousness induced by DBS and validated that their different DBS conditions link back to different states. Presently, readers are simply expected to take the authors at their word that higher amplitude central thalamic stimulation was the only condition that consistently increased consciousness under anesthesia. While it is likely that readers will also engage with the Tasserie paper, it is currently unclear if the datasets are factually identical in terms of treatment and analysis. Are all stimulations used? Were any omitted? Were any central thalamic stimulations ever ineffective at the higher voltage to restore consciousness? Is

there a cutoff the authors used? Were any ventral lateral stimulations ever mildly effective? At the very least, the authors should reiterate for the reader's benefit basic evidence (quantified if possible) so they can more accurately compare results between these conditions.

4) The statistical reporting has been greatly improved by the provided supplemental document and it must be fully incorporated into the manuscript. There should be a dedicated statistical section in the methods providing additional details. Specifically, the authors should make it clear how they performed the ANOVA analyses as well as the pairwise T-tests, including which software, if any, they used. Points in the method section making specific claims about the relationship of different groups should have the relevant statistical results provided in the text, and the rest should be found in the provided tables in the supplement. The authors should also clearly state in the figure captions, where they provide p-values, the name of the relevant test that the p-values reflect. They should similarly clarify the name of the effect size measure. Full transparency is increasingly necessary and demanded by readers. The authors should ensure that readers have easy access to all statistical information to back their claims.

5) The authors report to have done pair-wise comparisons of different metrics between different states. They also report that some animals participated in more than one experimental condition. This source of non-independence should be controlled for where it occurs, preferably with multivariate analysis or within-subjects, paired designs. It is unclear if the authors have done so.

6) Figures 1 & 2 demonstrate that the depth of processing, as measured by the principal gradient of the first eigenmode of functional connectivity, is reduced by anesthesia and increased with the thalamic DBS. This same logical structure is shared in Figure 4, 5, & 7 and depends on the demonstration both of positive (significant) effects for some conditions, and insignificant effects for others. While this is a fine logic in principle, the authors have committed statistical errors in their current phrasing of and interpretation of their null results. For example, the authors state about Fig 1&2: "The range of the principal gradient of macaque functional connectivity is significantly reduced by anaesthesia, regardless of the

specific agent used (Figure 1 and Figure S1), and is significantly increased back to awake levels by low amplitude stimulation of the centro-median thalamus (Figure 2 and Figure S1).” This claim about a “significant increase back to awake levels” is presently supported by the insignificant differences between wake and the low amplitude central stimulation condition demonstrated in the provided Supplement Table 3. It would be more accurate for the authors to report that the results are statistically indistinct from the wake state unless they are willing to add measures that can back the null result. This logical argument is repeated for the results in figures 4, 5, and 7 and should be amended, removed, or supported by more appropriate statistics (see next point).

7) In reference to the point above, comparing the pattern of significant and insignificant effects is further fraught because efforts to avoid type 1 error, which is necessary to interpret results with multiple comparisons, can increase the probability of type 2 error. False discovery rate corrections make it easier to find an insignificant effect, as it influences the alpha of the test. Thus, if a significant effect is present, but underpowered, it is more likely to be deemed insignificant. If the authors wish to make strong claims about the insignificant effects, they should include power analyses for their tests to verify they are sufficiently powered to find effects when they exist. A better solution, however, would be to rely on Bayesian statistics, like the bayes factor, which lend credence to insignificant findings. Most specifically, these should be applied to the comparisons between the wake state and the effective thalamic stimulation conditions to improve their interpretability.

8) In figure 8, the authors present the effect sizes (cohen’s D) for different tests as a multivariate summary of their results for both the multi-anaesthesia and DBS data sets. While this figure provides a helpful summary of the findings, it does not provide any additional compelling analysis. The current presentation seems to argue for synergistic interpretation of the three metrics, suggesting they capture different elements of the differences between the states. This cannot be verified when comparing the effects from different models. It would be more compelling to see multivariate decoding analyses or multivariate regression results comparing the effect sizes of the three metrics within model on the ability to discriminate conscious states. This will allow the authors to comment on the separability of the metrics, and which metrics contribute more to conscious state

discrimination. They can more clearly argue if any of the metrics seem complimentary, or if any are statistically redundant.

9) Again for figure 8, the effect sizes for some measures are quite different between the multi-anesthesia and DBS datasets, despite the maximum levels representing the distances between the wake state and general anesthesia. How do the authors explain the substantial differences between the datasets? Further, the current axes make it hard to compare between Panels A and B.

Discussion:

1) Appropriately, the authors bring up the consistent findings between ketamine and other anesthetics revealed by their analyses. This result is quite interesting for a number of reasons, few of which are addressed in the discussion. As the authors state, Ketamine acts on NMDA channels, and thus had a different mechanism of action from the other anesthetics used. Thus, the similar findings with Ketamine may, as the authors assert, suggest a consistent mechanism for loss of consciousness. However, even at anesthetic doses, the state produced by higher doses of ketamine is quite different potentially from the state produced by other general anesthetics. Ketamine often preserves high frequency activity, especially in frontal cortex, and patients often report vivid and volatile dreams during ketamine anesthesia. Thus, a growing number of scientists believe that ketamine does not render animals or humans “unconscious” at doses used for general anesthesia, but rather “disconnects” them from the external world. It is possible then that the mechanisms described in this paper do not reflect neural correlates of consciousness, but rather, correlates of sensory disconnection. The authors should discuss this alternate interpretation and any evidence that suggests one interpretation over the other. This should especially be contextualized with respect to their working definition of consciousness and the relative conscious state expected in the different anesthetic and stimulation conditions.

2) The discussion highlights three common weaknesses of the current manuscript, listed as follows: 1 – over reliance on methods from previous manuscripts that are never contextualized or explained to current readers. 2 – inconsistent language about and

definitions for consciousness. 3 – imprecise descriptions of results leading to inaccurate or ambiguous conclusions. Consider the following example found in paragraph 7, where the authors state: “low-amplitude stimulation of the central thalamus has weaker effect on behaviour, than high-amplitude stimulation (Tasserie et al., 2022). Thus, gradient range is influenced by CT stimulation even before this stimulation is sufficient to induce restoration of responsiveness”. Here, the authors put forward behavior as an accurate measure of consciousness. This is already a controversial definition that is not consistently applied across the entire manuscript. Further, it is entirely uninterpretable in the given manuscript because, outside of a citation to their previous paper, the authors never indicate how they measured behavior/responsiveness nor how they quantified the differences in behavior between conscious and unconscious states across the two source data sets. At the same time, the descriptions of the effects on behavior are not specific. In neighboring sentences, the authors describe low-amplitude stimulation as failing to restore responsiveness or producing weaker effects than the high-amplitude stimulation. While both may be accurate, the differences between these statements are not trivial and lead to different interpretations. Is the low-amplitude stimulation condition really “unconscious?” Maybe, if the condition actually leads to no behavioral responsiveness. Maybe not there is some responsiveness resorted, but simply weaker than the high-amplitude condition.

3) Overall, the authors must provide stronger evidence specifically linking their measures of structural/functional connectivity and integration to consciousness. In the original paper, Tasserie 2022, the authors provide evidence relating DBS to consciousness, including arousal scores, entropy, and results from the local/global paradigm. Given the inevitable variability in the level of consciousness for individual stimulations across these scores, I expected the authors to numerically demonstrate the relationship between individual trial measures of gradient range, hierarchical integration, and harmonic energy and behavioral arousal or conscious experience. How do the authors justify this omission when it is critical to drawing the desired link between their measures and consciousness more specifically?

4) Related to the above point, the authors frame their discussion around mechanisms of consciousness, but none of their manipulations selectively influence consciousness. General anesthetics influence neural mechanisms that exceed the scope of consciousness (providing

analgesia and suppressing memory for example). Central thalamic DBS may selectively reinstate consciousness, but it is also possible that it drives arousal and other mechanisms function to reinstate consciousness. If the presence or lack of behavioral responsiveness is the primary measure of consciousness used in the paper, this only further increases the likelihood that the results the authors present here are more strongly linked to sensory connectedness than to consciousness per se. This should either be discussed or refuted if possible.

5) Paragraph 7 importantly describes the different effects across measures, starting first with the effect sizes depicted in figure 8. While it does seem clear that there are effect size differences, these are highly variable across the two datasets. This should be discussed.

6) The authors note in paragraph 7 a somewhat puzzling finding that the gradient range results were maximally influenced by low amplitude central thalamic stimulation. This warrants additional discussion. If gradient range is a correlate of consciousness, why is the effect so weak for the high-amplitude stimulation condition. Why might low-amplitude stimulation produce this result?

7) Figure 8 was quite interesting and did not receive much discussion. I'm curious about the degree to which the authors invite comparisons between the multi-anesthesia data set and the DBS dataset, especially since they are presented on different scales. Superficially, it seems based on the effect size measures provided that the CT high condition had many similarities to the light-sevo condition in terms of net differences from wakefulness. Is it reasonable then to assume there is some behavioral match between the conditions?

8) Most of the paper's findings are largely expected. As the authors note, other papers have demonstrated the interplay between structural and functional connectivity varies across conscious states. Many papers have shown that indices of neural communication and

complexity, both in specific pathways and larger-scale networks, are altered across conscious states. Other papers have even demonstrated that different measures of functional connectivity are consistently reduced across low-conscious states and reinstated by central thalamic DBS in macaque monkeys. It is thus imperative that the authors take greater care in outlining the conceptual leap of this study over others in recent literature. Rather than simply focusing on their design, which other studies have used, they should focus on the interpretability of their findings and how it lends new ideas to the study of consciousness. I recommend including a paragraph discussing other measures of consciousness and how they compare conceptually and functionally to the current study, as well as the advantages of the current measure compared to what others have used.

9) The authors use what they describe as a “local” activation of the central thalamus to produce a distributed effect on cortical integration. This might be expected given the recent evidence linking the central thalamus to consciousness, but also the historic context of central thalamic nuclei as being anatomically “nonspecific”, with broad projections to other brain areas. Specifically the authors have targeted CM, with predominant projections to basal ganglia and sparser connections to cortex. Their targeted region is also very close to CL, with strong projections to frontal and parietal cortex. The authors should use some space in the discussion to address the anatomy of the thalamus and explore why central thalamus, and not VT can influence functional connectivity at different scales in a way that benefits consciousness.

Minor Comments:

1. The authors should review the manuscript carefully to ensure that all relevant citations have been provided correctly as intended throughout the manuscript. As demonstrated by the following examples, citations sometimes seemed to be omitted or ill-matched to the statements being presented.

a. It is unclear why the authors include Suzuki and Larkum, 2020 as their primary citations for the following statement: “In this sense, our results help reconcile the traditional locationist approach to the neural correlates of consciousness with recent advances in

understanding brain function in terms of distributed patterns (Suzuki and Larkum, 2020).” This is an exceptional paper, but from this reviewers recollection, does not seem to match the point being made. If this is the correct citation, more context would be of use.

b. In the discussion, the authors may have provided the wrong list of citations. The authors write: “and the effects of selective thalamic stimulation on consciousness and arousal in animals (Alkire et al., 2007, 2009; Lewis et al., 2015; Bastos et al., 2021; Tasserie et al., 2022) and human patients (Tononi, 2004; Schiff et al., 2007; Staunton, 2008; Mashour et al., 2020). Indeed, recent studies also reported that electrical stimulation of specific central-lateral thalamic nuclei (but not control sites) can restore arousal during anaesthetic-induced loss of responsiveness in macaques (Tononi, 2004; Schiff et al., 2007; Staunton, 2008; Mhuirheartaigh et al., 2010; Lioudyno et al., 2013; Ní Mhuirheartaigh et al., 2013; Vijayan et al., 2013; Akeju et al., 2014; Flores et al., 2017; Hemmings et al., 2019; Kelz et al., 2019; Mashour et al., 2020; Redinbaugh et al., 2020, 2022; Afrasiabi et al., 2021; Bastos et al., 2021; Gammel et al., 2023; Kantonen et al., 2023), counteracting the loss of high-frequency (gamma-band) activity and communication between the thalamus and deep cortical layers, which is induced by anaesthesia.” Based on the statements, it seems like they intend to cite Redinbaugh, Neuron 2020, and not the large list of papers included, which instead seem identical to a larger list earlier in the paper broadly citing that the thalamus may play a role in consciousness. If this was not the intention, it seems inappropriate to cite many disparate papers for such specific results. It is also unclear why Redinbaugh 2020 has been omitted from the earlier point concerning evidence of thalamic stimulation on consciousness and arousal in animals if this is not the intention.

Response to reviewer comments for manuscript “*Local Orchestration of Global Functional Patterns Supporting Loss and Restoration of Consciousness in the Primate Brain*”

We thank the Reviewers for their constructive feedback on our manuscript, which has given us the opportunity to improve our work. Please find below our point-to-point responses to each comment. For ease of reading, reviewers’ feedback is provided in **bold**; and quoted passages from the revised manuscript are shown as indented text.

Reviewer #1 (Remarks to the Author):

Summary

This paper aims to identify ‘signatures of consciousness’ across different forms of anesthesia, and test whether this can be reversed using deep-brain stimulation (DBS) of the thalamus. The results reveal summary measures of large-scale brain patterns that are altered under anesthesia and recovered upon DBS-induced awakening.

General

A strength of this paper is the combination of DBS and pharmacology. However, I have questions about the novel insights and the claims about causality. Please find my detailed comments below, approximately in order of importance.

Major

1. Although I appreciate the value of DBS, to me the results still remain relatively correlational in nature because the summary measures of the large-scale pattern are not explicitly targeted with DBS. Therefore, although it is shown that the disappearance/appearance coincides with changes in consciousness, the brain pattern does not necessarily cause consciousness. As such, I would suggest that the authors reduce the strength of the causal interpretations throughout the paper.

Answer: We thank the reviewer for this comment. We agree with the reviewer and have reduced the causal interpretations throughout the paper accordingly.

2. The ‘signatures of consciousness’ in this paper involve complex methodology yet end up in overly simplistic single-number summary values (gradient 1 range, hierarchical integration, harmonic energy). It is unclear to me how mechanistically meaningful these results are. What insights does this really offer in terms of the biological basis of consciousness?

Answer: The methods we used in this article have been proposed previously as markers of states of consciousness, with the exception of hierarchical integration which (to the best of our knowledge) we apply to consciousness research for the first time. Our results generalise

recently published research using similar methods in humans, demonstrating both generalisability to another species, and generalisability to different anaesthetics, as well as strengthening their association with consciousness through the demonstration that these markers track DBS-induced awakening. We also identified a specific thalamic nucleus as potentially playing a role in establishing these distributed markers, which to our knowledge was not known before, and represents a novel biological insight - although we acknowledge that our present data do not provide a mechanistic model for how the centro-median thalamus would achieve this effect. We now address these advances in the Discussion of our revised manuscript.

“Overall, we showed that distributed patterns of functional activity and connectivity of the primate brain co-vary with sensory connectedness to the environment (and possibly consciousness), its suppression by different anaesthetics, and its restoration by subregion-specific thalamic stimulation. The resulting insights align both with the well-known increase in structure-function coupling observed across a variety of pharmacological and pathological states of unconsciousness across species, and also with prominent theories of consciousness that postulate a central role for integrative processes, which we show here to be reliably disrupted in the primate brain upon anaesthetic-induced sensory disconnection from the environment. Thus, the present work provides several advances. First, we generalise recent findings about the effects of anaesthesia on the human brain (connectome harmonics, gradient range) to non-human primates. Second, we show that these markers generalise across different anaesthetics, including with very different molecular mechanisms of action. Third, we show that these markers of distributed brain function are restored when sensory connectedness to the environment is restored by electrical stimulation of a specific thalamic sub-region, despite continuous anaesthesia, thereby dissociating them from the consciousness-unrelated effects of anaesthetics. Additionally, the DBS results show that although these markers pertain to the distributed functioning of the cortex, they can be influenced by the state of a specific subregion of the thalamus, thereby relating localised and distributed brain function. In this sense, our results help reconcile the traditional locationist approach to the neural correlates of consciousness with recent advances in understanding brain function in terms of distributed patterns. Thus, the neural markers that we consider here are not specific to humans, they are not specific to a particular drug, they track connectedness with the environment even in the presence of continuous anaesthetic administration, and they are under the influence of a specific thalamic sub-region.”

Additionally, we now also provide an additional multivariate analysis of our combined neural markers against behavioural arousal, to evaluate their relative importance (Figure 8 and Figures S15, S16).

3. How consistent were the actual observed whole-brain patterns across animals and types of anesthesia? Given the complex methodology and reduction to simple summary metrics, more extensive validations are needed throughout this work.

Answer: We appreciate this remark, and we have taken the following steps to address it.

Firstly, we now provide in the Supplementary Information, QC plots showing of the rs-fMRI timeseries, functional connectivity, and functional connectivity dynamics, for each scanning session of each animal under each condition.

Secondly, for the gradient analysis with diffusion map embedding, which involves several free parameters, we repeated our analyses varying each of them: the diffusion anisotropy parameter alpha (default 0.5, changed to 0.1 or 0.9); the density parameter (default 10%; changed to 50% or 90%); and the similarity kernel (default: normalised angle similarity; changed to cosine similarity). We show that our main results are also found with these alternative implementations of diffusion map embedding, with certain choices (cosine similarity, greater density) also highlighting additional significant differences that are not observed with the default parameters. As a result of this complementary exploration, we have added Supplementary Figures S4, S5, S6, S7, S8, S9, S10 and S11, and Supplementary Tables S3 to S18.

Thirdly, we have changed how we display the data in our figures. We now use different colours to distinguish the different animals, thereby displaying both the variability across animals, and the variability within each animal and each condition. We also statistically accounted for this variability by changing our statistical approach to linear mixed effects modelling, whereby animal identity is included as a random effect.

We hope that these mitigation steps will provide reassurance about the robustness of our results.

4. The methods state that: “Trials were kept when the row signal did not present signs of artifactual activity, functional connectivity was coherent with the average and dynamic connectivity presented consistent patterns across time”. These criteria are overly vague and it would be useful if the authors could specify explicit thresholds to define ‘coherent’ and ‘consistent’ used for trial selection.

Answer: We appreciate the request for a more thorough explanation of our additional QC procedure. This additional QC step involves visual inspection by an expert neuroimager, and it is the same as was previously implemented in Signorelli et al (2021) for the Multi-anaesthesia dataset. In the Methods section of our revised manuscript, we now provide a more extensive description of the visual QC inspection we performed:

“Furthermore, an extra quality control (QC) cleaning procedure was performed to ensure the quality of the data after time-series extraction (Signorelli et al. 2021). This quality control procedure is based on trial-by-trial visual inspection by an expert neuroimager (C.M.S.), and it is the same as was previously implemented in Signorelli et al (2021). Its adoption ensures that we employ consistent criteria across our two datasets, by adopting the more stringent of the two. We plotted the time series of each region, as well as the static functional connectivity matrix (FC), the dynamic connectivity (dFC) and a Fourier analysis to detect unconventional spikes of activity. For each dataset, visual inspection was first used to become familiar with the characteristics of the entire dataset: how the amplitude spectrum, timeseries, FC and dynamic FC look. Subsequently, each trial was inspected again with particular focus

on two main types of potential artefacts. The first one may correspond to issues with the acquisition and is given by stereotyped sinusoidal oscillatory patterns without variation. The second one may correspond to a head or other movement not corrected properly by our preprocessing procedure. This last artefact can be sometimes recognized by bursts or peaks of activity. Sinusoidal activity generates artificially high functional correlation and peak of frequencies in the Amplitude spectrum plot. Uncorrected movements generate peaks of activity with high functional correlation and sections of high functional correlations in the dynamical FC matrix. If we observed any of these anomalies we rejected the trial, opting to adopt a conservative policy. See Figures S17-S19 for examples of artifact-free and rejected trials.”

We also added Supplementary Figures S17, S18 and S19 with examples of artifact-free and rejected trials, and the expert-curated reasons for rejection:

Figure S17. Example QC plot for an artefact-free trial. The preprocessed and denoised fMRI time-series are plotted, as well as the Fourier spectrum (showing frequencies in the entire range admitted by the band-pass filter), functional connectivity (showing the expected higher correlation between homotopic regions in the two hemispheres, appearing as the two minor diagonals), and functional connectivity dynamics (showing the expected high correlation between consecutive time-points, appearing as a high-value diagonal).

Figure S18. Example QC plot for rejected trials. (A) DBS dataset, Off condition, Run 1: Abnormal oscillatory patterns of activity. Extremely high and uniform FC and dFC values. Concentration in a few frequencies on the spectrum. (B) DBS dataset, Off condition, Run 5: Abnormal oscillatory patterns of activity. High FC and dFC connectivity values. Concentration in a few frequencies on the spectrum. (C) DBS dataset, Off condition, Run 34: Abnormal oscillatory patterns of activity. Peaks of activity around 30 timepoints. Abnormally high FC and dFC values. Concentration in a few frequencies on the spectrum. (D) DBS dataset, 5V CT stimulation condition, Run 15: Sinusoidal patterns of activity resulting in abnormally high FC. Artefactual peak of activity around 300 timepoints also visible as a zone of high dFC.

Figure S18. Example QC plot for rejected trials (continued). (A) DBS dataset, 5V CT stimulation condition, Run 27: Abnormal oscillatory patterns of activity (too much sinusoidal activity), biasing the FC to exhibit unusually high values, followed by a peak of activity around 200-300 timepoints, reflected in a zone of high dFC. (B) Multi-Anaesthesia dataset, Deep Propofol condition, Run 15: Abnormal burst of activity and sinusoidal waves in the first half of the recording, also clearly visible in the dFC. (C) Multi-Anaesthesia dataset, Light Sevoflurane condition, Run 7: Burst of activity and peaks visible in the dFC, and unusual correlation patterns in the FC, with extreme values. (D) Multi-Anaesthesia dataset, Deep Sevo condition, Run 19: Burst of activity with extreme peaks, leading to artifactually high dFC in the first half of the recording, and extreme FC values.

The reader can notice the main differences when comparing them. This visual method proved to be a more sensitive alternative than the automated QC method used by Tasserie et al (2022) based on a cut-off threshold of three standard deviations from signal activity. The automated QC method identifies 6 trials with potential artefacts under awake condition (DBS dataset). However, our visual inspection identified 11 possible artifactual trials (including the same 6 as previously). By adopting the more stringent of the two QC methods, we ensure that we employ consistent criteria across our two datasets (i.e., the more stringent ones from Signorelli et al., 2021). We hope these additions will clarify any concerns regarding the QC procedure.

5. Why were matrices thresholded row-wise to retain only the strongest connections? This approach risks losing a substantial amount of useful information.

Answer: This step is part of the default Diffusion Map Embedding workflow, and is the commonly accepted density parameter in the field (e.g., Margulies et al., 2016; vos de Wael et al., 2020), including recent investigations of perturbed consciousness (Girn et al., 2022; Timmermann et al., 2023; Huang et al., 2023). Nevertheless, as part of our sensitivity and robustness analyses (see response to Q3) we evaluated whether using a different density parameter would change the results. Indeed, we found that 10% may be excessive sparsity for our data: for both the Multi-Anaesthesia and DBS datasets, we found that using 50% or 90% density reveals additional significant differences that were not recovered with the standard parameter (Figures S9, S10).

6. Why was the difference between the maximum and minimum values used for the gradient range, rather than the variance explained? Wouldn't the variance explained offer a more robust index of the gradient strength?

Answer: We used the gradient's range as our marker in accordance with recent literature that has used it as a way of quantifying the hierarchical depth of information processing (Girn et al., 2022; Timmermann et al., 2023; Huang 2023), the difference between minimum and maximum points along a specific gradient corresponds to the diffusion distance between the extremes of the gradient (hence the name of the diffusion map embedding algorithm). In particular, Huang et al (2023) showed that this measure may reflect awareness in humans, being affected by perturbations such as disorders of consciousness and propofol general anaesthesia. Nonetheless, as part of our validation analyses we also adopted alternative gradient-based measures: the ratio of the first eigenvalue to the sum of all eigenvalues (related to the variance explained), which was relatively insensitive to anaesthesia (Figure S4); and the dispersion of the first 3 gradients as per Huang et al (2023), which provided similar results (Figure S6).

Minor

7. The figures used to describe anesthesia versus deep-brain stimulation are strange and confusing. I would strongly encourage the authors to label images using words instead of figures.

Answer: We have replaced the figures with text, as requested.

8. Triple shared first authorship and triple shared last authorship seems a little bit excessive, leading to 6 major contributing authors. This is a question for the Editorial Team, but perhaps there should be some limits on shared roles?

Answer: We appreciate this concern. The expertise of multiple teams was required to make this work possible. The Editor has confirmed that the journal does accept multiple first and last shared authorship.

9. Figure 2 compares statistically against the CT high condition. Why is this condition especially picked out for comparison?

Answer: We thank the reviewer for pointing out the lack of clarity regarding the comparison with CT high condition. The main reason was that in a previous article (Tasserie et al., Sci Adv. 2022), it has been shown that deep brain stimulation (DBS) of central thalamus (CT) restores both arousal and awareness following consciousness loss induced by general anaesthesia. However, upon considering this comment we have come to the conclusion that in the context of the present work, it is more appropriate instead to focus on comparisons against the no-stimulation anaesthesia ("off") condition, since the DBS dataset serves to test the hypothesis that DBS stimulation can counter the effects of anaesthesia, which requires comparing anaesthesia both with wakefulness and with DBS. In the revised manuscript, we have now amended our statistical approach accordingly, and we clarify this in each figure and table.

Reviewer #2 (Remarks to the Author):

The authors present compelling research making use of two valuable and unique fMRI datasets from macaque monkeys used for previous publications. One dataset involves whole-brain imaging of macaques under the influence of different general anesthetics. The other dataset makes use of a now established technique to reverse anesthesia using central thalamic deep brain stimulation first demonstrated by Redinbaugh & colleges in 2020, followed by Bastos & colleges in 2021, and last by the authors of this manuscript in 2022. The authors demonstrate that several measures relating to structural and functional connectivity are consistently altered under general anesthesia and selectively reversed by the central thalamic stimulation, not control stimulation of the VT. Ultimately, they show that anesthesia disrupts signatures of functional integration, which are reversed by stimulations that rouse the monkeys from anesthesia. These results are well-aligned with recent papers using similar methods and support existing theories of consciousness and suggest that conscious experience is enabled by the integration of complex information across different brain regions.

The most notable result of the paper relates to the three-fold finding on global changes of structural and functional connectivity.

In general, the authors report that general anesthesia increases the harmonic energy, reflecting the interdependence of functional connectivity on underlying anatomical structure, and reduces the gradient range and hierarchical integration, reflecting the processing capacity and the network integration respectively. These effects are reversed by central thalamic stimulation, which is associated with increased behavioral responsiveness under general anesthesia.

These results are well-aligned with growing evidence that consciousness depends on key network interactions enabling the complex flow of information, which are consistently perturbed under anesthesia and during other less-conscious states.

This work is likely to be of significance to the field for many reasons. First, as the authors point out, these findings take advantage of “whole-brain” data afforded by fMRI and are well-aligned with different theoretical conceptualizations of consciousness. However, this significance is masked in the current version of the manuscript due to overuse of jargon and limited effort made to contextualize the findings relative to other metrics.

Non-expert readers will struggle to conceptualize these analyses and interpret them into functional mechanisms of the brain that might contribute to consciousness. This is exacerbated by the fact that the authors do not adequately contextualize their research relative to other studies and proposed metrics of consciousness, despite citing a number of studies that have used them with similar state-based comparisons.

The methods of the paper appear sound and state of the art, taking strong advantage of an interesting experimental design as the benefits of fMRI data. However, while the reported results are visually compelling, the current manuscript obscures much of the

statistical information, making it difficult to assess the validity of various claims. This lack of transparency will foster distrust amongst readers and greatly diminishes the significance of the work. Overall, I would not recommend publication without substantial major revisions of the paper to correct three major flaws.

1) Readability: the current manuscript uses too much jargon and does not do a good job of applying terms in a clear, consistent manner throughout the manuscript.

2) Statistical transparency: the current manuscript does not accurately describe all experimental results and makes it hard for readers to find the necessary statistical information to back up their claims.

3) Contextual Significance: the current manuscript does not fully flesh out the relationship between the authors' results and the growing literature not only on consciousness, but central thalamic DBS in macaques, burying the lead.

Answer: We thank the reviewer for providing feedback to improve our work. We concur that we had not been sufficiently clear about the methods when describing our results, and that the broader literature on consciousness deserves more space. Likewise, we appreciate the opportunity to provide greater clarity about our statistical analysis.

Briefly, we have expanded our Introduction to explain both previous results in the same data (to provide context and make this manuscript self-contained), as well as outlining our measures and why we use them. In the Results, we updated our statistics as well as our data visualisation, as well as adding further analyses. We updated the Discussion to provide broader context about previous literature, additional measures of consciousness, and the anatomy and role of the thalamus.

We hope that these additions will satisfy the reviewer's concerns. Our point-to-point responses are provided below.

Introduction:

1) The introduction is somewhat confusing and makes it hard to track the goal of the paper. The authors introduce many concepts with increasingly opaque jargon that are not fleshed out and connected the neural mechanisms until deep in the results section. Readers are likely familiar with concepts such as structural and functional connectivity, but may struggle with terms like "eigenmode decomposition" without additional context and help from the authors. This could be improved by offering clearer definitions of terms, as well as citations drawing links between the proposed measures and the plethora of terms and concepts included in the introduction.

Answer: We have modified the Introduction in order to make it more clear and avoid jargon. We have also included relevant citations. In particular, we have added, for each measure, a dedicated paragraph to explain its meaning and relevance, which is further expanded in the corresponding section of the Results (and full details provided in the Methods).

"Firstly, we consider the brain's hierarchical organisation across scales, by studying the principal gradient of functional connectivity (Margulies et al. 2016; Girn et al. 2022),

The brain's intrinsic functional organisation can be represented in terms of continuous, spatially overlapping whole-brain patterns, termed functional gradients (Maergulies 2016; vos de Wael 2020). Analogous to the principal components of PCA, gradients map functional connectivity to a low-dimensional space where proximity indicates functional similarity. Each gradient corresponds to a dimension in this space. Along each dimension (represented by an eigenvector), regions will be located closer in space, the more similar they are in terms of their functional connectivity with the rest of the brain. The regions whose FC is most different along that dimension constitute opposite extremes that are maximally functionally different. In particular, the principal functional gradient (eigenvector associated with the principal eigenvalue) captures the direction of maximal spatial variation in functional organisation: its range can be interpreted as reflecting the distance between the two extremes of the cortical processing hierarchy, reflecting the depth of information processing (Girn et al. 2022; Timmermann et al. 2023). This principal gradient can also be reshaped by pharmacological intervention (Girn et al. 2022; Timmermann et al. 2023), rendering this approach a promising perspective to consider for the present study."

"Secondly, we study the hierarchical integration of functional signals across scales (Wang et al. 2021). Whereas the range of the principal functional gradient reflects the putative depth of the information-processing hierarchy, hierarchical organisation can also manifest in terms of nested relationships between the system's parts at different scales: lower elements in the hierarchy are recursively combined to form the higher elements (Hilgetag and Goulas 2020). This perspective makes it possible to consider the interplay of integration and segregation of brain signals across scales - which is a central feature of prominent scientific accounts of consciousness (Dehaene and Changeux 2011). Classical graph-theoretic measures such as small-worldness and modularity quantify integration and segregation at a single scale, making them inadequate to capture these properties across multiple hierarchical modules (Rubinov and Sporns 2010; Newman 2006; Wang et al. 2021). Instead, we can obtain insight into the hierarchical relationships between different scales of distributed activation in the brain by going beyond the principal gradient alone (or first 2-3 gradients), and instead considering all functional eigenmodes, as recently introduced by (Wang et al. 2021). This is achieved by characterising the concordance or discordance of regions' allegiance across eigenmode scales, corresponding to regions that are jointly activated (same sign) or alternate (opposite sign) at a given scale. This process results in a nested, modular structure that identifies a hierarchical sub-division of the functional connectome into nested modules, up to the level where each module coincides with a single region, indicative of completely segregated activity. Hierarchical integration and segregation can then be quantified by the relative prevalence of the different eigenmodes, indicated by their associated eigenvalues (Wang et al. 2021)."

"Finally, functional brain activity and connectivity unfold over the network of physical white matter pathways between brain regions: the structural connectome. Therefore, for our final investigation we jointly consider brain structure and function, by leveraging the mathematical framework of "harmonic mode decomposition" (Atasoy, Donnelly, and Pearson 2016) to decompose brain activity into distributed patterns of structure-function coupling: the "harmonic modes" of the structural connectome (Atasoy, Donnelly, and Pearson 2016; Atasoy et al. 2017). Through this decomposition, we

quantify the extent to which brain activity is constrained by the underlying network of structural connectivity. Each harmonic mode is a cortex-spanning activation pattern (eigenmode of the structural connectome) characterised by a specific granularity. The use of connectome-specific harmonic decomposition of cortical activity allows us to quantify the contribution of structural organisation to brain activity, across different spatial granularities scales: from large-scale to fine-grained. This approach therefore goes beyond previous investigations that assessed the similarity of structural and functional connectivity at a single scale (Barttfeld et al., 2015; Uhrig et al., 2018; Demertzi et al., 2019; Gutierrez-Barragan et al., 2021). Additionally, this approach is of particular interest because recent results in humans have shown that distributed harmonic patterns of structure-function dependence relate with human consciousness (Luppi et al. 2023), so we aim to investigate here the cross-species generalisation of these results, and their potential susceptibility to thalamic stimulation.”

2) The goals of the paper as listed in the introduction (test the local vs distributed approach) do not quite match the goals as listed in the discussion (to show selective reinstatement of effects in the DBS condition). In the introduction, the methods are treated like a means to an end, while they are treated like the end goal in the discussion. This should be fixed and made more consistent.

Answer: We have modified the Introduction in order to better capture the original intention of this manuscript. We have paid attention to match with the Discussion section.

From the revised Introduction:

- “The first goal of the present investigation is to establish whether these distributed markers of consciousness that have been recently identified in the human brain, can be generalised to the non-human primate brain, and whether they behave consistently across anaesthetics with distinct molecular mechanisms.”
- “Our second goal is to determine whether the reorganisation of distributed brain function induced by anaesthesia can be reversed by targeted electrical stimulation of the central thalamus, concomitantly with restoration of behavioural evidence of consciousness.”

From the revised Discussion:

- “Our goals were twofold. First, we sought to determine whether distributed signatures of consciousness that have been recently identified in the human brain, can be generalised to the non-human primate brain, and whether they behave consistently across anaesthetics with distinct molecular mechanisms. Second, we aimed to determine whether the reorganisation of distributed brain function consistently induced by different anaesthetics can be reversed by targeted stimulation of different subregions of the thalamus, concomitant with the restoration of behavioural arousal.”

3) Some terms vital to the reader’s understanding of the paper go undefined across the manuscript. This is especially noticeable with the term “consciousness”. As the authors are no-doubt aware, consciousness is a term with many working definitions across the field of neuroscience, but most popularly, as a multivariate construct

combining elements of animal arousal and awareness. Consciousness can be somewhat separated from the idea of conscious states, where consciousness can be assumed to be higher, lower, or possibly not present (a claim that requires substantial evidence and analysis). Unfortunately, the authors make no effort currently to provide their working definition of consciousness and flexibly switch across the paper between terms. The DBS effects are described as increasing conscious arousal, or sometimes behavioral responsiveness, while anesthesia is referred to as producing a “loss of consciousness” and is described as a consciousness state. Overall, the terminology is imprecise and mixed, greatly hindering both readability and the fidelity of their conclusions later in the paper.

Answer: Indeed, consciousness is a difficult construct to define, especially in non-human animals who cannot report their subjective experience using language. To clarify: here we used the behavioural assessment of arousal (according to an 11-point preclinical scale which includes responsiveness to the environment) as our proxy for consciousness. This is common practice in animal studies, and we note that in clinical practice in humans, both the diagnosis of disorders of consciousness, and the assessment of sedation depth (e.g., Ramsay scale) are based on similar behavioural criteria. We have now added a full description of the arousal scale (see Methods sub-section “Behavioural assessment of arousal”) as well as its outcome in both datasets (see Figure S2 and Table S25), and a new analysis relating this measure to our neural signatures (Figure 8). We have also amended our manuscript to be more specific in our description of the effects of anaesthesia and DBS. Nonetheless, we also acknowledge that behavioural responsiveness is conceptually distinct from consciousness, and it is an imperfect marker.

“Since non-human primates cannot use language to report their subjective experience, we rely on behavioural measures of arousal as our proxy for consciousness, as is common practice when working with animal models of anaesthesia, and also widely used in clinical practice (Mashour et al. 2021). Nonetheless, we acknowledge that unresponsiveness is not equal with unconsciousness (Sanders et al. 2012; Luppi et al. 2021).”

We believe that the inference to unconsciousness may be justified for deep propofol and sevoflurane anaesthesia, because in humans, levels of anaesthetics that induce analogous levels of behavioural unresponsiveness also suppress subjective experience, as far as can be determined. Additionally, in the original analysis of the present DBS dataset, it was also shown that anaesthesia suppresses neural signatures of awareness including a rich repertoire of functional brain states independent from the structure (dynamic analysis of resting-state) and the ability to detect a complex sequence rule violation (evoked response to auditory stimulation). These neural markers were restored by high-amplitude CT DBS. Although our current study only uses resting-state fMRI data, we now explain these results in the revised Introduction, to clarify that potential markers of consciousness beyond arousal are restored by CT DBS.

“In this dataset, it was shown that high-amplitude CT stimulation restores behavioural markers of arousal (eye opening, movement, reflexes and exploratory behaviour) that were suppressed by anaesthesia, as well as restoring neural signatures including dynamic uncoupling of functional from structural connectivity, the complexity of EEG

signals, and neural processing of auditory stimuli beyond sensory cortices, which have been argued to be necessary conditions for consciousness (Casali 2021; Dehaene 2011; Mashour 2020). ”

Although these results arguably warrant interpretation in terms of awareness, we refrain from doing so based on the observation (in humans) that ketamine may induce equivalent suppression of arousal and responsiveness, but without suppressing subjective experience - instead enacting a sensory disconnection from the environment. Since our interest is in common features of different anaesthetics, sensory disconnection from the environment - rather than full-blown loss of consciousness - represents the “minimum common denominator” across the anaesthetics employed in our two datasets. Therefore, we address arousal as our objective marker, and frame our interpretation in terms of presence or absence of sensory connectedness to the environment.

“Despite the different molecular mechanisms of action, at anaesthetic doses such as the ones used here ketamine induces full loss of behavioural responsiveness, as well as suppression of complex cortical and thalamic responses to deviant stimuli in the local-global auditory paradigm (Uhrig 2016). However, there is evidence that even at anaesthetic doses, ketamine may induce loss of responsiveness not by suppressing consciousness (as propofol and sevoflurane are do) but rather by inducing sensory disconnection from the environment, while potentially preserving subjective experience (Bonhomme 2016 Anesthesiology; Luppi 2023 Science Advances The preservation of subjective experience despite sensory disconnection from the environment (and consequently, loss of behavioural responsiveness to the environment) is commonly experience during dreaming - and indeed, ketamine-anaesthetised volunteers are known to report vivid hallucinations and dream-like experiences (Bonhomme et al. 2016). This phenomenon highlights the fact that behavioural unresponsiveness is an imperfect marker of unconsciousness, since it can also occur as a result of sensory disconnection, or motor impairment, neither of which is the same as unconsciousness (Luppi et al. 2021; Sanders et al. 2012). Since it is not presently possible to obtain subjective reports from the animals involved in our study, a more conservative interpretation of our results is that they pertain to sensory disconnection from the environment, which is a common effect shared by ketamine, propofol, and sevoflurane. Therefore, in the remainder we interpret our results as pertaining to sensory disconnection from the environment (SDE), although we consider it likely that at least the animals anaesthetised with sevoflurane and propofol (which was the drug used for the DBS experiments) were in fact fully unconscious.”

4) Because the authors are introducing so many terms and not providing clear definitions in the introduction, it is easy to get confused or misunderstand the authors’ logic, especially because some of the same terms are used to reference different things throughout the paper. For example, the authors refer both to a local vs distributed approach to the study of consciousness, meaning to differentiate between studies that seek to assess the relevance of a given brain area (local) from studies that seek to address global metrics of neural connectivity and function. They also then use the term localized to refer to smaller-scale network interactions from global interactions. Thus, in a given sentence, local may refer to either a specific brain region, or the scale of

network connectivity, which is confusing to readers and muddies interpretation of the author's words. The authors also use the term "integration" to describe functional connectivity, while also using it to reference their approach of combining different scales of analysis with fMRI. Readability would be greatly improved if the authors took more care to do the following:

- a. use unique terms to reference unique concepts**
- b. define terms when they are first used conceptually, and then operationalized as different computational metrics**
- c. Strictly adhere to terms and refrain from using them out of context.**

Answer: We apologise for our lack of clarity and sometimes inconsistent use of terminology. We have now clearly distinguished our use of the various terms. We have also rephrased to avoid using "integration" to the combination of different modalities or methodologies. Where some concepts (such as "hierarchy") have inherently multiple meanings in the literature, we now clarify which meaning we refer to upon use (e.g., hierarchical processing versus nested hierarchy). We have also replaced most uses of "global" with "distributed" where the latter better represents our approach. Likewise, we have replaced the use of "local/localised" with "region-specific" or "fine-grained" wherever appropriate, to further disambiguate its use. We hope that these amendments will improve the readability and clarity of our manuscript.

5) The way the authors suggest that one innovative component of the current study is its use of whole-brain imaging to provide a global approach. While I agree that this is a strength of the current study, I disagree with the framing of the debate, as it suggests that studies either test the contributions of individual areas, or operate at global network scales, and there is no overlap between the two. Indeed, there is a sizable middle ground inhabited by many studies that test the contributions of individual areas as well as complex information transfer in specific long-range projection pathways or through complex networks. It is unclear where the authors delineate between what counts as a local vs global study, or why they seem to suggest that electrophysiology studies mostly reside in the local category.

Answer: We would argue that the previous electrophysiological studies provide information that is more specific because they used electrodes to record from specific regions, whereas the fMRI data used here (first published in Tasserie et al., 2022) afford coverage of the entire cortex, and are therefore better suited to a distributed approach such as the one adopted here. To clarify, eigenmode-based approaches are inherently non-local because they involve re-representing the data in the domain of graph frequencies, a format that does not involve the distinction between spatial coordinates at all - just like re-representing a temporal signal into the domain of temporal frequencies does not distinguish between time-points any longer. It is in this sense that our measures (which are all based on eigenmodes) stand in contrast to location-focused approaches. To reduce the possibility of confusion, we have now avoided the use of "global", and instead more consistently use the term "distributed" which better encompasses our approach. We have also clarified that the location-focused approach can encompass not just single regions but also circuits and spatially discontinuous networks, covering via different projection pathways the middle ground between single regions and whole brain:

“The traditional approach to the neuroscience of consciousness has been to look for specific and localised brain regions (and more recently, collections of brain regions in the form of circuits and spatially discontinuous networks) that support consciousness”.

6) The authors introduce the overall structure of their manuscript by comparing metrics across different anesthetics, and then following DBS of the central thalamus to restore consciousness under anesthesia with the following sentence: “To this end, we leverage functional MRI (fMRI) data from non-human primates in the awake state, under loss of consciousness induced by three different anaesthetics (sevoflurane, propofol, ketamine) and restoration of consciousness by deep brain stimulation (DBS).” The authors should provide the relevant references for this DBS effect (Redinbaugh 2020, Bastos 2021, & Tasserie 2022) and clarify to readers that this is an established well-documented technique in macaque monkeys.

Answer: We thank the reviewer for this observation, and we agree. In the revised version of the manuscript, we have included the references (both in this passage and in other places where relevant) and also clarified that thalamic stimulation is by now a well-documented technique to modulate markers of consciousness in macaque monkeys.

“we leverage the experimental accessibility of animal models, analysing a unique recently published dataset of macaque resting-state fMRI acquired under deep propofol anaesthesia with and without direct intervention on the thalamus, in the form of deep brain stimulation (DBS) of the centro-median (CT) or ventral-lateral thalamus (VT) (Tasserie et al., 2022). Electrical stimulation of the central thalamus has become an established approach for modulating behavioural and neural markers of consciousness in non-human primates (Redinbaugh 2020, Bastos 2021, & Tasserie 2022)”

7) The authors make some bold claims about the goal of the paper and the interpretation of the findings in the introduction: “Comparing the brain-wide effects of different anaesthetics enables us to disentangle which aspects of the brain’s functional organisation support consciousness, being consistently targeted by anaesthetics, despite their distinct molecular mechanisms.” This claim does not seem born out by their own conclusions and discussion. The current manuscript does little to disentangle aspects of functional organization that play a causal role towards consciousness, likely because the experimental design does not test these causal factors. I welcome the authors to provide mechanistic arguments in their conclusion, but in the current version of the manuscript, they are not present and not warranted by the statistical analysis or design.

Answer: We agree with the reviewer. Indeed, our current design does not afford to disentangle the different aspects of brain’s functional organisation that support consciousness. However, here we investigate and compare brain-wide effects of different anaesthetics. We have rephrased as follows:

“Comparing the brain-wide effects of different anaesthetics enables us to characterise which aspects of the brain’s distributed functional organisation are consistently targeted by anaesthetics, despite their distinct molecular mechanisms. By further assessing whether different aspects of distributed brain function are restored upon concomitant restoration of behavioural arousal (our indicator of consciousness) by subregion-specific thalamic stimulation, we can further strengthen their association with consciousness. Specifically, a genuine neural correlate of consciousness should not only be compromised under anaesthesia: it should also be restored when consciousness (here indicated by arousal) is restored, even in the presence of continuous anaesthetic infusion.”

8) The authors make a serious logical error in their introduction related to the interpretation of their results by claiming they demonstrate a causal link between their structural measures and consciousness. The authors claim: “Crucially, we also aim to obtain more stringent evidence for the causal relevance of distributed signature of consciousness, by determining whether the reorganisation of distributed brain function consistently induced by different anaesthetics can be reversed by targeted stimulation of different subregions of the thalamus, a brain structure that has been repeatedly associated with supporting consciousness.” The authors do causally influence consciousness with both their anesthetic and DBS manipulations. These manipulations are associated with changes in functional organization, and presumably, consciousness. However, it is not appropriate to assume a causal link between the functional reorganization and consciousness without substantial analysis not provided in the current manuscript. While anesthesia is expected to reduce consciousness, it is unclear when loss of consciousness happens in this study in relation to the specific reductions noted in the measures of functional organization. The same is true for the DBS experiments. It is perfectly possible that the structural reorganization is an unnecessary correlate of consciousness, or fully insufficient to support consciousness. All the authors can truly claim in this paper is that it is a correlate of conscious states. Thus, It would be more appropriate for the authors to state that their causal manipulations allow them to reveal specific neural correlates of consciousness that hold across anesthetics and DBS conditions. Any arguments about why this effect might be causal should be reserved as a hypothesis for the discussion, and well substantiated by existing literature or theory. A later sentence offered by the authors accurately portrays the power of their design: “This dual causal manipulation - pharmacology and electrical stimulation - provides us with a unique opportunity to study functional changes that are observed during loss of consciousness and reappear upon recovery of consciousness, despite continuous anaesthetic infusion.” As the authors seem aware, their design does not afford the logical or computational power to make causal claims, but does help demonstrate that the effects are not specific to the stringent chemical effects of the anesthetic used in their DBS experiments.

Answer: We thank the reviewer for this important remark. We have reduced the causal interpretations throughout the paper (see also our response to the previous comment).

Results/Methods:

1) The authors provide considerable useful context in this section explaining the methods they use and constructs they seek to characterize. For example, they clearly describe how the eigenmode gradient depth relates to the capacity of neural computation and processing. Some of this information should be provided in the introduction to prime the readers with context and provide citations about the links between these constructs and consciousness. In general, I would recommend using more space in the introduction to describe measures of neural computation and their theoretical relationship to consciousness, and space in the results to flesh out the operationalization used in this paper.

Answer: We thank the reviewer for this observation, and we appreciate the opportunity to elaborate on our measures of interest and their relevance. In the revised Introduction, we have included a paragraph for each measure, explaining its meaning and rationale:

“Firstly, we consider the brain’s hierarchical organisation across scales, by studying the principal gradient of functional connectivity (Margulies et al. 2016; Girn et al. 2022), The brain’s intrinsic functional organisation can be represented in terms of continuous, spatially overlapping whole-brain patterns, termed functional gradients (Maergulies 2016; vos de Wael 2020). Analogous to the principal components of PCA, gradients map functional connectivity to a low-dimensional space where proximity indicates functional similarity. Each gradient corresponds to a dimension in this space. Along each dimension (represented by an eigenvector), regions will be located closer in space, the more similar they are in terms of their functional connectivity with the rest of the brain. The regions whose FC is most different along that dimension constitute opposite extremes that are maximally functionally different. In particular, the principal functional gradient (eigenvector associated with the principal eigenvalue) captures the direction of maximal spatial variation in functional organisation: its range can be interpreted as reflecting the distance between the two extremes of the cortical processing hierarchy, reflecting the depth of information processing (Girn et al. 2022; Timmermann et al. 2023). This principal gradient can also be reshaped by pharmacological intervention (Girn et al. 2022; Timmermann et al. 2023), rendering this approach a promising perspective to consider for the present study.”

“Secondly, we study the hierarchical integration of functional signals across scales (Wang et al. 2021). Whereas the range of the principal functional gradient reflects the putative depth of the information-processing hierarchy, hierarchical organisation can also manifest in terms of nested relationships between the system’s parts at different scales: lower elements in the hierarchy are recursively combined to form the higher elements (Hilgetag and Goulas 2020). This perspective makes it possible to consider the interplay of integration and segregation of brain signals across scales - which is a central feature of prominent scientific accounts of consciousness (Dehaene and Changeux 2011). Classical graph-theoretic measures such as small-worldness and modularity quantify integration and segregation at a single scale, making them inadequate to capture these properties across multiple hierarchical modules (Rubinov and Sporns 2010; Newman 2006; Wang et al. 2021). Instead, we can obtain insight into the hierarchical relationships between different scales of distributed activation in

the brain by going beyond the principal gradient alone (or first 2-3 gradients), and instead considering all functional eigenmodes, as recently introduced by (Wang et al. 2021). This is achieved by characterising the concordance or discordance of regions' allegiance across eigenmode scales, corresponding to regions that are jointly activated (same sign) or alternate (opposite sign) at a given scale. This process results in a nested, modular structure that identifies a hierarchical sub-division of the functional connectome into nested modules, up to the level where each module coincides with a single region, indicative of completely segregated activity. Hierarchical integration and segregation can then be quantified by the relative prevalence of the different eigenmodes, indicated by their associated eigenvalues (Wang et al. 2021)."

"Finally, functional brain activity and connectivity unfold over the network of physical white matter pathways between brain regions: the structural connectome. Therefore, for our final investigation we jointly consider brain structure and function, by leveraging the mathematical framework of "harmonic mode decomposition" (Atasoy, Donnelly, and Pearson 2016) to decompose brain activity into distributed patterns of structure-function coupling: the "harmonic modes" of the structural connectome (Atasoy, Donnelly, and Pearson 2016; Atasoy et al. 2017). Through this decomposition, we quantify the extent to which brain activity is constrained by the underlying network of structural connectivity. Each harmonic mode is a cortex-spanning activation pattern (eigenmode of the structural connectome) characterised by a specific granularity. The use of connectome-specific harmonic decomposition of cortical activity allows us to quantify the contribution of structural organisation to brain activity, across different spatial granularities scales: from large-scale to fine-grained. This approach therefore goes beyond previous investigations that assessed the similarity of structural and functional connectivity at a single scale (Barttfeld et al., 2015; Uhrig et al., 2018; Demertzi et al., 2019; Gutierrez-Barragan et al., 2021). Additionally, this approach is of particular interest because recent results in humans have shown that distributed harmonic patterns of structure-function dependence relate with human consciousness (Luppi et al. 2023), so we aim to investigate here the cross-species generalisation of these results, and their potential susceptibility to thalamic stimulation."

2) In the introduction and methods, the authors make it clear that this paper is based on datasets which have already been used in other publications. Many of the sections in the methods reference readers back to the original publications for specific details. While this is fine in principle, the authors must do more to clarify the link between the datasets and any differences that might exist. Are the data sets in question identical (all using the exact same animals, trials, stimulation events and anesthetic sessions with no changes)? If so, that should be clearly stated to alleviate any questions from readers. If not, the authors should clarify which portions of the dataset are shared between the source papers and why data might have been omitted. This is critical for readers to interpret the results between both papers. For example, some data from the Tasserie paper included auditory stimulation in a local-global paradigm. Is this data included in the current manuscript? This seems unlikely as the authors claim their analyses did not include task data, which is present in Tasserie 2022.

Answer: We thank the reviewer for raising these concerns. Indeed, the datasets used in this article are the same as in previous publications. The data acquisition and the preprocessing is exactly the same across this and previous publications. The animals used are also the same as previous publications. In both cases, we only use resting-state data. However, notice that the multi anaesthesia dataset and the DBS dataset do not use the same animals. The Multi-Anaesthesia dataset uses MION as contrast-enhancing agent, whereas the DBS dataset uses BOLD. This is explained in the Methods, and we have now also addressed differences between datasets in the Discussion:

“Direct comparisons between the neural data obtained in the two datasets are not straightforward: the Multi-Anaesthesia dataset used MION as a contrast agent to improve signal, whereas the DBS used the more commonly used BOLD signal. Additionally, the acquisition parameters were different (e.g., 2.4s vs 1.25s TR), and the DBS dataset did not compare anaesthesia and wakefulness in the same animals, as did the Multi-Anaesthesia dataset. Despite these differences, however, significant differences in the same direction were found for the deep propofol condition of the Multi-Anaesthesia dataset, and the propofol anaesthesia condition without stimulation (“off”) of the DBS dataset, which is the only contrast that was present in both datasets. This replication demonstrates the robustness of our markers to acquisition differences, in addition to their generalisation across anaesthetics and across species.”

We have also amended our Introduction as follows, to clarify that the two datasets are independent and that we only consider rs-fMRI data (i.e., no event-related data from the auditory stimulation).

“We leverage a previously published dataset of resting-state functional MRI (fMRI) data from non-human primates in the awake state, and under loss of consciousness induced by three different anaesthetics (sevoflurane, propofol, ketamine). We combine this with a separate recently published dataset of non-human primate resting-state fMRI under propofol anaesthesia and subsequent deep brain stimulation (DBS) of different sub-regions of the thalamus.”

We now also explicitly state in the Methods that we did not use the task-based fMRI data in this work:

“Event-related data were also acquired and are reported in Tasserie et al (2022), but here we only used the resting-state fMRI data, and will not discuss the event-related data further.”

We also clarify that :

“For the Multi-Anaesthesia dataset, five rhesus macaques were included for analyses (Macaca mulatta, one male, monkey J, and four females, monkey A, K, Ki, and R)”

whereas:

“For the DBS dataset, five male rhesus macaques (*Macaca mulatta*, 9 to 17 years and 7.5 to 9.1 kg) were included, three for the awake (non-DBS) experiments (monkeys B, J, and Y) and two for the DBS experiments (monkeys N and T).”

The trials retained for the Multi-Anaesthesia dataset are exactly the same as in Signorelli et al (2021). However, for the DBS dataset fewer trials were included than in Tasserie et al (2022). This is because we performed an additional quality control (QC) step, in addition to the automated artifact rejection implemented by Tasserie et al (2022), which used a computed threshold of three standard deviations from signal activity. This additional QC step involves visual inspection by an expert neuroimager, and it is the same as was previously implemented in Signorelli et al (2021), thereby ensuring that we employ consistent criteria across our two datasets, by adopting the more stringent of the two. For example, for the DBS dataset’s awake condition, the automated QC detected 6 trials with potential artefacts. Thanks to our visual inspection, for example, we identified 11 possible artifactual trials under awake condition (including the same 6 as previously). In the Methods, we specify the number of total trials acquired versus the total trials analysed:

“As a result, for the Multi-Anaesthesia data set a total of 119 runs are analysed in subsequent sections (the same as used in Signorelli et al., 2021): awake state 24 runs, ketamine anaesthesia 22 runs, light propofol anaesthesia 21 runs, deep propofol anaesthesia 23 runs, light sevoflurane anaesthesia 18 runs, deep sevoflurane anaesthesia 11 runs. For the DBS data set, a total of 156 runs are analysed in subsequent sections: awake state 36 runs, Off condition (propofol anaesthesia without stimulation) 28 runs, low-amplitude CT stimulation 31 runs, low-amplitude VT stimulation 18 runs, high-amplitude CT stimulation 25 runs, high-amplitude VT stimulation 18 runs.”

We have also added a more extensive description of the visual QC inspection we performed (please also refer to question 2, reviewer 1) in the Methods.

“Furthermore, an extra quality control (QC) cleaning procedure was performed to ensure the quality of the data after time-series extraction (Signorelli et al. 2021). This quality control procedure is based on trial-by-trial visual inspection by an expert neuroimager (C.M.S.), and it is the same as was previously implemented in Signorelli et al (2021). Its adoption ensures that we employ consistent criteria across our two datasets, by adopting the more stringent of the two. We plotted the time series of each region, as well as the static functional connectivity matrix (FC), the dynamic connectivity (dFC) and a Fourier analysis to detect unconventional spikes of activity. For each dataset, visual inspection was first used to become familiar with the characteristics of the entire dataset: how the amplitude spectrum, timeseries, FC and dynamic FC look. Subsequently, each trial was inspected again with particular focus on two main types of potential artefacts. The first one may correspond to issues with the acquisition and is given by stereotyped sinusoidal oscillatory patterns without variation. The second one may correspond to a head or other movement not corrected properly by our preprocessing procedure. This last artefact can be sometimes recognized by bursts or peaks of activity. Sinusoidal activity generates artificially high functional correlation and peak of frequencies in the Amplitude spectrum plot. Uncorrected movements generate peaks of activity with high functional correlation and

sections of high functional correlations in the dynamical FC matrix. If we observed any of these anomalies we rejected the trial, opting to adopt a conservative policy. See Figures S17-S19 for examples of artifact-free and rejected trials.”

Lastly, we added Supplementary Figures S17, S18 and S19 with examples of artifact-free and rejected trials, and the expert-curated reasons for rejection:

Figure S17. Example QC plot for an artifact-free trial. The preprocessed and denoised fMRI timeseries are plotted, as well as the Fourier spectrum (showing frequencies in the entire range admitted by the band-pass filter), functional connectivity (showing the expected higher correlation between homotopic regions in the two hemispheres, appearing as the two minor diagonals), and functional connectivity dynamics (showing the expected high correlation between consecutive time-points, appearing as a high-value diagonal).

Figure S18. Example QC plot for rejected trials. (A) DBS dataset, Off condition, Run 1: Abnormal oscillatory patterns of activity. Extremely high and uniform FC and dFC values. Concentration is a few frequencies on the spectrum. (B) DBS dataset, Off condition, Run 5: Abnormal oscillatory patterns of activity. High FC and dFC connectivity values. Concentration is a few frequencies on the spectrum. (C) DBS dataset, Off condition, Run 34: Abnormal oscillatory patterns of activity. Peaks of activity around 30 timepoints. Abnormally high FC and dFC values. Concentration in a few frequencies on the spectrum. (D) DBS dataset, 5V CT stimulation condition, Run 15: Sinusoidal patterns of activity resulting in abnormally high FC. Artefactual peak of activity around 300 timepoints also visible as a zone of high dFC.

Figure S18. Example QC plot for rejected trials (continued). (A) DBS dataset, 5V CT stimulation condition, Run 27: Abnormal oscillatory patterns of activity (too much sinusoidal activity), biasing the FC to exhibit unusually high values, followed by a peak of activity around 200-300 timepoints, reflected in a zone of high dFC. (B) Multi-Anaesthesia dataset, Deep Propofol condition, Run 15: Abnormal burst of activity and sinusoidal waves in the first half of the recording, also clearly visible in the dFC. (C) Multi-Anaesthesia dataset, Light Sevoflurane condition, Run 7: Burst of activity and peaks visible in the dFC, and unusual correlation patterns in the FC, with extreme values. (D) Multi-Anaesthesia dataset, Deep Sevo condition, Run 19: Burst of activity with extreme peaks, leading to artifactually high dFC in the first half of the recording, and extreme FC values.

3) The authors should provide more context from the Tasserie 2022 paper in the results and methods section to make it clear to readers how they measured changes in consciousness induced by DBS and validated that their different DBS conditions link back to different states. Presently, readers are simply expected to take the authors at their word that higher amplitude central thalamic stimulation was the only condition that consistently increased consciousness under anesthesia. While it is likely that readers will also engage with the Tasserie paper, it is currently unclear if the datasets are factually identical in terms of treatment and analysis. Are all stimulations used? Were any omitted? Were any central thalamic stimulations ever ineffective at the higher voltage to restore consciousness? Is there a cutoff the authors used? Were any ventral lateral stimulations ever mildly effective? At the very least, the authors should reiterate for the reader's benefit basic evidence (quantified if possible) so they can more accurately compare results between these conditions.

Answer: We appreciate the reviewer's constructive comment and have updated the revised manuscript accordingly.

To clarify: we used the same five stimulation conditions as in the study by Tasserie et al.: DBS off and ON, centred to CT or VT, at low or high stimulation amplitudes, without any omissions. In the Sci Adv article, changes in consciousness induced by DBS at these various conditions were measured using behavioural scores, physiological parameters, EEG, and fMRI signals, encompassing block-designed, resting, and task states.

The effects of stimulation consistently exhibited a remarkable specificity and sensibility. As reported in Tasserie et al (2022):

- Based on the behavioural score, only high CT DBS invariably induced arousal in an awake-like manner, while low CT DBS partially restored some features. In contrast, anaesthesia with no stimulation ("DBS off") or VT DBS, whether at low or high amplitude, never induced any behavioural response (Awake: 11/11 score on the preclinical arousal scale; DBS off, low VT DBS and high VT DBS: all 0/11 for all animals; low CT DBS: 3/11 and high CT: 9/11) .
- Similar consistency was observed in EEG spectral and entropy measures, where high CT DBS always exhibited significantly higher values compared to other stimulation conditions ($p < 0.01$ FDR corrected, i.e.. lower delta, higher theta and alpha).
- Whole-brain fMRI responses to different DBS targets and levels (cycling on and off the same stimulation using a block design pattern) reliably revealed that high CT DBS activated a broad cortical and subcortical network, including the cingulate cortex. This level of network activation was never achieved in the other conditions ($p < 0.01$ FWE corrected).
- Static analysis of resting-state fMRI unceasingly demonstrated that long-range bilateral cortico-cortical and thalamo-cortical correlations, present in awake animals, were reliably and uniquely restored under high CT DBS. None of the other DBS conditions achieved such a reconfiguration ($p < 0.001$ FDR corrected).
- Additionally, the investigation of temporal dynamics within these same rs-fMRI scans consistently showed that high CT DBS enriched cortical dynamics by decreasing the function-structure similarity (mean rank awake = 4.38, DBS off = 5.70 and high CT DBS = 4.55). Brain states analysis revealed a broad repertoire of brain states solely restored under high CT DBS (for e.g. mean rank high VT DBS = VLT DBS 5V = 6.41). This effect was never observed during the different control DBS experiments (respectively $p > 0.4$ and $BF > 3$; $p < 0.01$ FDR corrected).
- Furthermore, during a passive auditory task, high CT DBS consistently restored the processing of sequence rule violation (high order / global deviance, $p < 0.05$ FDR corrected), reactivating a homologous thalamo-prefronto-parieto-cingulate network to that observed in awake monkeys.

We have also added additional information in the Methods pertaining to behavioural assessment of arousal, DBS electrode implantation and stimulation protocol (see new sections "Behavioural assessment of arousal" and "Deep Brain Stimulation protocol"). Furthermore, we added Figures S1 and S2 to illustrate the location of the DBS electrodes and the effect of

different stimulation protocols on arousal, and we added Table S25 to show the arousal scores and drug levels for the Multi-Anaesthesia dataset.

We also added, in the Introduction, a description of the main results of Tasserie et al (2022), to familiarise the reader:

“we leverage the experimental accessibility of animal models, analysing a unique recently published dataset of macaque resting-state fMRI acquired under deep propofol anaesthesia with and without direct intervention on the thalamus, in the form of deep brain stimulation (DBS) of the centro-median (CT) and ventral-lateral thalamus (VT) (Tasserie et al., 2022). In this dataset, it was shown that high-amplitude CT stimulation restores behavioural markers of arousal (eye opening, movement, reflexes and exploratory behaviour) that were suppressed by anaesthesia, as well as restoring neural signatures including dynamic uncoupling of functional from structural connectivity, the complexity of EEG signals, and neural processing of auditory stimuli beyond sensory cortices, which have been argued to be signatures of consciousness (Casali 2021; Dehaene 2011; Mashour 2020). This effect was greatly diminished at lower amplitude DBS, and entirely absent upon stimulation of a different thalamic nucleus, demonstrating an exquisite level of spatial specificity.”

4) The statistical reporting has been greatly improved by the provided supplemental document and it must be fully incorporated into the manuscript. There should be a dedicated statistical section in the methods providing additional details. Specifically, the authors should make it clear how they performed the ANOVA analyses as well as the pairwise T-tests, including which software, if any, they used. Points in the method section making specific claims about the relationship of different groups should have the relevant statistical results provided in the text, and the rest should be found in the provided tables in the supplement. The authors should also clearly state in the figure captions, where they provide p-values, the name of the relevant test that the p-values reflect. They should similarly clarify the name of the effect size measure. Full transparency is increasingly necessary and demanded by readers. The authors should ensure that readers have easy access to all statistical information to back their claims.

Answer: We appreciate the need for both rigour and transparency. We have now added a dedicated Statistical Reporting section in the Methods, as per the reviewer’s recommendation. We now adopted linear mixed effects modelling to account for the non-independence both within and between conditions. We also complemented our traditional significance testing with the recently developed Bayes Factor Functions, which enable traditional test statistics to be used to obtain Bayes Factors as a function of expected effect size. We report ANOVA results in the main text, and indicate in each figure that statistical results were obtained from linear mixed effects modelling. Since adding 24 statistical tables to the main text would not be feasible, we provide them as a Supplementary Excel file. They include LME parameters, including confidence intervals, FDR-corrected p-values, and Bayes Factors for four ranges of effect sizes: negligible, small, medium, and large.

“Statistical Reporting

Overall significance was assessed with a one-way mixed effects analysis of variance (ANOVA, implemented using MATLAB's `fitlme` function), with condition as a fixed effect, and animal identity as random effect. Subsequently, we performed individual pairwise contrasts using linear mixed effects modelling (also implemented using MATLAB's `fitlme` function), with condition as the fixed effect and animal identity as the random effect. This approach enabled us to take into account the fact that the same animal could provide more than one data-point to each condition, as well as contributing data-points for more than one condition.

For the Multi-Anaesthesia dataset, each anaesthesia condition was compared against wakefulness, to test the hypothesis that our neural markers are affected by anaesthesia. For the DBS dataset, we compared the DBS-off (no stimulation) anaesthesia condition, against all other conditions: wakefulness, and all stimulation types. This allowed us to replicate the effects of anaesthesia on our neural markers of interest (by comparing Awake against no-stimulation anaesthesia), and to test the hypothesis that CT stimulation should counter the effects of anaesthesia on our neural markers. In both datasets, correction for multiple comparisons was carried out using the False Discovery Rate correction (Benjamini and Hochberg, 1995). We report all statistical tests and descriptive statistics in Supplementary Tables S1 to S24.

To assess statistical significance of the dominance analysis multiple regression, we built a null distribution of adjusted R^2 values by repeating the multiple regression 1,000 times, but with permuted assignment of the arousal scores, and evaluating how often the null adjusted R^2 was greater than the empirical adjusted R^2 .

A drawback of traditional significance testing is that it cannot formally accept the null hypothesis, and a failed rejection of the alternative hypothesis can occur because the null hypothesis is true, but also because statistical power is insufficient. In addition to standard significance testing, we therefore also adopted the recently developed R package for Bayes Factor Functions (Johnson et al., 2023) to quantify Bayesian evidence in support of the alternative hypothesis versus the null hypothesis, based on our test statistics. This approach makes it possible to distinguish whether the evidence actively supports the alternative hypothesis, or it actively supports the null hypothesis, or it is insufficient to discriminate between them - as a function of the expected effect size (Johnson et al., 2023). In our Supplementary Tables reporting statistical results, we report for each contrast the corresponding Bayes Factor across four ranges of standardised effect sizes: negligible (0 to 0.15); small (0.15 to 0.35); medium (0.35 to 0.65) and large (0.65 to 1). In each range, the summary BF10 was obtained as the geometric mean of the Bayes Factor Function values returned for the effect sizes in the range, in steps of 0.01 (geometric mean was used because Bayes Factors are interpreted as ratios)."

5) The authors report to have done pair-wise comparisons of different metrics between different states. They also report that some animals participated in more than one experimental condition. This source of non-independence should be controlled for where it occurs, preferably with multivariate analysis or within-subjects, paired designs. It is unclear if the authors have done so.

Answer: Given the great technical challenge of primate studies (as well as ethical considerations), it is standard practice in the field to obtain multiple data-points from the same animal. Our initial analysis followed our previous published work with these datasets (Tasserie 2022 Science Advances; Signorelli 2021 NeuroImage), which used non-parametric statistics and treated data-points as independent. However, in our revised manuscript we changed our statistical approach to linear mixed effects modelling, whereby animal identity is included as a random effect. To increase transparency for the reader, we also changed how we display the data in our figures. In our revised manuscript, we now use different colours to distinguish the different animals within each dataset, thereby displaying both the variability across animals, and the variability within each animal and each condition.

6) Figures 1 & 2 demonstrate that the depth of processing, as measured by the principal gradient of the first eigenmode of functional connectivity, is reduced by anesthesia and increased with the thalamic DBS. This same logical structure is shared in Figure 4, 5, & 7 and depends on the demonstration of both positive (significant) effects for some conditions, and insignificant effects for others. While this is a fine logic in principle, the authors have committed statistical errors in their current phrasing of and interpretation of their null results. For example, the authors state about Fig 1&2: “The range of the principal gradient of macaque functional connectivity is significantly reduced by anaesthesia, regardless of the specific agent used (Figure 1 and Figure S1), and is significantly increased back to awake levels by low amplitude stimulation of the centro-median thalamus (Figure 2 and Figure S1).” This claim about a “significant increase back to awake levels” is presently supported by the insignificant differences between wake and the low amplitude central stimulation condition demonstrated in the provided Supplement Table 3. It would be more accurate for the authors to report that the results are statistically indistinct from the wake state unless they are willing to add measures that can back the null result. This logical argument is repeated for the results in figures 4, 5, and 7 and should be amended, removed, or supported by more appropriate statistics (see next point).

Answer: We agree that our interpretation of the statistical results had not been sufficiently rigorous, and that the null hypothesis cannot be accepted (only fail to be rejected) in frequentist statistics. We have now updated our analysis and interpretation. In particular, whereas in the Multi-Anaesthesia dataset the hypothesis is that anaesthetics should induce a difference from wakefulness, in the DBS dataset the question is whether different stimulation regimes do or do not induce a change from anaesthesia (the “stimulation-off” condition). Therefore, we now report statistics for the comparison against anaesthesia without DBS. Upon finding a significant difference against the “off” condition induced by DBS, we use Bayes Factors (as recommended in the next point) to perform an additional comparison against wakefulness, and evaluate whether the effect of DBS is simply “in the direction of wakefulness”, or whether there is actual Bayesian evidence supporting the null hypothesis of no difference between DBS and Awake (Figures S3, S13 and S14). Only in such case do we speak of “restoration back to wakefulness levels”.

7) In reference to the point above, comparing the pattern of significant and insignificant effects is further fraught because efforts to avoid type 1 error, which is necessary to interpret results with multiple comparisons, can increase the probability of type 2 error. False discovery rate corrections make it easier to find an insignificant effect, as it influences the alpha of the test. Thus, if a significant effect is present, but underpowered, it is more likely to be deemed insignificant. If the authors wish to make strong claims about the insignificant effects, they should include power analyses for their tests to verify they are sufficiently powered to find effects when they exist. A better solution, however, would be to rely on Bayesian statistics, like the bayes factor, which lend credence to insignificant findings. Most specifically, these should be applied to the comparisons between the wake state and the effective thalamic stimulation conditions to improve their interpretability.

Answer: We are grateful for this excellent suggestion. We have now included Bayes Factors for all our comparisons, using the recently developed Bayes Factor Functions. This enables us to evaluate the statistical support for the alternative hypothesis or for the null hypothesis, as a function of the expected effect size.

8) In figure 8, the authors present the effect sizes (cohen's D) for different tests as a multivariate summary of their results for both the multi-anaesthesia and DBS data sets. While this figure provides a helpful summary of the findings, it does not provide any additional compelling analysis. The current presentation seems to argue for synergistic interpretation of the three metrics, suggesting they capture different elements of the differences between the states. This cannot be verified when comparing the effects from different models. It would be more compelling to see multivariate decoding analyses or multivariate regression results comparing the effect sizes of the three metrics within model on the ability to discriminate conscious states. This will allow the authors to comment on the separability of the metrics, and which metrics contribute more to conscious state discrimination. They can more clearly argue if any of the metrics seem complimentary, or if any are statistically redundant.

Answer: We appreciate this suggestion. We have now replaced Figure 8 in the revised manuscript. Instead, as recommended we performed multivariate analyses to identify the relative contributions of our three markers of brain functional organisation. We used dominance analysis to perform both multivariate classification and multivariate regression, against the arousal scores obtained from behavioural assessment. Of note, although the three metrics appear to provide similar contributions when distinguishing zero from minimal arousal, hierarchical integration appears to be the predictor with greatest relative importance when distinguishing high from low or no arousal.

We have added the following to the Results section of our revised manuscript, in the new subsection "Multivariate brain-behaviour association across anaesthesia and DBS":

"Finally, we bring together our three complementary analyses to generate a multivariate characterisation of the neural effects of anaesthesia and its reversal with thalamic DBS. To this end, we use dominance analysis, which determines the relative contribution of each independent variable to the overall fit (adjusted R^2) of a multiple

linear regression model (62,63), partitioning the total variance accounted for by each predictor. Here, we use as predictors the data pertaining to our three neural markers of interest: range of the principal gradient, hierarchical integration, and harmonic energy. As target variable, we use the arousal score corresponding to each condition, on a scale from 0 to 11 (Uhrig et al., 2016) (Figure S2 and Table S25). For this analysis, we combine data across the Multi-Anaesthesia and DBS datasets.

Our results reveal that the multiple linear regression model accounting for arousal score as a function of gradient range, hierarchical integration, and harmonic energy (Figure 8A) has a total R^2 of 0.44, which is significantly greater than chance ($p < 0.001$, permutation-based; Figure 8B). Among the three markers, over half (56%) of the total variance explained is accounted for by the hierarchical integration, with harmonic energy accounting for 30%, and gradient range for 14% (Figure 8A). Similar results are obtained if instead of considering arousal scores as continuous, we dichotomise them with a cut-off score of 9, thereby separating wakefulness and high-amplitude stimulation of the centro-median thalamus from all other conditions (anaesthesia with different drugs, and less effective stimulations) (Figure S15A). If instead we adopt a more lenient criterion, and set an arousal score cut-off of 3 so that low-amplitude CT stimulation and light anaesthesia are included along with high-amplitude CT and wakefulness (against deep anaesthesia and VT stimulation), then we see a different picture: the order of predictors' relative importance is preserved (hierarchical integration > harmonic energy > gradient range), but the values are nearly equal, with each neural marker accounting for almost exactly one-third of the total variance explained (Figure S15B). Adding Gradient Dispersion and Hierarchical Segregation as predictors provided little difference: Hierarchical Integration and Harmonic Energy remained the main contributors; Gradient Dispersion exhibited greater contribution (15%) than Gradient Range (8%); and Hierarchical Segregation was negligible, with only 3% relative importance (Figure S16).

In other words, our main three neural markers are complementary when it comes to distinguishing the complete absence of arousal from its even minimal restoration via CT stimulation. In contrast, when requiring a more fine-grained characterisation that distinguishes high levels of arousal (which are achieved by high-amplitude but not low-amplitude CT stimulation) from low or no arousal, then the marker of hierarchical integration is the most informative. Overall, this multi-dimensional representation in terms of structural and functional eigenmode reorganisation identifies relevant axes along which perturbations of consciousness can manifest, linking brain organisation in different states and behaviour.

Figure 8. Relating structural and functional eigenmodes of the brain to changes in arousal scores induced by anaesthesia and thalamic deep brain stimulation. (A) Dominance analysis compares all possible models obtained from distinct combinations of predictors, to distribute the variance explained between the predictors, in terms of percentage of relative importance. (B) We establish the statistical significance of our model by comparing the empirical variance explained (R^2) against a null distribution of R^2 obtained from repeating the multiple regression with randomly reassigned arousal scores. ”

9) Again for figure 8, the effect sizes for some measures are quite different between the multi-anaesthesia and DBS datasets, despite the maximum levels representing the distances between the wake state and general anaesthesia. How do the authors explain the substantial differences between the datasets? Further, the current axes make it hard to compare between Panels A and B.

Answer: There are several potential explanations for the differences in effect sizes observed. The Multi-Anaesthesia dataset includes multiple drugs, and the DBS dataset does not use the same animals for wakefulness and anaesthesia, whereas the Multi-anaesthesia dataset does. Additionally, one particularly relevant factor could be the fMRI acquisition, as the Multi-anaesthesia data were collected using a contrast agent (MION, iron mono-particles) to enhance the signal, while all scans under DBS were recorded using the more commonly used BOLD signal. The scanner parameters are also different between the two datasets: the TR is 1.25s in one dataset, but 2.4s in the other, i.e. nearly double. Nevertheless, we wish to point out that our results pertaining to the comparison between wakefulness and deep propofol anaesthesia (the only contrast that was present in both datasets) were overwhelmingly replicated across the two datasets. Also, we have now replaced the original Figure 8 (see previous response).

Discussion:

1) Appropriately, the authors bring up the consistent findings between ketamine and other anesthetics revealed by their analyses. This result is quite interesting for a number of reasons, few of which are addressed in the discussion. As the authors state, Ketamine acts on NMDA channels, and thus had a different mechanism of action from the other anesthetics used. Thus, the similar findings with Ketamine may, as the authors assert, suggest a consistent mechanism for loss of consciousness. However, even at anesthetic doses, the state produced by higher doses of ketamine is quite different potentially from the state produced by other general anesthetics. Ketamine often preserves high frequency activity, especially in frontal cortex, and patients often report vivid and volatile dreams during ketamine anesthesia. Thus, a growing number of scientists believe that ketamine does not render animals or humans “unconscious” at doses used for general anesthesia, but rather “disconnects” them from the external world. It is possible then that the mechanisms described in this paper do not reflect neural correlates of consciousness, but rather, correlates of sensory disconnection. The authors should discuss this alternate interpretation and any evidence that suggests one interpretation over the other. This should especially be contextualized with respect to their working definition of consciousness and the relative conscious state expected in the different anesthetic and stimulation conditions.

Answer: We agree that the interpretation of sensory disconnection is a more conservative interpretation, since sensory disconnection is shared by all three anaesthetics and is necessary (but not sufficient) for loss of consciousness. We have therefore changed our interpretation accordingly:

“Despite the different molecular mechanisms of action, at anaesthetic doses such as the ones used here ketamine induces full loss of behavioural responsiveness, as well as suppression of complex cortical and thalamic responses to deviant stimuli in the local-global auditory paradigm (Uhrig et al, 2016). However, there is evidence that even at anaesthetic doses, ketamine may induce loss of responsiveness not by suppressing consciousness (as propofol and sevoflurane are do) but rather by inducing sensory disconnection from the environment, while potentially preserving subjective experience (Bonhomme et al., 2016 Anesthesiology; Luppi 2023 Science Advances). The preservation of subjective experience despite sensory disconnection from the environment (and consequently, loss of behavioural responsiveness to the environment) is commonly experience during dreaming - and indeed, ketamine-anaesthetised volunteers are known to report vivid hallucinations and dream-like experiences (Bonhomme et al., 2016 Anesthesiology). This phenomenon highlights the fact that behavioural unresponsiveness is an imperfect marker of unconsciousness, since it can also occur as a result of sensory disconnection, or motor impairment, neither of which is the same as unconsciousness (Luppi et al., 2021; Sanders et al., 2012). Since it is not presently possible to obtain subjective reports from the animals involved in our study, a more conservative interpretation of our results is that they pertain to sensory disconnection from the environment, which is a common effect shared by ketamine, propofol, and sevoflurane. Therefore, in the remainder we interpret our results as pertaining to sensory disconnection from the environment (SDE), although we consider it likely that at least the animals anaesthetised with sevoflurane and propofol (which was the drug used for the DBS experiments) were in fact fully unconscious.”

2) The discussion highlights three common weaknesses of the current manuscript, listed as follows: 1 – over reliance on methods from previous manuscripts that are never contextualized or explained to current readers. 2 – inconsistent language about and definitions for consciousness. 3 – imprecise descriptions of results leading to inaccurate or ambiguous conclusions. Consider the following example found in paragraph 7, where the authors state: “low-amplitude stimulation of the central thalamus has weaker effect on behaviour, than high-amplitude stimulation (Tasserie et al., 2022). Thus, gradient range is influenced by CT stimulation even before this stimulation is sufficient to induce restoration of responsiveness”. Here, the authors put forward behavior as an accurate measure of consciousness. This is already a controversial definition that is not consistently applied across the entire manuscript. Further, it is entirely uninterpretable in the given manuscript because, outside of a citation to their previous paper, the authors never indicate how they measured behavior/responsiveness nor how they quantified the differences in behavior between conscious and unconscious states across the two source data sets. At the same time, the descriptions of the effects on behavior are not specific. In neighboring sentences, the authors describe low-amplitude stimulation as failing to restore responsiveness or producing weaker effects than the high-amplitude stimulation. While both may be accurate, the differences between these statements are not trivial and lead to different interpretations. Is the low-amplitude stimulation condition really “unconscious?” Maybe, if the condition actually leads to no behavioral responsiveness. Maybe not there is some responsiveness resorted, but simply weaker than the high-amplitude condition.

Answer: We agree that our previous terminology was insufficiently clear, and we appreciate the opportunity to provide additional nuance.

First, in the revised Introduction we have added a dedicated paragraph for each of our measures of interest, to explain the measure and its rationale to the reader:

“Firstly, we consider the brain’s hierarchical organisation across scales, by studying the principal gradient of functional connectivity (Margulies et al. 2016; Girn et al. 2022), The brain’s intrinsic functional organisation can be represented in terms of continuous, spatially overlapping whole-brain patterns, termed functional gradients (Margulies 2016; vos de Wael 2020). Analogous to the principal components of PCA, gradients map functional connectivity to a low-dimensional space where proximity indicates functional similarity. Each gradient corresponds to a dimension in this space. Along each dimension (represented by an eigenvector), regions will be located closer in space, the more similar they are in terms of their functional connectivity with the rest of the brain. The regions whose FC is most different along that dimension constitute opposite extremes that are maximally functionally different. In particular, the principal functional gradient (eigenvector associated with the principal eigenvalue) captures the direction of maximal spatial variation in functional organisation: its range can be interpreted as reflecting the distance between the two extremes of the cortical processing hierarchy, reflecting the depth of information processing (Girn et al. 2022; Timmermann et al. 2023). This principal gradient can also be reshaped by

pharmacological intervention (Girn et al. 2022; Timmermann et al. 2023), rendering this approach a promising perspective to consider for the present study.”

“Secondly, we study the hierarchical integration of functional signals across scales (Wang et al. 2021). Whereas the range of the principal functional gradient reflects the putative depth of the information-processing hierarchy, hierarchical organisation can also manifest in terms of nested relationships between the system’s parts at different scales: lower elements in the hierarchy are recursively combined to form the higher elements (Hilgetag and Goulas 2020). This perspective makes it possible to consider the interplay of integration and segregation of brain signals across scales - which is a central feature of prominent scientific accounts of consciousness (Dehaene and Changeux 2011). Classical graph-theoretic measures such as small-worldness and modularity quantify integration and segregation at a single scale, making them inadequate to capture these properties across multiple hierarchical modules (Rubinov and Sporns 2010; Newman 2006; Wang et al. 2021). Instead, we can obtain insight into the hierarchical relationships between different scales of distributed activation in the brain by going beyond the principal gradient alone (or first 2-3 gradients), and instead considering all functional eigenmodes, as recently introduced by (Wang et al. 2021). This is achieved by characterising the concordance or discordance of regions’ allegiance across eigenmode scales, corresponding to regions that are jointly activated (same sign) or alternate (opposite sign) at a given scale. This process results in a nested, modular structure that identifies a hierarchical sub-division of the functional connectome into nested modules, up to the level where each module coincides with a single region, indicative of completely segregated activity. Hierarchical integration and segregation can then be quantified by the relative prevalence of the different eigenmodes, indicated by their associated eigenvalues (Wang et al. 2021).”

“Finally, functional brain activity and connectivity unfold over the network of physical white matter pathways between brain regions: the structural connectome. Therefore, for our final investigation we jointly consider brain structure and function, by leveraging the mathematical framework of "harmonic mode decomposition" (Atasoy, Donnelly, and Pearson 2016) to decompose brain activity into distributed patterns of structure-function coupling: the "harmonic modes" of the structural connectome (Atasoy, Donnelly, and Pearson 2016; Atasoy et al. 2017). Through this decomposition, we quantify the extent to which brain activity is constrained by the underlying network of structural connectivity. Each harmonic mode is a cortex-spanning activation pattern (eigenmode of the structural connectome) characterised by a specific granularity. The use of connectome-specific harmonic decomposition of cortical activity allows us to quantify the contribution of structural organisation to brain activity, across different spatial granularities scales: from large-scale to fine-grained. This approach therefore goes beyond previous investigations that assessed the similarity of structural and functional connectivity at a single scale (Barttfeld et al., 2015; Uhrig et al., 2018; Demertzi et al., 2019; Gutierrez-Barragan et al., 2021). Additionally, this approach is of particular interest because recent results in humans have shown that distributed harmonic patterns of structure-function dependence relate with human consciousness (Luppi et al. 2023), so we aim to investigate here the cross-species generalisation of these results, and their potential susceptibility to thalamic stimulation.”

Additionally, we have now clarified that our proxy measure for consciousness is the behavioural assessment of arousal, as is typical in the animal literature, while also acknowledging its imperfect nature:

“Since non-human primates cannot use language to report their subjective experience, we rely on behavioural measures of arousal as our proxy for consciousness, as is common practice when working with animal models of anaesthesia, and also widely used in clinical practice. Nonetheless, we acknowledge that unresponsiveness is not equal with unconsciousness.”

We have also taken additional care to distinguish reference to unconsciousness versus sensory disconnectedness from the environment:

“However, there is evidence that even at anaesthetic doses, ketamine may induce loss of responsiveness not by suppressing consciousness (as propofol and sevoflurane are do) but rather by inducing sensory disconnection from the environment, while potentially preserving subjective experience (Bonhomme et al., 2016 Anesthesiology). The preservation of subjective experience despite sensory disconnection from the environment (and consequently, loss of behavioural responsiveness to the environment) is commonly experience during dreaming - and indeed, ketamine-anaesthetised volunteers are known to report vivid hallucinations and dream-like experiences (Bonhomme et al., 2016 Anesthesiology). This phenomenon highlights the fact that behavioural unresponsiveness is an imperfect marker of unconsciousness, since it can also occur as a result of sensory disconnection, or motor impairment, neither of which is the same as unconsciousness (Luppi et al., 2021; Sanders et al., 2012). Since it is not presently possible to obtain subjective reports from the animals involved in our study, a more conservative interpretation of our results is that they pertain to sensory disconnection from the environment, which is a common effect shared by ketamine, propofol, and sevoflurane. Therefore, in the remainder we interpret our results as pertaining to sensory disconnection from the environment (SDE), although we consider it likely that at least the animals anaesthetised with sevoflurane and propofol (which was the drug used for the DBS experiments) were in fact fully unconscious.”

To make the manuscript self-contained, and increase transparency for the reader, we now provide Figure S2 and Table S25 to report the effects of anaesthesia and DBS on arousal, quantified using the same 11-point behavioural scale across both datasets. We also report the criteria and related outcomes (including explanation of CT 3V versus 5V DBS) in the dedicated section of the Methods, “Behavioural assessment of arousal”:

“We used a preclinical behavioural scale adapted from Uhrig et al. 2016 to assess the arousal levels of the monkeys. This scale, based on the Human Observers Assessment of Alertness and Sedation Scale (Chernik et al., 1990) and previously utilised in non-human primate (NHP) research (Vincent et al., 2007), was used consistently across all experimental conditions, in both datasets. The arousal testing occurred outside the MRI environment and was conducted at the beginning and end

of each scanning session, for each condition, once the animals were no longer under paralysis.

The assessment encompassed six criteria as follows:

- exploration of the surrounding world, from 0 to 2:
 - 0 = total absence,
 - 1 = small search of external clue,
 - 2 = total investigation of the environment (such as head orientation to a sound);
- spontaneous movements, from 0 to 2:
 - 0 = total absence,
 - 1 = small torso and/or limb movement,
 - 2 = large torso and/or limb movement
- shaking / prodding, from 0 to 2:
 - 0 = total absence,
 - 1 = small body movement,
 - 2 = large body movement;
- toe pinch, from 0 to 2:
 - 0 = total absence,
 - 1 = small reflex (weak body movement or eye blinking or cardiac rate change),
 - 2 = clear reaction (strong body movement and eye blinking or eye opening and cardiac rate change);
- eyes opening, from 0 to 2:
 - 0 = total absence,
 - 1 = small blinks or eye movements,
 - 2 = full eye opening;
- corneal reflex, from 0 to 1:
 - 0 = absent,
 - 1 = present.

The behavioural score ranged from 0 to 11, where 11 represented the maximum note achievable and 0 the lowest.

In all cases, and for both datasets, we observed no differences in arousal scores between different animals in the same condition. For both datasets, the behavioural score during wakefulness was the maximum of 11/11 for all the animals (Monkey A, Monkey K, and Monkey J from the Multi-Anaesthesia dataset, and Monkeys B, J and Y from the DBS dataset): exploration of the surrounding world = 2; spontaneous movements = 2; shaking/prodding = 2; toe pinch = 2; eyes opening = 2; corneal reflex = 1.

For results pertaining to the different anaesthesia conditions of the Multi-Anaesthesia dataset, see Table S25.

In the anaesthesia without DBS ("off") condition, monkeys N and T displayed the minimum behavioural score of 0 over 11, same as the deep anaesthesia from the Multi-Anaesthesia dataset: exploration of the surrounding world = 0; spontaneous

movements = 0; shaking/prodding = 0; toe pinch = 0; eyes opening = 0; corneal reflex = 0.

For anaesthetised macaques under CT DBS at low amplitude (3V), we measured a clinical score of 3 over 11 (exploration of the surrounding world = 0; spontaneous movements = 0; shaking/prodding = 0; toe pinch = 1, eyes opening = 1; corneal reflex = 1).

When the CT electrical stimulation amplitude was increased to 5V (high-amplitude CT DBS), animals reached a total score of 9 over 11 (exploration of the surrounding world = 1; spontaneous movements = 1; shaking/prodding = 2; toe pinch = 2; eyes opening = 2; corneal reflex = 1).

For VT DBS, both low (3V) and high (5V) amplitude stimulation led to a clinical score of 0, identical to what is observed in the absence of any stimulation. See also Figure S2.”

We also added, in the Introduction, a description of the main results of Tasserie et al (2022), to familiarise the reader:

“We leverage the experimental accessibility of animal models, analysing a unique recently published dataset of macaque resting-state fMRI acquired under deep propofol anaesthesia with and without direct intervention on the thalamus, in the form of deep brain stimulation (DBS) of the centro-median (CT) and ventral-lateral thalamus (VT) (Tasserie et al., 2022). In this dataset, it was shown that high-amplitude CT stimulation restores behavioural markers of arousal (eye opening, movement, reflexes and exploratory behaviour) that were suppressed by anaesthesia, as well as restoring the complexity of EEG signals, and neural processing of auditory stimuli beyond sensory cortices, both of which have been argued to be signatures of consciousness (Casali 2021; Dehaene 2011; Mashour 2020). This effect was greatly diminished at lower amplitude DBS, and entirely absent upon stimulation of a different thalamic nucleus, demonstrating an exquisite level of spatial specificity.”

We also added direct analyses relating our neural markers to these quantifications of arousal (Figure 8 and Figures S15, S16).

3) Overall, the authors must provide stronger evidence specifically linking their measures of structural/functional connectivity and integration to consciousness. In the original paper, Tasserie 2022, the authors provide evidence relating DBS to consciousness, including arousal scores, entropy, and results from the local/global paradigm. Given the inevitable variability in the level of consciousness for individual stimulations across these scores, I expected the authors to numerically demonstrate the relationship between individual trial measures of gradient range, hierarchical integration, and harmonic energy and behavioral arousal or conscious experience. How do the authors justify this omission when it is critical to drawing the desired link between their measures and consciousness more specifically?

Answer: As requested, in our revised manuscript we now numerically demonstrate the relationship between individual trial measures of gradient range, hierarchical integration, harmonic energy, and behavioural arousal (which were available on the same scale for both datasets), through the new Dominance analysis approach (Figure 8 and Figures S15, S16). Please see the response to Q8 pertaining to Results, and the new Results subsection on “Multivariate brain-behaviour association across anaesthesia and DBS”.

Figure 8. Relating structural and functional eigenmodes of the brain to changes in arousal scores induced by anaesthesia and thalamic deep brain stimulation. (A) Dominance analysis compares all possible models obtained from distinct combinations of predictors, to distribute the variance explained between the predictors, in terms of percentage of relative importance. (B) We establish the statistical significance of our model by comparing the empirical variance explained (R^2) against a null distribution of R^2 obtained from repeating the multiple regression with randomly reassigned arousal scores.

4) Related to the above point, the authors frame their discussion around mechanisms of consciousness, but none of their manipulations selectively influence consciousness. General anesthetics influence neural mechanisms that exceed the scope of consciousness (providing analgesia and suppressing memory for example). Central thalamic DBS may selectively reinstate consciousness, but it is also possible that it drives arousal and other mechanisms function to reinstate consciousness. If the presence or lack of behavioral responsiveness is the primary measure of consciousness used in the paper, this only further increases the likelihood that the results the authors present here are more strongly linked to sensory connectedness than to consciousness per se. This should either be discussed or refuted if possible.

Answer: In accordance with this and previous comments, we have reframed our Discussion and interpretation to clarify that our proxy for consciousness is the behavioural assessment of arousal. It should be noted that our study relies on resting-state fMRI in the absence of any active behavioural task to probe consciousness. However, the study of brain dynamics at rest across states of consciousness could reveal that structure-function correlation reflects not only the arousal level of consciousness, but also takes in account the awareness level. In fact,

Demertzi et al. (Demertzi Sci Adv 2019) measured rs-fMRI in patients who were diagnosed in a vegetative state/unresponsive wakefulness syndrome (UWS) or in a minimally conscious state (MCS) with repetitive standardised behavioural assessments and who were scanned under a resting condition, sedation free, or under anaesthesia with propofol. The brains of unresponsive patients predominantly exhibit patterns of low inter-regional phase coherence, primarily mediated by structural connections, and have a lower chance of switching between patterns. This complex pattern was further demonstrated in patients with covert perception who were able to perform mental neuroimaging tasks, confirming the impact of this pattern on consciousness. Anaesthesia increased the probability of less complex patterns to the same level and confirmed its effect on consciousness. Their results demonstrate that consciousness relies on the brain's ability to maintain rich brain dynamics, thus generalising the consciousness signature that we previously described in non-human primates (Barttfeld 2015, Uhrig 2018).

We also acknowledge that general anaesthetics exert a wide range of effects on the brain. However, this is precisely why considering different anaesthetics is advantageous, and an asset of the present study: the more we can find neural effects that are consistent across different anaesthetics, especially with different molecular mechanisms of action, the more we can be confident that such consistent neural effects pertain to the common effect of anaesthetics, rather than to idiosyncratic side-effects of any single one of them. Because of this point, and since ketamine is not guaranteed to suppress subjective experience, we have now clarified that our results are most accurately interpreted in terms of sensory disconnectedness from the environment, rather than full-blown unconsciousness (though the latter may also have occurred, at least for the other anaesthetics). We now clarify this in the revised Discussion:

“Despite the different molecular mechanisms of action, at anaesthetic doses such as the ones used here ketamine induces full loss of behavioural responsiveness, as well as suppression of complex cortical and thalamic responses to deviant stimuli in the local-global auditory paradigm (Uhrig et al., 2016). However, there is evidence that even at anaesthetic doses, ketamine may induce loss of responsiveness not by suppressing consciousness (as propofol and sevoflurane are do) but rather by inducing sensory disconnection from the environment, while potentially preserving subjective experience (Bonhomme et al., 2016 Anesthesiology). The preservation of subjective experience despite sensory disconnection from the environment (and consequently, loss of behavioural responsiveness to the environment) is commonly experience during dreaming - and indeed, ketamine-anaesthetised volunteers are known to report vivid hallucinations and dream-like experiences (Bonhomme et al., 2016 Anesthesiology). This phenomenon highlights the fact that behavioural unresponsiveness is an imperfect marker of unconsciousness, since it can also occur as a result of sensory disconnection, or motor impairment, neither of which is the same as unconsciousness (Luppi et al., 2021; Sanders et al., 2012). Since it is not presently possible to obtain subjective reports from the animals involved in our study, a more conservative interpretation of our results is that they pertain to sensory disconnection from the environment, which is a common effect shared by ketamine, propofol, and sevoflurane. Therefore, in the remainder we interpret our results as pertaining to sensory disconnection from the environment (SDE), although we consider it likely that

at least the animals anaesthetised with sevoflurane and propofol (which was the drug used for the DBS experiments) were in fact fully unconscious.”

Similarly, the combination of different stimulation sites in our DBS dataset shows that what matters is not merely the injection of current, but first and foremost its specific location, with amplitude also playing an important role. Since this occurred while anaesthesia was ongoing, it also allows us to dissociate “having an amount of propofol sufficient to induce sensory disconnection from the environment” from “being in a state of sensory disconnection”, since the latter can be reversed when the former is present. We have also clarified this in the revised Discussion:

“Crucially, the restoration of such distributed cortical patterns can be triggered by selective stimulation of a specific subcortical region, as demonstrated by our spatially specific intervention: the centro-median thalamic nucleus - in contrast to the much weaker effects elicited by control site stimulation of the ventral lateral nucleus of the thalamus. This is of particular relevance because being compromised upon sensory disconnection from the environment is a necessary feature for a signature of consciousness, but it is not sufficient. A more stringent requirement is that the neural signature should also be restored when sensory connectedness is restored. Although (Luppi et al 2023 Comms Biol) did show that harmonic energy is restored upon post-anaesthetic recovery, they used spontaneous recovery after anaesthetic discontinuation and therefore could not dissociate the presence of propofol in the bloodstream (and the various consciousness-unrelated effects that it may have on the brain) from the presence of environmental connectedness - only show that both co-varied with the harmonic energy. Our DBS results do achieve such a dissociation, both with harmonic energy and with other markers, by restoring both distributed brain function and sensory connectedness despite continuous anaesthetic presence. ”

5) Paragraph 7 importantly describes the different effects across measures, starting first with the effect sizes depicted in figure 8. While it does seem clear that there are effect size differences, these are highly variable across the two datasets. This should be discussed.

Answer: Following our response to Q9 about the Results, we have added the following text to the Discussion of our revised manuscript:

“Direct comparisons between the neural data obtained in the two datasets are not straightforward: the Multi-Anaesthesia dataset used MION as a contrast agent to improve signal, whereas the DBS used the more commonly used BOLD signal. Additionally, the acquisition parameters were different (e.g., 2.4s vs 1.25s TR), and the DBS dataset did not compare anaesthesia and wakefulness in the same animals, as did the Multi-Anaesthesia dataset. Despite these differences, however, significant differences in the same direction were found for the deep propofol condition of the Multi-Anaesthesia dataset, and the propofol anaesthesia condition without stimulation (“off”) of the DBS dataset, which is the only contrast that was present in both datasets.

This replication demonstrates the robustness of our markers to acquisition differences, in addition to their generalisation across anaesthetics and across species.”

6) The authors note in paragraph 7 a somewhat puzzling finding that the gradient range results were maximally influenced by low amplitude central thalamic stimulation. This warrants additional discussion. If gradient range is a correlate of consciousness, why is the effect so weak for the high-amplitude stimulation condition. Why might low-amplitude stimulation produce this result?

Answer: We agree that this finding does not have a straightforward interpretation. It indicates that the relationship between DBS-induced CT activity (and therefore arousal) and the range of the principal gradient is nonlinear, since greater range is observed for lower stimulation level. Ultimately, such nonlinearities are to be expected in the context of a complex system such as the primate brain. Additionally, note that 3V CT stimulation is not inert, instead restoring eye-opening, corneal reflex, and some exploration of the environment. Indeed, our dominance analysis classification shows that gradient range is nearly as good as the best performing feature when discriminating between the complete absence of arousal, or any non-zero levels (Figure S15B), whereas it is less suitable to distinguish high from low or no arousal (Figure S15A).

We have added the following text to address this point in our revised Discussion:

“Upon considering together the results of gradient analysis and hierarchical integration, it appears that the effects of low-amplitude CT stimulation primarily reflect on the principal functional gradient(s), whereas at high-amplitude the effect is less restricted to the principal gradients only, and instead exhibits a broader reach, concomitant with greater restoration of sensory connectedness to the environment. These observations reinforce the value of considering distributed brain function across multiple scales: the full effects of high-amplitude stimulation appear to be spread across multiple functional eigenmodes, such that only considering the first one provides an incomplete picture with weaker link to behaviour, whereas greater insight is obtained by considering all, and by combining both functional and structural eigenmodes.”

7) Figure 8 was quite interesting and did not receive much discussion. I’m curious about the degree to which the authors invite comparisons between the multi-anesthesia data set and the DBS dataset, especially since they are presented on different scales. Superficially, it seems based on the effect size measures provided that the CT high condition had many similarities to the light-sevo condition in terms of net differences from wakefulness. Is it reasonable then to assume there is some behavioral match between the conditions?

Answer: Pertaining to behavioural similarities between light sevo and high-amplitude CT DBS, we have now added arousal scores in Figure S2 and Table S25, which show that light sevoflurane induces an arousal score that is most similar to low-amplitude CT DBS (rather than high-amplitude CT DBS). However, the above-mentioned differences between datasets and between acquisition parameters (see response to Q5) make direct comparisons of neural

markers difficult. We have therefore removed the original Figure 8 as potentially misleading. Instead, we have now replaced the initial Figure 8 with a more formal quantification of the relative contributions of our measures of brain organisation to predicting behavioural arousal, as provided by dominance analysis. Please see the response to Q8 pertaining to Results, and the new Results subsection on “Multivariate brain-behaviour association across anaesthesia and DBS”.

8) Most of the paper’s findings are largely expected. As the authors note, other papers have demonstrated the interplay between structural and functional connectivity varies across conscious states. Many papers have shown that indices of neural communication and complexity, both in specific pathways and larger-scale networks, are altered across conscious states. Other papers have even demonstrated that different measures of functional connectivity are consistently reduced across low-conscious states and reinstated by central thalamic DBS in macaque monkeys. It is thus imperative that the authors take greater care in outlining the conceptual leap of this study over others in recent literature. Rather than simply focusing on their design, which other studies have used, they should focus on the interpretability of their findings and how it lends new ideas to the study of consciousness. I recommend including a paragraph discussing other measures of consciousness and how they compare conceptually and functionally to the current study, as well as the advantages of the current measure compares to what others have used.

Answer: We appreciate the need to be clearer about the advances and scientific insights provided by our paper, and we welcome the opportunity to do so. Although we believe that it is reasonable to expect that recent results obtained in humans should generalise to the non-human primate brain, we would also argue that this expectation is by no means a guarantee, but rather a scientific hypothesis to be tested empirically - and indeed, testing this hypothesis was one of the key goals of the present manuscript (as we now clarify in the revised Introduction):

“Therefore, the first goal of the present investigation is to establish whether these distributed markers of consciousness that have been recently identified in the human brain, can be generalised to the non-human primate brain, and whether they behave consistently across anaesthetics with distinct molecular mechanisms.”

Note that the markers referred to here, harmonic energy and gradient range, were only published this year (Luppi et al., 2023; Huang et al., 2023), and only in humans. To our knowledge, the marker of hierarchical integration has not been applied to consciousness at all, outside of the present manuscript.

Similarly, the fact that some other markers of consciousness were previously shown to be restored by thalamic stimulation, does not guarantee that different markers also will. None of the markers considered here were investigated by the Redinbaugh, Bastos, or Tasserie papers. In fact, Bastos and Redinbaugh did not use fMRI at all. This was also a scientific hypothesis that we set out to test empirically, and we also make this clearer in the revised Introduction:

“Therefore, our second goal is to determine whether the reorganisation of distributed brain function induced by anaesthesia can be reversed by targeted electrical stimulation of the central thalamus, concomitantly with restoration of behavioural evidence of consciousness.”

Indeed, we found that some of our markers were more sensitive than others, both in terms of detecting significant differences between conditions, and in terms of providing greater predictive ability with respect to trial-level arousal markers in our new dominance analysis (Figure 8 and Figures S15, S16).

Overall, there are multiple ways in which our results go beyond the existing literature. First, we generalise recent human findings (harmonic energy, gradient range) to non-human primates. These markers were only published in 2023 and had not been shown in species other than human. They were not investigated by Tasserie, Bastos, or Redinbaugh. Second, we show that these markers generalise across different anaesthetics, including with very different molecular mechanisms of action. Third, we show that these markers are restored when connectedness to the environment is restored by stimulation, despite continuous anaesthesia. Although Luppi 2023 *Comms Biol* did show that harmonic energy is restored upon post-anaesthetic recovery, they used spontaneous recovery and therefore could not dissociate the presence of propofol in the bloodstream from the presence of environmental connectedness - only show that both co-varied with the harmonic energy. Our DBS results do achieve such a dissociation, demonstrating that harmonic energy (as well as the other markers) track connectedness rather than the mere presence of anaesthetic. Additionally, the DBS results show that although these markers pertain to the functioning of the cortex as a whole, they can all be influenced by the state of a specific subregion of the thalamus, thereby demonstrating highly region-specific control of the whole brain state. Thus, the markers that we consider here are not specific to humans, they are not specific to a particular drug, they track connectedness rather than drug concentration, and they are under the influence of a specific subcortical sub-region. We would argue that each of these insights on its own would be a valuable addition to our neuroscientific understanding. None of these insights were available prior to this work.

We now clarify this in our revised Discussion:

“We also note that in our multivariate dominance analysis, both approaches that considered all eigenmodes (hierarchical integration, which considers all functional eigenmodes; and harmonic mode decomposition, which considers all structural eigenmodes) exhibited greater relative importance for predicting behavioural arousal, than consideration of the principal gradient (principal functional eigenmode, or dispersion of the first three) alone. [...] These observations reinforce the value of considering distributed brain function across multiple scales: the full effects of high-amplitude stimulation appear to be spread across multiple functional eigenmodes, such that only considering the first one provides an incomplete picture with weaker link to behaviour, whereas greater insight is obtained by considering all, and by combining both functional and structural eigenmodes.”

And also:

“Overall, we showed that distributed patterns of functional activity and connectivity of the primate brain co-vary with sensory connectedness to the environment (and possibly consciousness), its suppression by different anaesthetics, and its restoration by subregion-specific thalamic stimulation. The resulting insights align both with the well-known increase in structure-function coupling observed across a variety of pharmacological and pathological states of unconsciousness across species, and also with prominent theories of consciousness that postulate a central role for integrative processes, which we show here to be reliably disrupted in the primate brain upon anaesthetic-induced sensory disconnection from the environment. Thus, the present work provides several advances. First, we generalise recent findings about the effects of anaesthesia on the human brain (connectome harmonics, gradient range) to non-human primates. Second, we show that these markers generalise across different anaesthetics, including with very different molecular mechanisms of action. Third, we show that these markers of distributed brain function are restored when sensory connectedness to the environment is restored by electrical stimulation of a specific thalamic sub-region, despite continuous anaesthesia, thereby dissociating them from the consciousness-unrelated effects of anaesthetics. Additionally, the DBS results show that although these markers pertain to the distributed functioning of the cortex, they can be influenced by the state of a specific subregion of the thalamus, thereby relating localised and distributed brain function. In this sense, our results help reconcile the traditional locationist approach to the neural correlates of consciousness with recent advances in understanding brain function in terms of distributed patterns. Thus, the neural markers that we consider here are not specific to humans, they are not specific to a particular drug, they track connectedness with the environment even in the presence of continuous anaesthetic administration, and they are under the influence of a specific thalamic sub-region.”

We have also added the following text in the Discussion, addressing additional measures of consciousness that were investigated in the original analysis of our DBS dataset, and in previous reports of awakening induced by thalamic stimulation:

“The original analysis of our DBS dataset indicated that high-amplitude CT stimulation also restores long-range functional correlations and processing of auditory stimuli beyond sensory cortex, as well as re-increasing the dynamic decoupling between structural and functional connectivity that had been suppressed by anaesthesia, and re-increasing the complexity of EEG signals (Tasserie et al., 2022). Earlier studies had already reported that electrical stimulation of specific central-lateral thalamic nuclei (but not control sites) can restore arousal and putative electrophysiological markers of consciousness during anaesthetic-induced loss of responsiveness in macaques (Bastos et al. 2021; Redinbaugh et al. 2020; Tasserie et al. 2022), counteracting the loss of high-frequency (gamma-band) activity and communication between the thalamus and deep cortical layers, which is induced by anaesthesia.”

In the Discussion of our revised manuscript, we now also address additional prominent markers of consciousness that are of particular relevance for the present study:

“Of particular relevance, a previous study in humans indicated that propofol-induced loss of behavioural responsiveness (LOBR) does not coincide with loss of brain

responsiveness: noxious stimuli still elicited brain responses after volunteers had ceased overtly responding (Warnaby 2013). Loss of brain responsiveness (assessed with fMRI) occurred at higher propofol doses, and coincided with a plateau of EEG slow waves, termed “slow wave activity saturation” (SWAS). It is tempting to speculate that the period between LOBR and SWAS, characterised by isolation of the thalamocortical system from sensory stimuli, may correspond to the situation of sensory disconnection despite preserved subjective experience, prior to full unconsciousness at SWAS, and possibly analogous to the effect of ketamine anaesthesia. Given the central role of the thalamus in this phenomenon, it would be of great interest to determine whether SWAS is also observed in the macaque, whether it coincides with disruption of distributed brain function as assessed here, and whether it is countered by high- or low-amplitude CT stimulation. More broadly, it will be of great interest to assess whether CT DBS restores not only the prevalence of high-gamma oscillations and the complexity of spontaneous EEG signals (as already demonstrated), but also the complexity of EEG signals after perturbation with Transcranial Magnetic Stimulation pulses: this “Perturbational Complexity Index”, inspired by Integrated Information Theory’s emphasis on the need for both complexity and integration of brain signals in order to support consciousness (Tononi et al., 1994, 1998, 2016) represents one of the most accurate markers of consciousness currently available in humans, sensitive to both the severity of disorders of consciousness and the type and depth of anaesthesia (Casali 2013; Sarasso 2021). Does CT activity favour complex propagation of cortical perturbations? Of note, a recent mouse model indicated that under anaesthesia, thalamic stimulation (in the ventral posteromedial thalamic nucleus) elicited a more complex PCI response than stimulation of motor cortex (Casas-Torremocha 2023). Bringing together the diversity of existing markers of consciousness into a unified framework represents an ongoing challenge, and will be essential for translational applications to the clinic.”

We hope that these additional discussions, as well as the clarifications of our measures of interest in the Introduction, will provide the reader with the required context to situate the current study in the rapidly-evolving literature on the neuroscience of consciousness.

9) The authors use what they describe as a “local” activation of the central thalamus to produce a distributed effect on cortical integration. This might be expected given the recent evidence linking the central thalamus to consciousness, but also the historic context of central thalamic nuclei as being anatomically “nonspecific”, with broad projections to other brain areas. Specifically the authors have targeted CM, with predominant projections to basal ganglia and sparser connections to cortex. Their targeted region is also very close to CL, with strong projections to frontal and parietal cortex. The authors should use some space in the discussion to address the anatomy of the thalamus and explore why central thalamus, and not VT can influence functional connectivity at different scales in a way that benefits consciousness.

Answer: We thank the reviewer for giving us the opportunity to elaborate on this important point: we have updated and expanded the Discussion accordingly.

“The central thalamus (CT) consists of several intra and paralaminar thalamic nuclei that act as intermediaries between the brainstem/basal forebrain arousal systems and the cortex. These central thalamic neurons play a pivotal role in regulating arousal by establishing connections with extensive cortical networks⁷⁸. Intralaminar thalamic (ILT) nuclei, also discriminated by ‘matrix’ cells (in opposition to ‘core’) serve dual functions, with orthodromic connections (exchanges from deep structures) and antidromic functions (strong calbindin protein staining, broadcasting signals to the supra-granular layer I of the cortex)^{79–81}. ILT nuclei receive afferent connections from the Mesencephalic Reticular Formation, which is part of the Ascending Reticular Activating System^{82,83}. They also receive inputs from the brainstem, superior colliculus, PedunculoPontine Tegmentum and basal ganglia^{34,84}. Efferent ILT pathways extend to the Striatum (comprising the Caudate Nucleus and Putamen)^{85,86}. At the cortical level, CT projects to the medial prefrontal, anterior cingulate and somatosensory cortex, as well as primary and supplementary motor areas, frontal eye field, associative regions and nucleus accumbens through well-established structural pathways^{34,87}.”

In the current study, DBS electrodes targeted the centro-median thalamic nucleus or the the ventrolateral thalamus as a control site. However, as opposed to microstimulation (like used by Redinbaugh and colleagues), DBS modulation extended beyond CM neurons and broadcasted to a larger group of CT neurons. Thus, thalamic DBS achieves bicortical input via stimulation of corticothalamic axons (retrograde mechanism) and stimulation of thalamocortical axons (anterograde mechanism). This is supported by recent evidence that in another non-human primate (Marmoset), the central thalamus projects to higher-order frontal, cingulate, and posterior parietal regions⁸⁷. Indeed, as described in Tasserie et al²¹, fMRI maps provided evidence for the strong cortical modulation effects of our DBS protocol. Specifically, although high VL-DBS could activate a fronto-parietal network, no modulation effect on the cingulate cortex activity was evident, whereas CT-DBS activated a fronto-parieto-cingular network, consistent with CT anatomical projection. Only CT-DBS elicited robust modulation of the striatum activity, probably due to axonal projections between CT and the striatum^{87,88}. Restoration of the behavioural arousal and neural responsiveness to stimuli was observed only during high CT-DBS but never occurred in the other experimental conditions (low CM-DBS, low VL-DBS and high VL-DBS), suggesting that it may be key to stimulate among the nuclei included in the volume of activated tissue modelled for high CT-DBS²¹. Other groups previously applied optogenetic in rodent models of consciousness loss and could demonstrate a direct link between the modulation of CT neurons and the transition in the state of consciousness^{75,89}. In particular, our findings are consistent with previous studies^{30,31,33} since modelling the volume of activated tissue²¹ indicated that our electrical stimulation encompassed the same targets as these groups, including intra-laminar central thalamic nuclei, as described by Schiff⁷⁸.”

We also mention the connectivity of thalamus with PFC, and evidence from rodent studies of the latter’s capacity to also modulate arousal upon stimulation or inactivation:

“Of note, the evidence of anatomical connectivity between central thalamus and PFC⁸⁷ means that our results are also consistent with rodent evidence that PFC stimulation

via carbachol administration awakens rats from sevoflurane anaesthesia⁹⁴, whereas tetrodotoxin-mediated inactivation of rat medial PFC had the opposite effect, delaying recovery⁹⁵. Likewise, PFC inactivation also diminishes the ability of basal forebrain stimulation to promote arousal from sevoflurane anaesthesia⁹⁶. Although such studies are yet to be translated to humans or non-human primates, they suggest the possibility that PFC stimulation may also represent an avenue of achieving similar effects as CT stimulation, both thanks to direct connections between the two, as well as via striatum-mediated disinhibition of the thalamus via descending arousal pathways⁹⁷.”

Minor Comments:

1. The authors should review the manuscript carefully to ensure that all relevant citations have been provided correctly as intended throughout the manuscript. As demonstrated by the following examples, citations sometimes seemed to be omitted or ill-matched to the statements being presented

a. It is unclear why the authors include Suzuki and Larkum, 2020 as their primary citations for the following statement: “In this sense, our results help reconcile the traditional locationist approach to the neural correlates of consciousness with recent advances in understanding brain function in terms of distributed patterns (Suzuki and Larkum, 2020).” This is an exceptional paper, but from this reviewer’s recollection, does not seem to match the point being made. If this is the correct citation, more context would be of use.

Answer: We thank the reviewer for spotting this inaccuracy, and we have removed the reference in question.

b. In the discussion, the authors may have provided the wrong list of citations. The authors write: “and the effects of selective thalamic stimulation on consciousness and arousal in animals (Alkire et al., 2007, 2009; Lewis et al., 2015; Bastos et al., 2021; Tasserie et al., 2022) and human patients (Tononi, 2004; Schiff et al., 2007; Staunton, 2008; Mashour et al., 2020). Indeed, recent studies also reported that electrical stimulation of specific central-lateral thalamic nuclei (but not control sites) can restore arousal during anaesthetic-induced loss of responsiveness in macaques (Tononi, 2004; Schiff et al., 2007; Staunton, 2008; Mhuirheartaigh et al., 2010; Liudyno et al., 2013; Ní Mhuirheartaigh et al., 2013; Vijayan et al., 2013; Akeju et al., 2014; Flores et al., 2017; Hemmings et al., 2019; Kelz et al., 2019; Mashour et al., 2020; Redinbaugh et al., 2020, 2022; Afrasiabi et al., 2021; Bastos et al., 2021; Gammel et al., 2023; Kantonen et al., 2023), counteracting the loss of high-frequency (gamma-band) activity and communication between the thalamus and deep cortical layers, which is induced by anaesthesia.” Based on the statements, it seems like they intend to cite Redinbaugh, Neuron 2020, and not the large list of papers included, which instead seem identical to a larger list earlier in the paper broadly citing that the thalamus may play a role in consciousness. If this was not the intention, it seems inappropriate to cite many disparate papers for such specific results. It is also unclear why Redinbaugh 2020 has been omitted from the earlier point concerning evidence of thalamic stimulation on consciousness and arousal in animals if this is not the intention.

Answer: We thank the reviewer for spotting this error. In the new version of the manuscript, we have eliminated the unintended references and included Redinbaugh et al., 2020 in the list of articles for reference in the sentence “the effects of selective thalamic stimulation on consciousness and arousal in animals”.

REVIEWER COMMENTS

Reviewer #1 (Remarks to the Author):

The authors have addressed all of my comments. I have no further questions.

Reviewer #2 (Remarks to the Author):

The manuscript makes excellent use of two valuable fMRI macaque datasets to demonstrate global neural signatures associated with conscious states. The authors have addressed many of the concerns I had with the previous version of the manuscript, especially with respect to their statistical analyses and the strength of their claims which now are a better match to the expectations of the field. The current version is greatly improved, providing insight about the neural mechanisms of consciousness likely to be of considerable interest to neuroscientists and clinicians alike. Following some minor revisions, suggested below, I would recommend this paper for publication.

Results:

1. The authors have introduced a new analyses involving multivariate linear regression of arousal scores simultaneously against their measures of consciousness (Gradient Range, Hierarchical integration, and Harmonic Energy). This is marked improvement on the previous figure and much more informative. However, in addition to the reported dominance analyses, the authors should provide more results from this model to bolster the impact of figure 8 and their claims. In general, I would like to see figures displaying the regression slopes and the fitted trend to the raw data. In the current version of the manuscript, the authors have not demonstrated that their arousal score operates linearly, nor that the expected effects of their main predictors should have a strictly linear relationship to arousal. While it is possible to guess what the regressions would look like based on the data present in figures 1B, 2B, 4, & 7, the regressions and their relevant statistics should still be supplied so readers can gauge the strength of association and shape of association to the arousal scores proper.

2. Displaying the results of the dominance analyses in bar graphs seems like a strange

choice, specifically because the % relative importance is not independent between model predictors, but rather cumulative to 100%. Instead, it would be more appropriate to present the data as a pie chart. The same should apply to the supplemental figures.

Discussion:

1. I approve of the Authors more conservative decision, in light of the ketamine data, to discuss their results with respect to sensory disconnection from the environment (SDE) as opposed to consciousness per se. I also agree that they are likely correct that full unconsciousness (lack of experience) likely occurred at some point for the macaques under propofol anesthesia, though this cannot easily be confirmed. However, the bulk of the argument, as outlined in paragraph 4 of the discussion is that ketamine cannot be linked to true unconsciousness because of the high degree of vivid dreaming reported after ketamine anesthesia, but that sevoflurane and propofol probably do produce true unconsciousness. A number of studies have now shown evidence of dreaming under different kinds of anesthesia, but the current presentation of the argument is a little messy. It is likely more accurate to say that propofol and sevoflurane likely produce epochs of true unconsciousness, while it is unclear if ketamine does.

2. The authors present ketamine as an NMDA antagonist. Indeed, this is one primary mechanism of the drug, but I believe it has a number of other less-specific targets especially at anesthetic doses, which produce unresponsiveness. This is important to discuss in paragraph 5 as the authors note differences in the direction of effects at lower and higher doses of the drug. This could be that NMDA channels may mediate psychedelic effects, but that sensory disconnection relies more on the non NMDA targets.

Minor Concerns:

1. There is a typo in the labels of Figure 1. The term Gradient is misspelled on one of the axes.

2. There may be another incorrect citation. In the discussion, the authors write "Another limitation is that our analysis was exclusively cortical (except for the thalamic stimulation), yet other subcortical structures such as the basal ganglia and brainstem^{42,84,107–110} are

known to play an important role in anaesthetic-induced and pathological loss of responsiveness³⁰. As this reviewer recalls, the final paper cited, (#30, Bastos et al, 2021, elife) has little to do with pathological loss of responsiveness associated with the basal ganglia. Once again, I urge the authors to make sure that all citation numbers are accurate and added to the text as intended.

Reviewer #2 (Remarks to the Author):

The manuscript makes excellent use of two valuable fMRI macaque datasets to demonstrate global neural signatures associated with conscious states. The authors have addressed many of the concerns I had with the previous version of the manuscript, especially with respect to their statistical analyses and the strength of their claims which now are a better match to the expectations of the field. The current version is greatly improved, providing insight about the neural mechanisms of consciousness likely to be of considerable interest to neuroscientists and clinicians alike. Following some minor revisions, suggested below, I would recommend this paper for publication.

We are grateful for the reviewer's help in improving our manuscript's strength and clarity, and we are pleased to hear that it is now deemed suitable for publication. Below, please find our responses to the remaining requests for minor revisions.

Results:

1. The authors have introduced a new analyses involving multivariate linear regression of arousal scores simultaneously against their measures of consciousness (Gradient Range, Hierarchical integration, and Harmonic Energy). This is marked improvement on the previous figure and much more informative. However, in addition to the reported dominance analyses, the authors should provide more results from this model to bolster the impact of figure 8 and their claims. In general, I would like to see figures displaying the regression slopes and the fitted trend to the raw data. In the current version of the manuscript, the authors have not demonstrated that their arousal score operates linearly, nor that the expected effects of their main predictors should have a strictly linear relationship to arousal. While it is possible to guess what the regressions would look like based on the data present in figures 1B, 2B, 4, & 7, the regressions and their relevant statistics should still be supplied so readers can gauge the strength of association and shape of association to the arousal scores proper.

We have now added linear regression plots as requested (Figures S15 and S18). Please also note that we complement our regression analysis with a classification analysis, which we repeated at different arousal thresholds (Figure S16).

2. Displaying the results of the dominance analyses in bar graphs seems like a strange choice, specifically because the % relative importance is not independent between model predictors, but rather cumulative to 100%. Instead, it would be more appropriate to present the data as a pie chart. The same should apply to the supplemental figures.

We have now replaced the bar charts with pie charts, as requested.

Figure 8. Relating structural and functional eigenmodes of the brain to changes in arousal scores induced by anaesthesia and thalamic deep brain stimulation. a Dominance analysis compares all possible models obtained from distinct combinations of predictors, to distribute the variance explained between the predictors, in terms of percentage of relative importance (represented as pie chart). b We establish the statistical significance of our model by comparing the empirical variance explained (R^2) against a null distribution of R^2 obtained from repeating the multiple regression with randomly reassigned arousal scores.

Discussion:

1. I approve of the Authors more conservative decision, in light of the ketamine data, to discuss their results with respect to sensory disconnection from the environment (SDE) as opposed to consciousness per se. I also agree that they are likely correct that full unconsciousness (lack of experience) likely occurred at some point for the macaques under propofol anesthesia, though this cannot easily be confirmed. However, the bulk of the argument, as outlined in paragraph 4 of the discussion is that ketamine cannot be linked to true unconsciousness because of the high degree of vivid dreaming reported after ketamine anesthesia, but that sevoflurane and propofol probably do produce true unconsciousness. A number of studies have now shown evidence of dreaming under different kinds of anesthesia, but the current presentation of the argument is a little messy. It is likely more accurate to say that propofol and sevoflurane likely produce epochs of true unconsciousness, while it is unclear if ketamine does.

We are pleased that the reviewer agrees with our more conservative approach to interpreting our results, and we thank the reviewer for helping us to further clarify this point. As suggested, we have rephrased the final sentence of paragraph 4:

Although propofol (which was the drug used for the DBS experiments) and sevoflurane have also been reported to induce occasional dreaming in humans^{69,70} it seems likely that they produce epochs of true unconsciousness, while it is unclear if ketamine does.

2. The authors present ketamine as an NMDA antagonist. Indeed, this is one primary

mechanism of the drug, but I believe it has a number of other less-specific targets especially at anesthetic doses, which produce unresponsiveness. This is important to discuss in paragraph 5 as the authors note differences in the direction of effects at lower and higher doses of the drug. This could be that NMDA channels may mediate psychedelic effects, but that sensory disconnection relies more on the non NMDA targets.

We thank the reviewer for this suggestion, and we have added the following sentence to paragraph 5 in the Discussion:

Biologically, it is possible that these different effects at different doses may be a result of ketamine's engagement of less-specific molecular targets at higher (i.e., anaesthetic) doses, beyond its principal action as NMDA antagonist.

Minor Concerns:

1. There is a typo in the labels of Figure 1. The term Gradient is misspelled on one of the axes.

Thank you for spotting this: we have fixed it.

2. There may be another incorrect citation. In the discussion, the authors write "Another limitation is that our analysis was exclusively cortical (except for the thalamic stimulation), yet other subcortical structures such as the basal ganglia and brainstem^{42,84,107–110} are known to play an important role in anaesthetic-induced and pathological loss of responsiveness³⁰. As this reviewer recalls, the final paper cited, (^{#30}, Bastos et al, 2021, elife) has little to do with pathological loss of responsiveness associated with the basal ganglia. Once again, I urge the authors to make sure that all citation numbers are accurate and added to the text as intended.

Thank you: we have removed the citation in question.